UT-Komaba/22-3

# Measurement-based quantum simulation of Abelian lattice gauge theories

Hiroki  Sukeno

*Department of Physics and Astronomy, State University of New York at Stony Brook, Stony Brook, NY 11794-3840, USA*

Takuya  Okuda

*Graduate School of Arts and Sciences, University of Tokyo, Komaba, Meguro-ku, Tokyo 153-8902, Japan*

(Dated: October 21, 2022)

Numerical simulation of lattice gauge theories is an indispensable tool in high energy physics, and their quantum simulation is expected to become a major application of quantum computers in the future. In this work, for an Abelian lattice gauge theory in $d$ spacetime dimensions, we define an entangled resource state (generalized cluster state) that reflects the spacetime structure of the gauge theory. We show that sequential single-qubit measurements with the bases adapted according to the former measurement outcomes induce a deterministic Hamiltonian quantum simulation of the gauge theory on the boundary. Our construction includes the $(2+1)$-dimensional Abelian lattice gauge theory simulated on three-dimensional cluster state as an example, and generalizes to the simulation of Wegner's lattice models $M_{(d,n)}$ that involve higher-form Abelian gauge fields. We demonstrate that the generalized cluster state has a symmetry-protected topological order with respect to generalized global symmetries that are related to the symmetries of the simulated gauge theories on the boundary. Our procedure can be generalized to the simulation of Kitaev's Majorana chain on a fermionic resource state. We also study the imaginary-time quantum simulation with two-qubit measurements and post-selections, and a classical-quantum correspondence, where the statistical partition function of the model $M_{(d,n)}$ is written as the overlap between the product of two-qubit measurement bases and the wave function of the generalized cluster state.

## CONTENTS

# I. INTRODUCTION

Gauge theory is a foundation of modern elementary particle physics. The numerical simulation of Euclidean lattice gauge theories [1] has been a great success, even in the non-perturbative regime that is hard to study analytically. On the other hand, there are situations such as real-time simulation and finite density QCD where the path integral formulation of lattice gauge theory suffers from the sign problem—a difficulty in the evaluation of amplitudes due to the oscillatory contributions in the Monte-Carlo importance sampling [2–5]. In the Hamiltonian formulation, the dimension of the Hilbert space grows exponentially with the size of the system. The quantum computer is expected to solve this issue, enabling us to simulate the quantum many-body dynamics in principle with resources linear in the system size [6, 7]. The quantum simulation of gauge theory is thus one of the primary targets for the application of quantum computers/simulators, whose studies are fueled by the recent advances in NISQ quantum technologies [8–14].

The goal of this paper is to present a new quantum simulation scheme for lattice gauge theories. Our scheme, which we call *measurement-based quantum simulation* (MBQS), is motivated by the idea of measurement-based quantum computation (MBQC) [15–19]. Just as in the common MBQC paradigm, our procedure consists of two steps: (i) preparation of an entangled resource state and (ii) single-qubit measurements with bases adapted according to the former measurement outcomes. In the usual MBQC, resource states (such as cluster states [15]) are constructed to achieve universal quantum computation. In MBQS, the resource states, the *generalized cluster states* (gCS), are tailored to simulate the gauge theories and reflect their spacetime structure.

Our prototype examples are the $(2 + 1)$-dimensional Ising model and the lattice $\mathbb{Z}_2$ gauge theory [20–22] simulated on appropriate generalized cluster states. Then we extend this idea to Wegner's lattice models $M_{(d,n)}$ [22] that involve higher-form $\mathbb{Z}_2$ gauge fields. It is common in MBQC to identify one of the spatial dimensions as time in gate-based quantum computation. Similarly, we regard the generalized cluster state as a space-time in which the

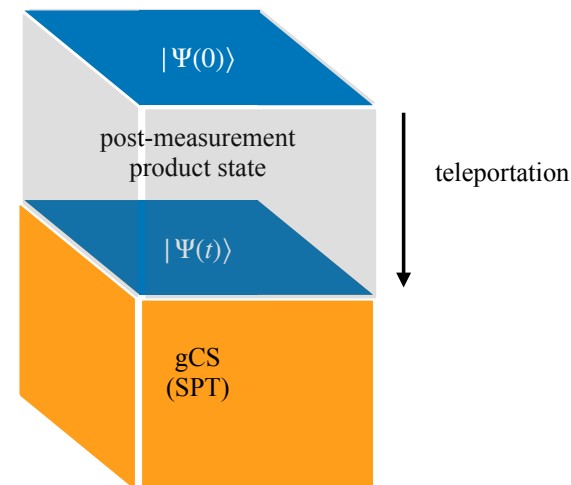

FIG. 1. The concept of MBQS. We start from gCS with the initial wave function at the boundary. After applying single-qubit measurements based on a measurement pattern, we obtain $|\Psi(t)\rangle$, the wave function after the evolution with the Hamiltonian of the model $M_{(d,n)}$, at the boundary of the reduced lattice.

lattice gauge theory lives. See Fig. 1 for an illustration of the concept of MBQS. We also discuss a relation between the generalized cluster state and the partition function, which is a specialized version of a relation between a graph state and the Ising model found in [23, 24]. Our relation implies that the expectation value of the Wilson loop can be estimated via the Hadamard test with a controlled constant-depth circuit (See *e.g.* [25]).

In the Hamiltonian formulation of gauge theory, physical states are required to obey the gauge invariance condition called the Gauss law constraint. In noisy simulations it is expected to be especially important to minimize the effects of errors that violate gauge invariance [26, 27]. In this work we combine the well-known error correcting techniques in MBQC with the analysis of symmetries of the gauge theory and the resource state to formulate an effective method to enforce the Gauss law constraint.

We also present generalizations to gauge group $\mathbb{Z}_N$. Our generalized cluster state is expressed using the cell complexes, and the construction naturally leads us to the simulation of the model $M_{(d,n)}^{(\mathbb{Z}_N)}$, the $\mathbb{Z}_N$ generalization of Wegner's Ising model. As another non-trivial generalization of our MBQS, we present an approach to simulating Kitaev's Majorana chain [28] on a fermionic resource state.

Aside from studies of quantum computational methods, we present more formal aspects of MBQS regarding symmetries. We show that a generalized cluster state possesses a non-trivial symmetry-protected topological (SPT) order [29–36] protected by higher-form symmetries [37, 38]. Further, we propose that MBQS can be regarded as a type of bulk-boundary correspondence be-

tween the resource state and the simulated field theory. Specifically, the gauge symmetry of the boundary simulated theory is promoted to a higher-form symmetry of the bulk resource state. This feature is used in Section IV for the enforcement of the Gauss law constraint in MBQS. On the other hand, as we discuss in Section VII, the boundary simulated theory has a global (higher-form) symmetry, while the bulk resource state can be regarded as a model in which the boundary global symmetry is gauged. It can be seen as a new type of holographic correspondence.

This paper is organized as follows. In Section II, we review the models $M_{(d,n)}$ and introduce the generalized cluster states $gCS_{(d,n)}$. In Section III we provide the measurement-based protocols for the simulation of the Ising model $M_{(3,1)}$ and the $\mathbb{Z}_2$ gauge theory $M_{(3,2)}$. We explain our measurement pattern to execute MBQS. We also study generalization to the imaginary-time evolution. In Section IV, we discuss a procedure to detect certain types of errors and correct them based on the higher-form symmetry of the resource state, enabling us to enforce the Gauss law constraint. In Section V, we discuss generalization to the $\mathbb{Z}_N$ $(n-1)$-form theory in $d$ dimensions as well as the case with the Kitaev's Majorana fermion in $(1+1)$ dimensions. We also make a connection between the Euclidean path integral and the generalized cluster state for the model $M_{(d,n)}^{(\mathbb{Z}_N)}$. In Section VI, we show that the generalized cluster state is an SPT state. In Section VII, we discuss an interplay of the symmetries between the bulk and the boundary in our MBQS. Section VIII is devoted to Conclusions and Discussion. In the appendix we prove some equations used in the main text and discuss supplementary aspects of our MBQS and the generalized cluster states.

## II. LATTICE MODELS AND RESOURCE STATES

### A. Cell complex notation

Let us consider a $d$-dimensional hypercubic lattice. Let $\Delta_0$ be the set of 0-cells (vertices), $\Delta_1$ the set of 1-cells (edges), and $\Delta_2$ the set of 2-cells (faces), and so on. We write $C_i$ $(i = 0, 1, 2, ..., n)$ for the group of $i$-chains $c_i$ with $\mathbb{Z}_2$ coefficients (later this will be generalized to general Abelian groups), i.e., the formal linear combinations

$$c_i = \sum_{\sigma_i \in \Delta_i} a(c_i; \sigma_i)\sigma_i \tag{1}$$

with $a(c_i; \sigma_i) \in \mathbb{Z}_2 = \{0, 1 \bmod 2\}$. Sometimes we regard the chain $c_i$ as the union of the $i$-cells $\sigma_i$ such that $a(c_i; \sigma_i) = 1$. The boundary operator $\partial$ is a linear map $C_{i+1} \to C_i$ such that $\partial \sigma_{i+1}$ is the sum of the $i$-cells that appear on the boundary of $\sigma_i$. We get a chain complex

$$C_n \xrightarrow{\partial} \cdots \xrightarrow{\partial} C_1 \xrightarrow{\partial} C_0 \tag{2}$$

with $\partial^2 = 0$. Similarly, by considering the dual lattice [39], we get the dual chain complex

$$C_n^* \xrightarrow{\partial^*} \cdots \xrightarrow{\partial^*} C_1^* \xrightarrow{\partial^*} C_0^* \tag{3}$$

with $(\partial^*)^2 = 0$. There are natural identifications of $\Delta_i$ ($i$-cells) with $\Delta_{n-i}^*$ (dual $(n-i)$-cells), and $C_i$ ($i$-chains) with $C_{n-i}^*$ (dual $(n-i)$-chains) [40]. We will often consider placing qubits on all the $i$-cells $\sigma_i \in \Delta_i$ for some $i$. Then on each $\sigma_i$ we have Pauli operators $X(\sigma_i)$ and $Z(\sigma_i)$. For each $i$-chain $c_i$ we define

$$X(c_i) := \prod_{\sigma_i \in \Delta_i} X(\sigma_i)^{a(c_i; \sigma_i)},$$
$$Z(c_i) := \prod_{\sigma_i \in \Delta_i} Z(\sigma_i)^{a(c_i; \sigma_i)}. \tag{4}$$

For MBQS we consider a hypercubic lattice in $d$-dimensions, with the $(1, 2, ..., d-1)$-directions periodic and the $d$-th direction open. The value of the $d$-th coordinate $x_d$ ("time") specifies an *artificial time slice*. The boundaries $x_d = 0$ and $x_d = L_d$, where $L_d$ is the linear lattice size in the $d$-th direction, are examples. The bulk state to be introduced later will be the resource state for MBQS. As we proceed in the protocol of MBQS, the state originally defined on the $x_d = 0$ time slice will be teleported to a middle time slice $x_d = j$, where $j \in \{0, 1, \ldots, L_d\}$. Throughout the paper, unless otherwise stated, we use the notation where the bold fonts ($\boldsymbol{\Delta}, \boldsymbol{\sigma}, \boldsymbol{\partial}$, etc.) represent "bulk" quantities related to the $d$-dimensional lattice, whereas the normal fonts ($\Delta, \sigma, \partial$, etc.) are used for the $(d-1)$-dimensional lattice identified with the space of the simulated model.

A cell $\boldsymbol{\sigma}_i$ inside a time slice $x_d = j$ is of the form

$$\boldsymbol{\sigma}_i = \sigma_i \times \{j\}, \tag{5}$$

while a cell $\boldsymbol{\sigma}_i$ extending in the time direction takes the form

$$\boldsymbol{\sigma}_i = \sigma_{i-1} \times [j, j+1]. \tag{6}$$

Sometimes we express a point in the time direction as $pt$ and an interval as $I$.

### B. Model $M_{(d,n)}$

We consider a class of theories described by classical spin degrees of freedom living on $(n-1)$ cells in the $d$-dimensional hypercubic lattice whose action $I$ is given by

$$I[\{S_{\boldsymbol{\sigma}_{n-1}}\}] = -J \sum_{\boldsymbol{\sigma}_n \in \boldsymbol{\Delta}_n} S(\boldsymbol{\partial}\boldsymbol{\sigma}_n), \tag{7}$$

where $J$ is a coupling constant. $S_{\boldsymbol{\sigma}_{n-1}} \in \{+1, -1\}$ is a classical spin variable living on each $(n-1)$-cell $\boldsymbol{\sigma}_{n-1} \in \boldsymbol{\Delta}_{n-1}$ and

$$S(\boldsymbol{c}_i) = \prod_{\boldsymbol{\sigma}_i \in \boldsymbol{\Delta}_i} (S_{\boldsymbol{\sigma}_i})^{a(\boldsymbol{c}_i; \boldsymbol{\sigma}_i)} \tag{8}$$

for a given $i$-chain $\boldsymbol{c}_i = \sum_{\boldsymbol{\sigma}_i \in \boldsymbol{\Delta}_i} a(\boldsymbol{c}_i; \boldsymbol{\sigma}_i) \boldsymbol{\sigma}_i$. This class of theories is called "generalized Ising models" in the literature [22], where the action (7) is viewed as the (classical) Hamiltonian of such a classical spin model. For $n = 2$, (7) is the $\mathbb{Z}_2$ version of the action of Wilson's lattice gauge theory [1], whose degrees of freedom are 1-form gauge fields. When $n \geq 2$, the theory is described by $(n-1)$-form gauge fields, and the action is invariant under a local transformation at each $(n-2)$-cell,

$$\mathcal{G}_{\boldsymbol{\sigma}_{n-2}} : S(\boldsymbol{\sigma}_{n-1}) \to -S(\boldsymbol{\sigma}_{n-1}) \quad \text{for} \quad \boldsymbol{\sigma}_{n-1} \in \partial^* \boldsymbol{\sigma}_{n-2} . \tag{9}$$

This is a higher-form generalization of the standard discrete gauge transformation, which corresponds to the case with $n = 2$. For $n = 1$, $M_{(d,n)}$ is the Ising model in $d$ dimensions.

On infinite lattices, the models $M_{(d,n)}$ and $M_{(d,d-n)}$ are dual to each other [22], generalizing the Kramers-Wannier duality of the two-dimensional Ising model. On finite lattices, the duality changes the global structure of the model. See, *e.g.*, [41].

For each classical spin model in $d$ dimensions, one can construct a quantum spin model defined on a $(d-1)$-dimensional spatial lattice. See [20] and Appendix B. The qubits are placed on $(n-1)$-cells $\sigma_{n-1}$. The Hamiltonian is given by

$$H_{(d,n)} = -\sum_{\sigma_{n-1} \in \Delta_{n-1}} X(\sigma_{n-1}) - \lambda \sum_{\sigma_n \in \Delta_n} Z(\partial \sigma_n) , \tag{10}$$

where we used the notation (4) and $\lambda$ is a coupling constant. Gauge-invariant states $|\psi\rangle$ must satisfy the Gauss law constraint

$$G(\sigma_{n-2})|\psi\rangle = (-1)^{Q(\sigma_{n-2})}|\psi\rangle \tag{11}$$

for any $\sigma_{n-2} \in \Delta_{n-2}$, where $G(\sigma_{n-2})$ is defined as

$$G(\sigma_{n-2}) = X(\partial^* \sigma_{n-2}) , \tag{12}$$

and $Q(\sigma_{n-2}) = 1$ if there is an external charge on the cell $\sigma_{n-2}$ and $Q(\sigma_{n-2}) = 0$ otherwise. Conjugation by the operator $G(\sigma_{n-2})$ generates a gauge transformation in the Hamiltonian picture.

### C. Example 1: $M_{(3,1)}$ (Ising model in $2+1$ dimensions)

The Ising model $M_{(3,1)}$ in $2+1$ dimensions has the Hamiltonian

$$H_{(3,1)} = -\sum_{\sigma_0 \in \Delta_0} X(\sigma_0) - \lambda \sum_{\sigma_1 \in \Delta_1} Z(\partial \sigma_1) . \tag{13}$$

The second term is the nearest neighbor interaction between two vertices connected by edges. We have the following Trotter decomposition of the time evolution $e^{-iH_{(3,1)}t}$:

$$T_{(3,1)}(t) := \left( \prod_{\sigma_0 \in \Delta_0} e^{i\delta t X(\sigma_0)} \prod_{\sigma_1 \in \Delta_1} e^{i\delta t \lambda Z(\partial \sigma_1)} \right)^{x_3} , \tag{14}$$

with $t = x_3 \delta t$.

### D. Example 2: $M_{(3,2)}$ ($\mathbb{Z}_2$ gauge theory in $2+1$ dimensions)

The Hamiltonian of the model $M_{(3,2)}$, the $\mathbb{Z}_2$ gauge theory in $2+1$ dimensions, is [22]

$$H_{(3,2)} = -\sum_{\sigma_1 \in \Delta_1} X(\sigma_1) - \lambda \sum_{\sigma_2 \in \Delta_2} Z(\partial \sigma_2) . \tag{15}$$

The second sum is over plaquettes (faces) $\sigma_2$. The plaquette operator $Z(\partial \sigma_2)$ is the product of Pauli-$Z$ operators on the four edges surrounding $\sigma_2$. The Gauss law constraint is

$$X(\partial^* \sigma_0) = (-1)^{Q(\sigma_0)} , \tag{16}$$

where the left hand side is the product of Pauli-$X$ operators on the edges attached to the vertex $\sigma_0$. $Q(\sigma_0) \in \{0, 1\}$ is the external charge placed at $\sigma_0 \in \Delta_0$. The first-order Trotter approximation of the time evoltion $e^{-iH_{(3,2)}t}$ is given by

$$T_{(3,2)}(t) := \left( \prod_{\sigma_1 \in \Delta_1} e^{i\delta t X(\sigma_1)} \prod_{\sigma_2 \in \Delta_2} e^{i\delta t \lambda Z(\partial \sigma_2)} \right)^{x_3} , \tag{17}$$

with $t = x_3 \delta t$.

### E. Generalized cluster state $\text{gCS}_{(d,n)}$

Here we describe the resource state which we call the generalized cluster state, $\text{gCS}_{(d,n)}$.

We define the eigenvectors of the Pauli operators by

$$Z|0\rangle = |0\rangle , \quad Z|1\rangle = -|1\rangle , \tag{18}$$
$$X|+\rangle = |+\rangle , \quad X|-\rangle = -|-\rangle . \tag{19}$$

We place a qubit on every $(n-1)$-cell $\boldsymbol{\sigma}_{n-1} \in \boldsymbol{\Delta}_{n-1}$ and on every $n$-cell $\boldsymbol{\sigma}_n \in \boldsymbol{\Delta}_n$. For each $n$-chain $\boldsymbol{c}_n = \sum_{\boldsymbol{\sigma}_n \in \boldsymbol{\Delta}_n} a(\boldsymbol{c}_n; \boldsymbol{\sigma}_n) \boldsymbol{\sigma}_n$, we define

$$X(\boldsymbol{c}_n) := \prod_{\boldsymbol{\sigma}_n \in \boldsymbol{\Delta}_n} (X_{\boldsymbol{\sigma}_n})^{a(\boldsymbol{c}_n; \boldsymbol{\sigma}_n)} . \tag{20}$$

We similarly define Pauli $Z$ operators and Pauli operators on $(n-1)$-cells. A general Pauli operator takes the form

$$P = e^{i\alpha} X(\boldsymbol{c}_n) Z(\boldsymbol{c}'_n) X(\boldsymbol{c}_{n-1}) Z(\boldsymbol{c}'_{n-1}) , \tag{21}$$

where $\alpha$ is a c-number phase.

Now we define the stabilizers

$$K(\boldsymbol{\sigma}_n) = X(\boldsymbol{\sigma}_n)Z(\boldsymbol{\partial\sigma}_n), \qquad (22)$$
$$K(\boldsymbol{\sigma}_{n-1}) = X(\boldsymbol{\sigma}_{n-1})Z(\boldsymbol{\partial^*\sigma}_{n-1}). \qquad (23)$$

The generalized cluster state $|\text{gCS}_{(d,n)}\rangle$ is defined by the eigenvalue equations

$$K(\boldsymbol{\sigma}_{n-1})|\text{gCS}_{(d,n)}\rangle = K(\boldsymbol{\sigma}_n)|\text{gCS}_{(d,n)}\rangle = |\text{gCS}_{(d,n)}\rangle$$
$$\text{for all} \quad \boldsymbol{\sigma}_{n-1} \in \boldsymbol{\Delta}_{n-1}, \ \boldsymbol{\sigma}_n \in \boldsymbol{\Delta}_n. \qquad (24)$$

Explicitly, the cluster state can be written as

$$|\text{gCS}_{(d,n)}\rangle = \mathcal{U}_{CZ}|+\rangle^{\otimes(\boldsymbol{\Delta}_{n-1}\sqcup\boldsymbol{\Delta}_n)}. \qquad (25)$$

where $\mathcal{U}_{CZ}$ is the entangler that applies controlled-$Z$ gates ($CZ$ gates) between qubits on adjacent qubits:

$$\mathcal{U}_{CZ} := \prod_{\substack{\boldsymbol{\sigma}_{n-1}\in\boldsymbol{\Delta}_{n-1}\\ \boldsymbol{\sigma}_n\in\boldsymbol{\Delta}_n}} (CZ_{\boldsymbol{\sigma}_{n-1},\boldsymbol{\sigma}_n})^{a(\boldsymbol{\partial\sigma}_n;\boldsymbol{\sigma}_{n-1})} \qquad (26)$$

with $\boldsymbol{\partial\sigma}_n = \sum_{\boldsymbol{\sigma}_{n-1}\in\boldsymbol{\Delta}_{n-1}} a(\boldsymbol{\partial\sigma}_n;\boldsymbol{\sigma}_{n-1})\boldsymbol{\sigma}_{n-1}$. The $CZ$ gate is given by

$$CZ_{c,t} = |0\rangle_c\langle 0| \otimes I_t + |1\rangle_c\langle 1| \otimes Z_t. \qquad (27)$$

It is invariant under the exchange of the control ($c$) and the target ($t$) qubits.

## III. MEASUREMENT-BASED QUANTUM SIMULATION OF GAUGE THEORY

In this section, we introduce the MBQS protocols for the real-time evolution of the Ising model $M_{(3,1)}$ and the gauge theory $M_{(3,2)}$. See [42] for a pedagogical introduction to MBQC.

### A. Simulation of $M_{(3,1)}$

For the simulation of the model $M_{(3,1)}$, we use $\text{gCS}_{(3,1)}$ as the resource state. This is a cluster state whose qubits are placed on 1-cells (edges) and 0-cells (vertices). See Fig. 2 for an illustration. We describe the measurement protocol to simulate the time evolution with Hamiltonian $H_{(3,1)}$ given in (13).

#### 1. Measurement pattern

Our simulation protocol will involve two types (A and B) of measurements. Each measurement realizes a desired unitary operator, multiplied by an extra Pauli operator that depends on the non-deterministic measurement outcome. The desired operator simulates a factor in the Trotterized time evolution operator (14). The extra operator is called a *byproduct operator* and is determined by the measurement outcomes. As we will explain, we can adaptively choose the measurement bases according to the previous outcomes so that the simulated unitary operator is deterministic.

Let us explain the A-type measurement as part of the MBQS. In a two-dimensional layer at $x_3 = j$, we have qubits on the vertices $\sigma_0 \in \Delta_0$ and the edges $\sigma_1 \in \Delta_1$. See Fig. 2 (1). The qubits on the edges are entangled by the $CZ$ gate with the adjacent qubits on $\partial\sigma_1$. The wave function $|\Psi\rangle$ for the qubits on the vertices is arbitrary. Our claim is that the unitary operator

$$U^{(1)}_{(3,1)} := (Z(\partial\sigma_1))^s e^{-i\xi Z(\partial\sigma_1)} \qquad (28)$$

is realized by measuring the qubit on $\sigma_1$ with the basis

$$\mathcal{M}_{(A)} = \left\{ e^{i\xi X}|s\rangle \,\middle|\, s = 0,1 \right\}. \qquad (29)$$

Indeed [43],

$$_{\sigma_1}\langle s|\, e^{-i\xi X_{\sigma_1}} \prod_{\sigma_0\in\Delta_0} CZ^{a(\partial\sigma_1;\sigma_0)}_{\sigma_1,\sigma_0}|+\rangle_{\sigma_1}|\Psi\rangle$$
$$= \frac{1}{\sqrt{2}}(Z(\partial\sigma_1))^s e^{-i\xi Z(\partial\sigma_1)}|\Psi\rangle. \qquad (30)$$

We prove this equation in Appendix A. The Pauli operator $Z(\partial\sigma_1)^s$ is the byproduct operator from this measurement. Up to the byproduct operator and a choice of angle $\xi$, $U^{(1)}_{(3,1)}$ is essentially the time-evolution by the term $Z(\partial\sigma_1)$ in the Ising Hamiltonian (13). We refer to $s$ as the measurement outcome, and the measurement in the basis (29) as A-type.

Next, we explain the B-type measurement. It is defined as the measurement with the basis

$$\mathcal{M}_{(B)} = \left\{ e^{i\xi Z}|\tilde{s}\rangle \,\middle|\, s = 0,1 \right\}, \qquad (31)$$

where $|\tilde{s}\rangle$ is the eigenvector of the $X$ operator with the eigenvalue $(-1)^s$:

$$X|\tilde{s}\rangle = (-1)^s|\tilde{s}\rangle. \qquad (32)$$

In other words, $|\tilde{0}\rangle = |+\rangle$ and $|\tilde{1}\rangle = |-\rangle$. This measurement implements a gate teleportation. To see this, we consider a general state $|\Psi\rangle_1$ and an ancilla $|+\rangle_2$, and we entangle them with the $CZ$ gate. Then we measure the qubit 1 with the basis $\mathcal{M}_{(B)}$. The circuit is given by [44]

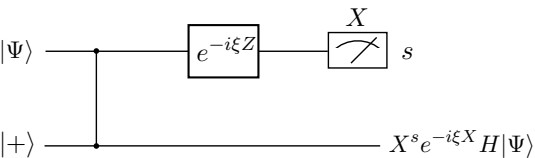

Here the realized unitary operator is

$$U^{(3)}_{(3,1)} = X^s e^{-i\xi X} H. \qquad (33)$$

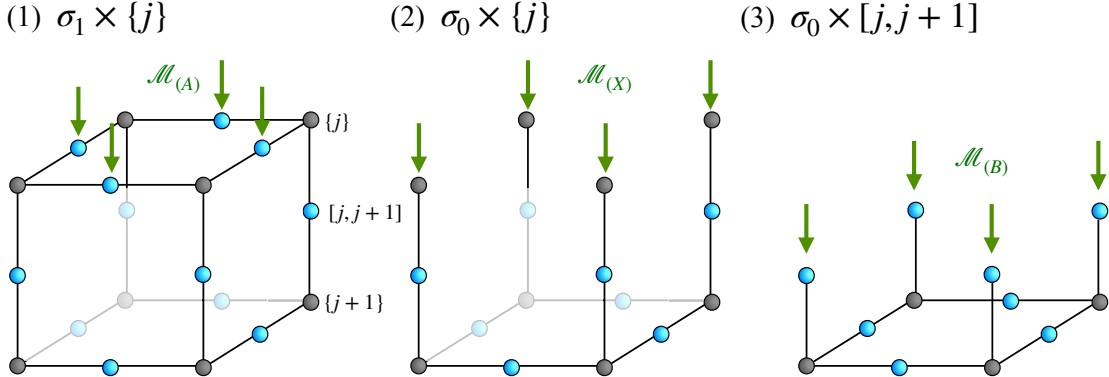

**(1)** $\sigma_1 \times \{j\}$      **(2)** $\sigma_0 \times \{j\}$      **(3)** $\sigma_0 \times [j, j+1]$

FIG. 2. (Color online) The unit cell in the generalized cluster state $\mathrm{gCS}_{(3,1)}$ used to simulate the model $M_{(3,1)}$. The blue (light) balls represent the qubits living on 1-cells $\boldsymbol{\sigma}_1 \in \boldsymbol{\Delta}_1$, while the black (dark) ones are those on 0-cells $\boldsymbol{\sigma}_0 \in \boldsymbol{\Delta}_0$. The green (gray) arrows show the measurement pattern in MBQS.

Indeed,

$$_1\langle \tilde{s}|e^{-i\xi Z_1}CZ_{12}|\Psi\rangle_1|+\rangle_2 = \frac{1}{\sqrt{2}}(X_2)^s e^{-i\xi X_2}H_2|\Psi\rangle_2 .$$
(34)

The proof is given in Appendix A. The Pauli operator $X^s$ is the byproduct operator from this measurement. The special case with the angle $\xi = 0$ will be called X-type, and we use the notations

$$\mathcal{M}_{(X)} = \{|\tilde{s}\rangle \,|\, s = 0, 1 \} ,$$
(35)

$$U_{(X)} = X^s H .$$
(36)

For the time evolution from $x_3 = j$ to $x_3 = j + 1$, the measurement pattern on the three-dimensional clus-

ter state $\mathrm{gCS}_{(3,1)}$ for our MBQS protocol is as follows.

| basis | $\mathcal{M}_{(A)} \rightarrow$ | $\mathcal{M}_{(X)} \rightarrow$ | $\mathcal{M}_{(B)}$ |
|---|---|---|---|
| layer | $pt$ | $pt$ | $I$ |
| 3d cell | $\boldsymbol{\sigma_1}$ | $\boldsymbol{\sigma_0}$ | $\boldsymbol{\sigma_1}$ |
| 2d cell | $\sigma_1$ | $\sigma_0$ | $\sigma_0$ |
| simulation | $U_{(3,1)}^{(1)}$ | $U_{(X)}$ | $U_{(3,1)}^{(3)}$ |
| $(\xi, s)$ | $(\xi_1, s_1)$ | $(0, s_2)$ | $(\xi_3, s_3)$ |

(37)

See Fig. 2. We have a set of measurement outcomes and angles, which depend on locations of cells. We denote them as $\{s_1(\sigma_1 \times \{j\}), s_2(\sigma_0 \times \{j\}), s_3(\sigma_0 \times [j, j+1])\}$ and $\{\xi_1(\sigma_1 \times \{j\}), \xi_3(\sigma_0 \times [j, j+1])\}$, respectively, where the subscripts indicate the order of the corresponding measurements within the time step. To avoid clutters, we often make the cell dependence of these parameters implicit. The total unitary operator for one time step is

$$U_{(3,1)}(\{\xi_i\}) = \prod_{\sigma_0 \in \Delta_0} U_{(3,1)}^{(3)} U_{(X)} \prod_{\sigma_1 \in \Delta_1} U_{(3,1)}^{(1)}$$
(38)

$$= \prod_{\sigma_0 \in \Delta_0} (X(\sigma_0))^{s_3(\sigma_0)} e^{-i\xi_3 X(\sigma_0)} H(\sigma_0)(X(\sigma_0))^{s_2(\sigma_0)} H(\sigma_0) \prod_{\sigma_1 \in \Delta_1} (Z(\partial\sigma_1))^{s_1(\sigma_1)} e^{-i\xi_1 Z(\partial\sigma_1)}$$
(39)

$$= \prod_{\sigma_0 \in \Delta_0} (X(\sigma_0))^{s_3(\sigma_0)} e^{-i\xi_3 X(\sigma_0)} (Z(\sigma_0))^{s_2(\sigma_0)} \prod_{\sigma_1 \in \Delta_1} (Z(\partial\sigma_1))^{s_1(\sigma_1)} e^{-i\xi_1 Z(\partial\sigma_1)}$$
(40)

As in the usual protocols of measurement-based quantum computation, the outcomes of measurements with bases $\mathcal{M}_{(A)}$ and $\mathcal{M}_{(X)}$ should be collected before performing the measurements with $\mathcal{M}_{(B)}$, and the parameter $\xi_3$ for each $\sigma_0$ should be chosen so that the unitary gate of the $X$ rotation is as wanted. Concretely, we choose the

parameters in the first step as follows:

$$\xi_1(\sigma_1) = -\lambda \delta t ,$$
(41)

$$\xi_3(\sigma_0) = -(-1)^{s_2(\sigma_0)}(-1)^{\sum_{\sigma_1 \in \Delta_1} s_1(\sigma_1) a(\partial\sigma_1; \sigma_0)} \delta t .$$
(42)

We use the relation

$$\prod_{\sigma_0 \in \Delta_0} e^{-i\xi_3 X(\sigma_0)} (Z(\sigma_0))^{s_2(\sigma_0)} \prod_{\sigma_1 \in \Delta_1} (Z(\partial\sigma_1))^{s_1(\sigma_1)}$$
$$= \prod_{\sigma_0 \in \Delta_0} (Z(\sigma_0))^{s_2(\sigma_0)} \prod_{\sigma_1 \in \Delta_1} (Z(\partial\sigma_1))^{s_1(\sigma_1)} \prod_{\sigma_0 \in \Delta_0} e^{-i\delta t X(\sigma_0)}$$
$$(43)$$

to propagate the byproduct operators forward. Then $U_{(3,1)}(\{\xi_i\})$ in eq. (40) becomes

$$\Sigma^{(1)}(\{s\}) \prod_{\sigma_0 \in \Delta_0} e^{iX(\sigma_0)\delta t} \prod_{\sigma_1 \in \Delta_1} e^{i\lambda Z(\partial\sigma_1)\delta t} , \qquad (44)$$

where $\Sigma^{(1)}(\{s\})$ is the product of all the byproduct operators from the 1st time step, *i.e.*,

$$\Sigma^{(1)}(\{s\}) = \prod_{\sigma_0} X(\sigma_0)^{s_3} Z(\sigma_0)^{s_2} \prod_{\sigma_1} Z(\partial\sigma_1)^{s_1} \qquad (45)$$

and the remaining product of operators is $T_{(3,1)}(\delta t)$ defined in eq. (14).

In the following steps, we continue with the same measurement pattern, except that the measurement angles are adjusted according to the former measurement outcomes as we propagate the byproduct operators to the frontmost position. After $j$ Trotter steps we have

$$\left( \prod_{k=1}^{j} \Sigma^{(k)}(\{s\}) \right) |\psi(t)\rangle \quad (t = j\delta t) \qquad (46)$$

with $\Sigma^{(k)}(\{s\})$ being the byproduct operators coming from the $k$-th time step and $|\psi(t)\rangle = T_{(3,1)}(t)|\psi(0)\rangle$. Performing another time step gives us

$$U_{(3,1)}(\{\xi_i\}) \left( \prod_{k=1}^{j} \Sigma^{(k)}(\{s\}) \right) |\psi(t)\rangle$$
$$= \left( \prod_{k=1}^{j+1} \Sigma^{(k)}(\{s\}) \right) |\psi(t + \delta t)\rangle \qquad (47)$$

by choosing $\{\xi_i\}$ appropriately according to the preceding byproduct operators. In the end, we have

$$\left( \prod_{k=1}^{L_3} \Sigma^{(k)}(\{s\}) \right) |\psi(T)\rangle \qquad (T = L_3 \delta t) \qquad (48)$$

and the effect of the total byproduct operator can be removed by the post processing.

### B. Simulation of $M_{(3,2)}$

In this section we present the MBQS protocol for the $\mathbb{Z}_2$ lattice gauge theory $M_{(3,2)}$. The resource state is $gCS_{(3,2)}$, whose qubits are placed on 2-cells (faces) and 1-cells (edges), and the entanglers are applied appropriately. This state is also known as the Raussendorf-Bravyi-Harrington (RBH) cluster state [45]. See Fig. 3 for illustration.

#### 1. Measurement pattern

First, let us focus on a 2-cell (face) $\sigma_2$ and the four 1-cells (edges) $\partial\sigma_2$ surrounding it, in a layer at the level $x_3 \in \{1, ..., L_3\}$, As a straightforward generalization of the simulation of the interaction term in (28), we find that the simulation of the plaquette term with a byproduct operator

$$U_{(3,2)}^{(1)} := (Z(\partial\sigma_2))^s e^{-i\xi Z(\partial\sigma_2)} , \qquad (49)$$

is induced by measuring the qubit on $\sigma_2$ with the basis

$$\mathcal{M}_{(A)} := \left\{ e^{i\xi X} |s\rangle \,\middle|\, s = 0, 1 \right\} . \qquad (50)$$

In other words,

$$_{\sigma_2}\langle s| e^{-i\xi X_{\sigma_2}} \prod_{\sigma_1 \in \Delta_1} CZ_{\sigma_2, \sigma_1}^{a(\partial\sigma_2; \sigma_1)} |+\rangle_{\sigma_2} |\Psi\rangle$$
$$= \frac{1}{\sqrt{2}} (Z(\partial\sigma_2))^s e^{-i\xi Z(\partial\sigma_2)} |\Psi\rangle . \qquad (51)$$

Here, $|\Psi\rangle$ is a general wave function of qubits defined on 1-cells at $x_3$.

We have already seen in Section III A that we can simulate the $X$-rotation gate with a teleportation. We utilize

$$\mathcal{M}_{(B)} = \left\{ e^{i\xi Z} |\tilde{s}\rangle \,\middle|\, s = 0, 1 \right\} , \qquad (52)$$
$$U_{(3,2)}^{(4)} := X^s e^{-i\xi X} H \qquad (53)$$

and

$$\mathcal{M}_{(X)} = \left\{ |\tilde{s}\rangle \,\middle|\, s = 0, 1 \right\} , \qquad (54)$$
$$U_{(X)} := X^s H . \qquad (55)$$

Now we present our measurement pattern in Table I. The measurement basis $\mathcal{M}_{(3,2)}^{(G)}$, which we will explain below, is used to enforce gauge invariance, See Fig. 3 for an illustration.

#### 2. Enforcing gauge invariance

In simulating gauge theory dynamics on real devices, a time evolution that violates gauge-invariance, or more precisely the Gauss law constraint (16), would be induced due to the noise that occurs to physical qubits. We consider the following two methods to enforce gauge invariance. (i) Add an energy cost term for violating gauge invariance to the Hamiltonian of the simulated gauge theory:

$$H = -\sum_{\sigma_1 \in \Delta_1} X(\sigma_1) - \lambda \sum_{\sigma_2 \in \Delta_2} Z(\partial\sigma_2)$$
$$- \Lambda \sum_{\sigma_0 \in \Delta_0} (-1)^{Q(\sigma_0)} X(\partial^* \sigma_0) , \qquad (56)$$

| basis | $\mathcal{M}_{(A)}$ | $\rightarrow$ $\mathcal{M}_{(X)}$ | $\rightarrow$ $\mathcal{M}_{(A)}$ $\left( \mathcal{M}_{(X)} \right.$ | $\rightarrow$ $\mathcal{M}_{(B)}$ |
|---|---|---|---|---|
| layer | $pt$ | $pt$ | $I$ | $I$ |
| 3d cell | $\boldsymbol{\sigma_2}$ | $\boldsymbol{\sigma_1}$ | $\boldsymbol{\sigma_1}$ | $\boldsymbol{\sigma_2}$ |
| 2d cell | $\sigma_2$ | $\sigma_1$ | $\sigma_0$ | $\sigma_1$ |
| simulation | $U_{(3,2)}^{(1)}$ | $U_{(X)}$ | Gauss law | $U_{(3,2)}^{(2)}$ |
| $(\xi, s)$ | $(\xi_1, s_1)$ | $(0, s_2)$ | $(\xi_3, s_3)$ $\left( (0, s_3) \right.$ | $(\xi_4, s_4)$ |

TABLE I. Measurement pattern for (2+1)d $\mathbb{Z}_2$ gauge theory. In the third step, $\mathcal{M}_{(A)}$ is used for **Method (i)** (with energy cost terms) and $\mathcal{M}_{(X)}$ is used for **Method (ii)** (with syndromes).

where the coefficient $\Lambda$ is taken so large that the cost term becomes much more significant than the other terms but is small enough so that the Trotter decomposition is justified. We expect, but do not prove, that such a cost term would suppress the violation of gauge invariance. See [26] for the study of a similar cost term. (ii) Actively correct the errors based on the measurement outcomes, much in the spirit of topological quantum memory [46]. In the following we explain how the protocols for the two methods work in the error-free situation. In Section IV, we will explain the protocols in a specific error model in

detail.

**(i) Gauss law enforcement by energy cost**. In the first method, we use the measurement basis $\mathcal{M}_{(3,2)}^{(G)} = \mathcal{M}_{(A)}$ so that it induces

$$U_{(3,2)}^{(G,\text{i})} := (Z(\partial^* \sigma_0))^s e^{-i\xi Z(\partial^* \sigma_0)} \ . \tag{57}$$

In this case, the total unitary for a single time step in a unit 2-cell in 2d becomes

$$
\begin{aligned}
&U_{(3,2)}^{(\text{i})}(\{\xi_i\}) \\
&:= \prod_{\sigma_1 \in \Delta_1} U_{(3,2)}^{(4)} \prod_{\sigma_0 \in \Delta_0} U_{(3,2)}^{(G,\text{i})} \prod_{\sigma_1 \in \Delta_1} U_{(X)} \prod_{\sigma_2 \in \Delta_2} U_{(3,2)}^{(1)} \\
&= \prod_{\sigma_1 \in \Delta_1} X(\sigma_1)^{s_4(\sigma_1)} e^{-i\xi_4 X(\sigma_1)} H(\sigma_1) \prod_{\sigma_0 \in \Delta_0} Z(\partial^* \sigma_0)^{s_3(\sigma_0)} e^{-i\xi_3 Z(\partial^* \sigma_0)} \\
&\quad \prod_{\sigma_1 \in \Delta_1} X(\sigma_1)^{s_2(\sigma_1)} H(\sigma_1) \prod_{\sigma_2 \in \Delta_2} Z(\partial \sigma_2)^{s_1(\sigma_2)} e^{-i\xi_1 Z(\partial \sigma_2)} \\
&= \prod_{\sigma_1 \in \Delta_1} X(\sigma_1)^{s_4(\sigma_1)} e^{-i\xi_4 X(\sigma_1)} \prod_{\sigma_0 \in \Delta_0} X(\partial^* \sigma_0)^{s_3(\sigma_0)} e^{-i\xi_3 X(\partial^* \sigma_0)} \prod_{\sigma_1 \in \Delta_1} Z(\sigma_1)^{s_2(\sigma_1)} \prod_{\sigma_2 \in \Delta_2} Z(\partial \sigma_2)^{s_1(\sigma_2)} e^{-i\xi_1 Z(\partial \sigma_2)} \ ,
\end{aligned}
\tag{58}
$$

We write the wave function with the Trotterized time evolution as

$$|\psi(t)\rangle = T_{(3,2)}^{(\text{i})}(t)|\psi(0)\rangle \ , \tag{59}$$

$$T_{(3,2)}^{(\text{i})}(t) = \left( \prod_{\sigma_1 \in \Delta_1} e^{iX(\sigma_1)\delta t} \prod_{\sigma_0 \in \Delta_0} e^{i\Lambda(-1)^{Q_{\sigma_0}} X(\partial^* \sigma_0)\delta t} \prod_{\sigma_2 \in \Delta_2} e^{i\lambda Z(\partial \sigma_2)\delta t} \right)^j \qquad (t = j\delta t) \tag{60}$$

and denote the by product operators coming from the $j$-th time step as $\Sigma^{(j)}$. Then we obtain the following for $(j+1)$-th step by appropriately choosing $\{\xi_i\}_{i=1,2,4}$:

$$U_{(3,2)}^{(\text{i})}(\{\xi_i\}) \left( \prod_{k=1}^{j} \Sigma^{(k)} \right) |\psi(t)\rangle = \left( \prod_{k=1}^{j+1} \Sigma^{(k)} \right) |\psi(t + \delta t)\rangle \ . \tag{61}$$

**(ii) Gauss law enforcement by error correction**. The second method is well known in the context of the topological MBQC [47–49]. We use the measurement ba-

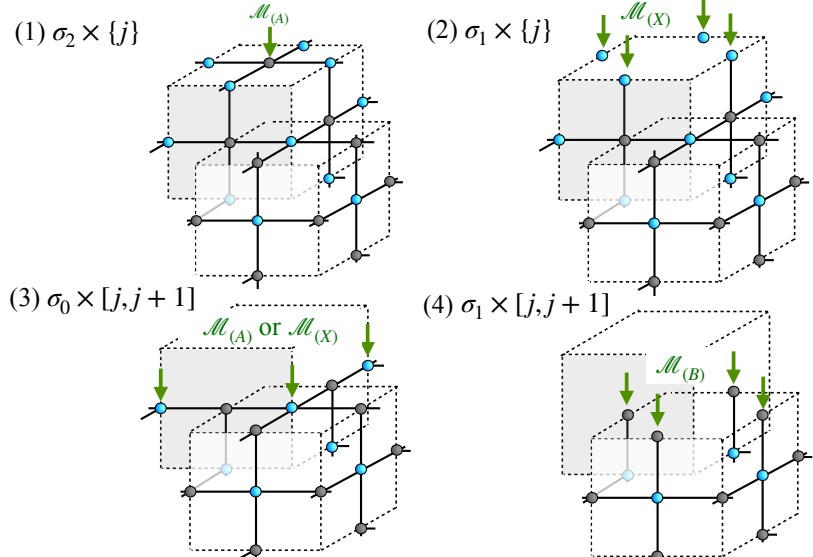

FIG. 3. (Color online) The protocol for the model $M_{(3,2)}$. The black (dark) balls represent the qubits living on 2-cells $\boldsymbol{\sigma}_2 \in \boldsymbol{\Delta}_2$ and the blue (light) ones are those on 1-cells $\boldsymbol{\sigma}_1 \in \boldsymbol{\Delta}_1$.

sis $\mathcal{M}_{(3,2)}^{(G)} = \mathcal{M}_{(X)}$. In this case we obtain

$$
\begin{aligned}
&{}_{\sigma_0}\langle \tilde{s}| \prod_{\sigma_1 \in \Delta_1} CZ_{\sigma_0,\sigma_1}^{a(\partial^*\sigma_0;\sigma_1)} |+\rangle_{\sigma_0} |\Psi\rangle \\
&= \frac{1}{2} \sum_{a=0,1} (-1)^{as} (Z(\partial^*\sigma_0))^a |\Psi\rangle .
\end{aligned} \tag{62}
$$

Here $\partial^*\sigma_0$ is a summation of 1-cells (edges) that surround

the 0-cell (vertex) and $|\Psi\rangle$ is a general wave function of qubits defined on 1-cells at the time interval $[j, j+1]$ (2-cells in three dimensions). Thus we obtain

$$
P_{(3,2)}^{(G,\mathrm{ii})} = \frac{1}{2} \sum_{a=0,1} (-1)^{as} (Z(\partial^*\sigma_0))^a . \tag{63}
$$

Then the total operator for a single time step in 2d becomes

$$
\begin{aligned}
&\prod_{\sigma_1 \in \Delta_1} U_{(3,2)}^{(4)} \prod_{\sigma_0 \in \Delta_0} P_{(3,2)}^{(G,\mathrm{ii})} \prod_{\sigma_1 \in \Delta_1} U_{(X)} \prod_{\sigma_2 \in \Delta_2} U_{(3,2)}^{(1)} \\
&= \prod_{\sigma_1 \in \Delta_1} X(\sigma_1)^{s_4(\sigma_1)} e^{-i\xi_4 X(\sigma_1)} H(\sigma_1) \prod_{\sigma_0 \in \Delta_0} \left( \frac{1}{2} \sum_{a=0,1} (-1)^{as_3(\sigma_0)} Z(\partial^*\sigma_0)^a \right) \\
&\quad \prod_{\sigma_1 \in \Delta_1} X(\sigma_1)^{s_2(\sigma_1)} H(\sigma_1) \prod_{\sigma_2 \in \Delta_2} Z(\partial\sigma_2)^{s_1(\sigma_2)} e^{-i\xi_1 Z(\partial\sigma_2)} \\
&= \prod_{\sigma_1 \in \Delta_1} X(\sigma_1)^{s_4(\sigma_1)} e^{-i\xi_4 X(\sigma_1)} \prod_{\sigma_0 \in \Delta_0} \left( \frac{1}{2} \sum_{a=0,1} (-1)^{as_3(\sigma_0)} X(\partial^*\sigma_0)^a \right) \\
&\quad \prod_{\sigma_1 \in \Delta_1} Z(\sigma_1)^{s_2(\sigma_1)} \prod_{\sigma_2 \in \Delta_2} Z(\partial\sigma_2)^{s_1(\sigma_2)} e^{-i\xi_1 Z(\partial\sigma_2)} .
\end{aligned}
$$

$$\tag{64}$$
$$\tag{65}$$

The operator

$$
P(\sigma_0; s) := \frac{1}{2} \sum_{a=0,1} (-1)^{as(\sigma_0)} X(\partial^*\sigma_0)^a \tag{66}
$$

is a projector that restricts the measurement outcome

$s_3(\sigma_0)$ to be the eigenvalue of $X(\partial^*\sigma_0)$ of the simulated state. At the $j$-th time step ($j \geq 0$), assume that the measurement outcome was $s_3(\sigma_0 \times [j, j+1]) = x(\sigma_0)$ with $x(\sigma_0) \in \{0, 1\}$. Then at the $(j+1)$-th time step, since the $Z$ byproduct operators from the measurements at $\sigma_1 \times \{j+1\}$ may flip the eigenvalue of $X(\partial^*\sigma_0)$, the

measurement outcome becomes $s_3(\sigma_0 \times [j+1, j+2]) =$ $x(\sigma_0) + \sum_{\sigma_1 \in \Delta_1} a(\partial^* \sigma_0; \sigma_1) s_2(\sigma_1 \times \{j+1\})$. Therefore we obtain the following relation for the error-free MBQS:

$$s_3(\sigma_0 \times [j, j+1]) + s_3(\sigma_0 \times [j+1, j+2]) + \sum_{\sigma_1 \in \Delta_1} a(\partial^* \sigma_0; \sigma_1) s_2(\sigma_1 \times \{j+1\}) = 0 \qquad (j \geq 0) . \qquad (67)$$

In Section IV, we introduce an error model and consider an error correction scheme, and we will see that the left-hand side of the relation above serves as a syndrome for the error correction.

We remark that the byproduct operators are handled in the same manner as before. The parameters $\{\xi_1, \xi_4\}$ are chosen accordingly to the former measurement outcomes.

### C. MBQS of imaginary time evolution

The properties of the ground state of a gauge theory can be an interesting target of study. The imaginary-time evolution with Hamiltonian $H$ can be used to prepare the ground state via

$$|GS\rangle = \lim_{\tau \to \infty} \frac{e^{-\tau H}}{\text{Tr}(e^{-\tau H})} |\Psi_{\text{in}}\rangle \qquad (68)$$

from a generic initial state $|\Psi_{\text{in}}\rangle$. However, it is non-trivial to implement the imaginary-time evolution on a quantum computer because $e^{-\tau H}$ is not unitary. In this subsection, we wish to explain how we can perform the imaginary-time evolution of the gauge theory $M_{(3,2)}$ in MBQS by including ancillary qubits and allowing us to implement two-qubit measurements. A method to realize any linear operator for qudit systems using measurements is given in the literature, see e.g. [25, 50, 51][52].

Let us consider $gCS_{(3,2)}$, the cluster state on the three-dimensional lattice with qubits on the 1- and 2-cells. For each 2-cell $\boldsymbol{\sigma}_2$, we consider a qubit on its copy $\tilde{\boldsymbol{\sigma}}_2$ and attach the state $|0\rangle_{\tilde{\boldsymbol{\sigma}}_2}$ as a direct product to the cluster state:

$$|gCS_{(3,2)}\rangle \otimes |0\rangle^{\otimes \tilde{\boldsymbol{\Delta}}_2} . \qquad (69)$$

Here $\tilde{\boldsymbol{\Delta}}_2$ is a copy of the set of 2-cells $\boldsymbol{\Delta}_2$.

For the plaquette interaction with measurement at $\boldsymbol{\sigma}_2 = \sigma_2 \times pt$, we generalize the single-qubit A-type measurement basis (50) to a set of the two-qubit measurement basis that includes

$$|\phi_1^A\rangle = e^{-\alpha} \left( e^{\alpha X_a} |0,0\rangle_{a,b} + \sqrt{\sinh(2|\alpha|)} |-, 1\rangle_{a,b} \right) , \qquad (70)$$

$$|\phi_2^A\rangle = e^{-\alpha} \left( e^{\alpha X_a} |1,0\rangle_{a,b} - \text{sgn}(\alpha) \sqrt{\sinh(2|\alpha|)} |-, 1\rangle_{a,b} \right) , \qquad (71)$$

where $a$ and $b$ refer to two qubits. They are both normalized and mutually orthogonal, $\langle \phi_i^A | \phi_j^A \rangle = \delta_{ij}$. Two other states $|\phi_{3,4}^A\rangle$ can be constructed so that the entire basis $\{|\phi_j^A\rangle\}_{j=1}^4$ is orthonormal. When the measurement is successful, i.e. if the outcome is $|\phi_j^A\rangle$ with $j = 1$ or $j = 2$, the non-unitary operator

$$V_{(\alpha, M)} := \frac{1}{\sqrt{2}} e^{-\alpha} Z(\partial \sigma_2)^s e^{\alpha Z(\partial \sigma_2)} , \qquad (72)$$

with $s = j - 1$ is implemented. Indeed, we have the equality

$$_{\sigma_2, \tilde{\sigma}_2} \langle \phi_j^A | \prod_{\sigma_1 \in \Delta_1} CZ_{\sigma_2, \sigma_1}^{a(\partial \sigma_2; \sigma_1)} |+\rangle_{\sigma_2} |0\rangle_{\tilde{\sigma}_2} |\Psi\rangle$$

$$= \frac{1}{\sqrt{2}} e^{-\alpha} Z(\partial \sigma_2)^s e^{\alpha Z(\partial \sigma_2)} |\Psi\rangle , \qquad (73)$$

where $|\Psi\rangle$ is a general wave function defined on 1-cells within a time slice. See Fig. 4 for an illustration. Writing the cluster state with free ancillary qubits in the middle of simulation as $|\varphi\rangle$, the probability of obtaining $j = 1$ or $j = 2$ is

$$\text{prob}(j; \alpha) := \text{Tr} \left( |\phi_j^A\rangle_{(\sigma_2, \tilde{\sigma}_2)} {}_{(\sigma_2, \tilde{\sigma}_2)}\langle \phi_j^A | \cdot |\varphi\rangle\langle\varphi| \right) \qquad (74)$$

$$= \frac{e^{-2\alpha}}{2} \text{Tr} \left( e^{2\alpha Z(\partial \sigma_2)} |\bar\varphi\rangle\langle\bar\varphi| \right) , \qquad (75)$$

where $|\bar\varphi\rangle$ is defined via the relation

$$|\varphi\rangle = \prod_{\sigma_1 \in \Delta_1} CZ_{\sigma_2, \sigma_1}^{a(\partial \sigma_2; \sigma_1)} |+\rangle_{\sigma_2} |0\rangle_{\tilde{\sigma}_2} |\bar\varphi\rangle . \qquad (76)$$

Thus we have $\frac{e^{-4\alpha}}{2} \leq \text{prob}(j; \alpha) \leq \frac{1}{2}$. Thus, the probability of finding either $j = 1, 2$ becomes exponentially close to 1 when $\alpha$ is small:

$$\text{prob}(1; \alpha) + \text{prob}(2; \alpha) \gtrsim e^{-4\alpha} . \qquad (77)$$

This means that as we take the Trotter step small, the probability of success for each measurement is exponentially close to 1. On the other hand, the success probability of the entire algorithm becomes small as we increase the total time to be simulated.

For the imaginary time evolution corresponding to the $X$ term in the Hamiltonian (15), we generalize the B-type measurement basis (52) to a two-qubit measurement

basis $\{|\phi_j^B\rangle\}_{j=1}^4$ including

$$|\phi_1^B\rangle = e^{-\alpha}\left(e^{\alpha Z_a}|+,0\rangle_{a,b} + \sqrt{\sinh(2|\alpha|)}|1,1\rangle_{a,b}\right) , \tag{78}$$

$$|\phi_2^B\rangle = e^{-\alpha}\big(e^{\alpha Z_a}|-,0\rangle_{a,b}$$
$$- \mathrm{sgn}(\alpha)\sqrt{\sinh(2|\alpha|)}|1,1\rangle_{a,b}\big) , \tag{79}$$

which satisfy $\langle\phi_i^B|\phi_j^B\rangle = \delta_{ij}$. We measure with this basis the two-qubits $a, b$ in the state $|\varphi\rangle = CZ_{a,c}|\Psi\rangle_a|0\rangle_b|+\rangle_c$. See Fig. 4. We obtain for $j = 1, 2$ $(s = 0, 1)$

$$_{a,b}\langle\phi_j^B|\left(CZ_{a,c}|\Psi\rangle_a|0\rangle_b|+\rangle_c\right)$$
$$= \frac{1}{\sqrt{2}}\,e^{-\alpha}(X_c)^s e^{\alpha X_c}H_c|\Psi\rangle_c , \tag{80}$$

where $a = \sigma_1 \times I$ is a 1-cell stretching in the time direction, $b = \tilde{a}$ is its ancillary qubit, and $c = \sigma_1 \times pt$ is the 1-cell in the next time slice. The probability to find either $j = 1, 2$ is $\mathrm{prob}(1, \alpha) + \mathrm{prob}(2, \alpha) \gtrsim e^{-4\alpha}$.

## IV. ENFORCEMENT OF GAUSS LAW CONSTRAINT AGAINST ERRORS

In Section III B, we presented two measurement patterns for the MBQS of the model $M_{(3,2)}$, the $\mathbb{Z}_2$ gauge theory in $(2+1)$ dimensions. The two, Methods (i) and (ii), differed in how to deal with the Gauss law constraint. The protocols for the two methods were explained assuming that there were no errors. In this section, we analyze the effect of errors with the following simplified error model where we assume that the measurement is always perfect while the resource state may be affected by the phase and bit flip errors. Namely, we consider a faulty resource state

$$|\psi^E\rangle := Z(\boldsymbol{e}_1)Z(\boldsymbol{e}_2)X(\boldsymbol{e}_1')X(\boldsymbol{e}_2')|\psi\rangle , \tag{81}$$

where

$$|\psi\rangle = \mathcal{U}_{CZ}\left(|\psi_{2\mathrm{d}}\rangle \otimes |+\rangle_{\mathrm{bulk}}\right) \tag{82}$$

and $\boldsymbol{e}_n$ $(\boldsymbol{e}_n')$ are the $n$-chains which support $Z$ $(X)$ errors. We assume that the state $|\psi_{2\mathrm{d}}\rangle$ is defined on the plane $x_3 = 0$, and that $\boldsymbol{e}_1$ does not include cells in $\Delta_1 \times \{L_3\} \cup \Delta_0 \times [L_3 - 1, L_3]$ [53]. Here we use abbreviated notations such as $\{\sigma_0 \times \{j\} | \sigma_0 \in \Delta_0\} =: \Delta_0 \times \{j\}$. The ideas presented here can be generalized to the MBQS of imaginary-time evolution.

### A. Gauss law enforcement by error correction

Here we explain how the gauge invariance violation of the simulated state can be suppressed by syndrome measurements and error correction. The essence of ideas is adopted from the literature [46–48, 54]. We make a

connection to the symmetry of the resource state, which will play an important role in identifying an SPT order of the state in Section VI.

We focus on the real-time evolution, and examine how the error chains affect the time-evolution operators. We claim that *only the phase error on the bulk 1-chain $Z(\boldsymbol{e}_1)$ leads to the violation of the Gauss law constraint, but one can perform error corrections to suppress contributions that violate the gauge invariance.* The other types of errors $Z(\boldsymbol{e}_2)$, $X(\boldsymbol{e}_1')$, and $X(\boldsymbol{e}_2')$ do not affect the eigenvalue of $X(\partial^*\sigma_0)$. They affect the time evolution but the faulty time evolution operator which we denote as $U^{E+R}(t)$ still commutes with $X(\partial^*\sigma_0)$.

#### 1. Wave function without correction

Let $T_{(3,2)}^{(\mathrm{ii})}(t)$ $(t = x_3\delta t)$ be the time evolution unitary we wish to realize in the model $M_{(3,2)}$ with Method (ii) in the error-free situation:

$$T_{(3,2)}^{(\mathrm{ii})}(t) := \left(\prod_{\sigma_1} e^{-i\tilde{\xi}_4 X(\sigma_1)} \prod_{\sigma_2} e^{-i\tilde{\xi}_1 Z(\partial\sigma_2)}\right)^{x_3} \tag{83}$$

with $\tilde{\xi}_1 = -\lambda\delta t$ and $\tilde{\xi}_4 = -\delta t$. This is simply $T_{(3,2)}(t)$ in (17). (The parameters $\tilde{\xi}_1$ and $\tilde{\xi}_4$ should not be confused with the bare parameters that define the measurement angles, $\{\xi_1, \xi_4\}$, whose signs are chosen depending on the preceding measurement outcomes $\{s_1, s_2, s_4\}$.)

With a faulty resource state, we find that the following state is induced at the new boundary at $x_3$:

$$Z(e_1^{(x_3)})X(e_1'^{(x_3)})Z(b_1^{(x_3)})X(b_1'^{(x_3)})$$
$$\times U^E(t)|\psi_{2\mathrm{d}}\rangle_{\Delta_1 \times \{x_3\}} . \tag{84}$$

Here the operators $Z(b_1^{(x_3)})X(b_1'^{(x_3)})$ are the byproduct operators resulting from the former measurements. The error $Z(e_1^{(x_3)})$ and $X(e_1'^{(x_3)})$ are "extra byproduct operators" caused by errors, given later in eq. (89) and (90). The explicit form of the faulty unitary $U^E(t)$ will be given in eq. (88). In following paragraphs, we explain details of the extra byproduct operators and the faulty time evolution.

#### 2. Direct effect on time evolution

The first effect of errors is *direct* influence of the error chains on the unitaries. We note that, for A-type measurements, an $X$ error changes the measurement outcome compared to the one without it, $s \to s+1$, while a $Z$ error changes the angle, $\xi \to -\xi$. For B-type measurements, an $X$ error changes the angle, $\xi \to -\xi$, while a $Z$ error changes the eigenvalue, $s \to s + 1$. See Table II. For example, the A-type measurement at $\sigma_2 \times \{j\}$ is affected

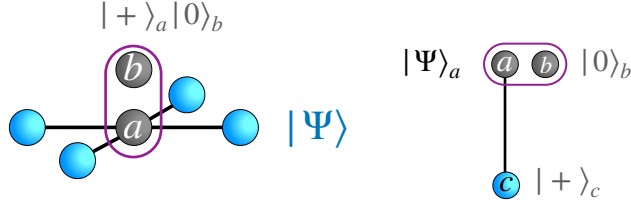

FIG. 4. (Color online) Measurements in the imaginary time MBQS. The gray (dark) balls represent the ancillary qubits to implement the imaginary time evolution, and the black lines indicate the $CZ$ gates for the resource state. (Left) Setup for the A-type measurement to implement $e^{\alpha Z(\partial \sigma_n)}$. We measure the two qubits encircled by the purple (gray) line in the basis $|\phi^A_{j=1,2,3,4}\rangle$. If $|\phi^A_{j=1,2}\rangle$ comes out, the measurement is a success. If it fails, one should throw away the result from the simulation. (Right) Setup for the B-type measurement to implement $e^{\alpha X(\sigma_{n-1})}$. We measure the two qubits encircled by the purple (gray) line in the basis $|\phi^B_{j=1,2,3,4}\rangle$. Likewise, the measurement is a success when we obtain $|\phi^B_{j=1,2}\rangle$.

by a $Z$ error as (color online)

$$\sigma_2 \langle s| e^{-i\xi_1 X_{\sigma_2}} Z(\sigma_2) \prod_{\sigma_1 \in \Delta_1} CZ^{a(\partial\sigma_2;\sigma_1)}_{\sigma_2,\sigma_1} |+\rangle_{\sigma_2}|\Psi\rangle$$

$$= \sigma_2 \langle s|(-1)^s e^{-i(-\xi_1)X_{\sigma_2}} \prod_{\sigma_1 \in \Delta_1} CZ^{a(\partial\sigma_2;\sigma_1)}_{\sigma_2,\sigma_1}|+\rangle_{\sigma_2}|\Psi\rangle$$

$$= \frac{1}{\sqrt{2}}(-1)^s (Z(\partial\sigma_2))^s e^{-i(-\xi_1)Z(\partial\sigma_2)}|\Psi\rangle. \tag{85}$$

The sign change is inherited from $\xi_1$ here to $\tilde{\xi}_1$ in (83). Relative to eq. (83), the sign of the $j$-th (from the right when the power is written as the product of $x_3$ factors) $\tilde{\xi}_1$ in $U^E(t)$ is flipped if $\boldsymbol{e}_2$ in (81) contains $\sigma_2 \times \{j\}$. Similarly the sign of the $j$-th $\tilde{\xi}_4$ is flipped if $\boldsymbol{e}'_2$ contains

### 3. Extra byproduct operators

The other effect is *indirect* and comes from the extra byproduct operators caused by errors, which flip the signs of the angles as we strip them off to the left of the time-evolution unitaries (which were expressed as $Z(e^{(x_3)}_1)X(e'^{(x_3)}_1)$ in eq. (84)). More concretely, a similar calculation as we did in eq. (85) reveals that the $X$ error chains on $\sigma_2 \times \{j\}$ and $Z$ error chains on $\sigma_1 \times \{j\}$ give rise to extra $Z$-byproduct operators, and they flip the sign $\tilde{\xi}_4 \to -\tilde{\xi}_4$ as we move them to the front. Likewise, the $Z$ error chains on $\sigma_1 \times [j, j+1]$ cause extra $X$-byproduct operators that flip the sign $\tilde{\xi}_1 \to -\tilde{\xi}_1$.

$\sigma_1 \times [j, j+1]$.

Now we give the explicit formula of the unitary $U^E(t)$. We define the faulty parameters as

$$\tilde{\xi}^E_1(\sigma_2, j) := \tilde{\xi}_1 (-1)^{\#(\boldsymbol{e}_2 \cap \sigma_2 \times \{j\}) + \#(\boldsymbol{e}_2 \cap \sum^j_{k=0} \partial\sigma_2 \times [k, k+1])}, \tag{86}$$

$$\tilde{\xi}^E_4(\sigma_1, j) := \tilde{\xi}_4 (-1)^{\#(\boldsymbol{e}'_2 \cap \sigma_1 \times [j, j+1]) + \#(\boldsymbol{e}'_2 \cap \sum^j_{k=0} \partial^*\sigma_1 \times \{k\}) + \#(\boldsymbol{e}_1 \cap \sum^j_{k=0} \sigma_1 \times \{k\})}, \tag{87}$$

where the intersection pairing (see footnote below eq. (3)) should be taken by regarding one of the pair of chains as its dual. Then the faulty evolution unitary is given by

$$U^E(t) = \prod^{x_3-1}_{j=0} \left( \prod_{\sigma_1} e^{-i\tilde{\xi}^E_4(\sigma_1,j)X(\sigma_1)} \prod_{\sigma_2} e^{-i\tilde{\xi}^E_1(\sigma_2,j)Z(\partial\sigma_2)} \right), \tag{88}$$

where the product is ordered from right to left as $j$ increases.

We also give explicit formulas for the chains of extra byproduct operators:

$$e^{(j)}_1 = \sum^j_{k=0} \sum_{\sigma_2 \in \Delta_2} \#\left(\boldsymbol{e}'_2 \cap \sigma_2 \times \{k\}\right) \times \partial\sigma_2 \tag{89}$$

$$+ \sum^j_{k=0} \sum_{\sigma_1 \in \Delta_1} \#\left(\boldsymbol{e}_1 \cap \sigma_1 \times \{k\}\right) \times \sigma_1,$$

$$e'^{(j)}_1 = \sum^j_{k=0} \sum_{\sigma_1 \in \Delta_1} \#\left(\boldsymbol{e}_2 \cap \sigma_1 \times [k, k+1]\right) \times \sigma_1. \tag{90}$$

Crucially, the structure of the Pauli operators that appear in exponents of $U^E(t)$ is the same as the error-free time evolution unitary, thus the faulty time-evolution unitary does commute with the Gauss law generator:

$$[G(\partial^*\sigma_0), U^E(t)] = 0. \tag{91}$$

Also note in eq. (89) and (90) that the only contribution that does not commute with $G(\sigma_0)$ is caused by the phase

| | A-type | B-type |
|---|---|---|
| cells | $\sigma_2 \times \{j\}$ | $\sigma_1 \times \{j\}$ <br> $\sigma_1 \times [j, j+1]$ <br> $\sigma_0 \times [j, j+1]$ |
| $X$ error | $(\xi, s+1)$ | $(-\xi, s)$ |
| $Z$ error | $(-\xi, s)$ | $(\xi, s+1)$ |

TABLE II. Effect of errors as changes from $(\xi, s)$.

error $Z(\sigma_1 \times \{j\})$. Besides, the syndrome measurement results are affected only by the error $Z(\sigma_0 \times [j, j+1])$. This confirms our claim we made at the beginning of the section that only phase errors on the 1-chain $Z(\boldsymbol{e}_1)$ cause the effect that is related to the violation of the gauge invariance.

#### 4. Syndromes and symmetry of resource state

Consider first the error-free resource state. We decompose the bulk qubits into $|+\rangle_{\text{bulk}} = |+\rangle_{\Delta_0 \times [0,1]} \otimes |+\rangle_{\text{other}}$.

We write

$$|\psi\rangle = \mathcal{U}_{CZ}(|\psi_{2\text{d}}\rangle_{\Delta_1 \times \{0\}} \otimes |+\rangle_{\Delta_0 \times [0,1]} \otimes |+\rangle_{\text{other}}) . \quad (92)$$

We note that for each $\sigma_0 \times \{0\}$ the product of the $X$-basis measurement results over $\partial^* \sigma_0 \times \{0\}$ and $\sigma_0 \times [0,1]$ is forced to be $(-1)^{Q(\sigma_0)}$. This is because

$$\begin{aligned} |\psi\rangle &= \mathcal{U}_{CZ}\Big((-1)^{Q(\sigma_0)} X(\partial^* \sigma_0)|\psi_{2\text{d}}\rangle_{\Delta_1 \times \{0\}} \otimes X(\sigma_0)|+\rangle_{\Delta_0 \times [0,1]} \otimes |+\rangle_{\text{other}}\Big) \\ &= (-1)^{Q(\sigma_0)} X(\partial^* \sigma_0 \times \{0\}) X(\sigma_0 \times [0,1])|\psi\rangle , \end{aligned} \quad (93)$$

where we used the relation $X(\partial^* \sigma_0)|\psi_{2\text{d}}\rangle = (-1)^{Q(\sigma_0)}|\psi_{2\text{d}}\rangle$. In other words, the perfect resource state satisfies

$$\begin{aligned} |\psi\rangle &= (-1)^{Q(\sigma_0)} X(\boldsymbol{\partial}^* \boldsymbol{\sigma_0})|\psi\rangle \\ &\text{for all} \quad \boldsymbol{\sigma_0} = \sigma_0 \times \{0\} \in \text{boundary 0-cell} . \end{aligned} \quad (94)$$

See Fig. 5 (Left). In terms of the measurement outcomes $\{s_2, s_3\}$, with the perfect resource state, it should be sat-

isfied that

$$\begin{aligned} s_3(\sigma_0 \times [0,1]) + \sum_{\sigma_1 \in \Delta_1} a(\partial^* \sigma_0; \sigma_1) s_2(\sigma_1 \times \{0\}) \\ + Q(\sigma_0) = 0 . \end{aligned} \quad (95)$$

As for the bulk, the corresponding relations are

$$|\psi\rangle = X(\boldsymbol{\partial}^* \boldsymbol{\sigma_0})|\psi\rangle \quad \text{for all} \quad \boldsymbol{\sigma_0} \in \text{bulk 0-cell} \quad (96)$$

(see Fig. 5 (Right)) and

$$s_3(\sigma_0 \times [j, j+1]) + s_3(\sigma_0 \times [j+1, j+2]) + \sum_{\sigma_1 \in \Delta_1} a(\partial^* \sigma_0; \sigma_1) s_2(\sigma_1 \times \{j+1\}) = 0 , \quad (97)$$

which is exactly the relation eq.(67). The relation (96) can be regarded as a consequence of a 1-form symmetry of which $X(\boldsymbol{\partial}^* \boldsymbol{\sigma_0})$ is one of generators. In Section VI, we will see that the state $\text{gCS}_{(3,2)}$ belongs to a nontrivial SPT phase protected by this symmetry (together with another 1-form symmetry).

With the $Z$ errors on a 1-chain $\boldsymbol{e}_1$, the relations eq. (95) and (97) are modified as follows:

$$s_3(\sigma_0 \times [0,1]) + \sum_{\sigma_1 \in \Delta_1} a(\partial^* \sigma_0; \sigma_1) s_2(\sigma_1 \times \{0\}) + Q(\sigma_0) = \#(\boldsymbol{\partial}^* \boldsymbol{\sigma_0} \cap \boldsymbol{e}_1) \quad (98)$$

with $\boldsymbol{\sigma_0} = \sigma_0 \times \{0\}$, and

$$s_3(\sigma_0 \times [j, j+1]) + s_3(\sigma_0 \times [j+1, j+2]) + \sum_{\sigma_1 \in \Delta_1} a(\partial^* \sigma_0; \sigma_1) s_2(\sigma_1 \times \{j+1\}) = \#(\boldsymbol{\partial}^* \boldsymbol{\sigma_0} \cap \boldsymbol{e}_1) \quad (99)$$

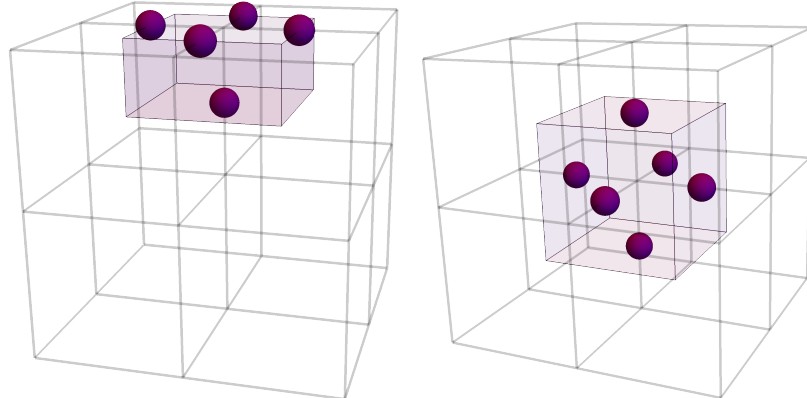

FIG. 5. (Color online) (Left) The top layer corresponds to the boundary $x_3 = 0$. The five balls represent the 1-cells contained in $\boldsymbol{\partial}^*\boldsymbol{\sigma_0}$ in eq. (94). The shaded planes represent the dual boundary of the dual 3-cell truncated at the boundary. (Right) The six balls represent the 1-cells contained in $\boldsymbol{\partial}^*\boldsymbol{\sigma_0}$ in eq. (96). The shaded planes represent the dual boundary of the dual 3-cell.

with $\boldsymbol{\sigma_0} = \sigma_0 \times \{j+1\}$ and $j \geq 0$.

The combination of measurement outcomes for $X(\boldsymbol{\partial}^*\boldsymbol{\sigma_0})$ on the left hand side of eq. (98) and (99) serves as the syndrome for our error correction, and we use these relations to infer the locations of the endpoints of the error chains. When the overlap of a $Z$ error 1-chain with $\boldsymbol{\partial}^*\boldsymbol{\sigma_0}$ is even, the error does not change the sign of the eigenvalue of $X(\boldsymbol{\partial}^*\boldsymbol{\sigma_0})$. However, the eigenvalue is flipped when the overlap is odd. Therefore one can identify the endpoints (0-cells) of the $Z$ error 1-chains using the $X$-measurement results on 1-chains (Fig. 6 (a)). Based on the locations of the 0-cells, one can construct the recovery 1-chains $\boldsymbol{r}_1 \in \boldsymbol{\Delta}_1$ by connecting them with the shortest paths. Importantly, the recovery chain satisfies $\boldsymbol{\partial}(\boldsymbol{r}_1 + \boldsymbol{e}_1) = 0$. One can use the minimal-weight perfect matching algorithm [55, 56], for example, to find such paths (Fig. 6 (b)).

### 5. Final simulated state

We divide the total time steps into $M$ sets of steps: $L_3 = \ell_1 + \cdots + \ell_M$, with $M$ and $\ell_i$ both positive integers. We define a stack $\mathcal{S}_i$ of layers (consisting of cells that are relevant for error correction) as

$$\mathcal{S}_i = \bigcup_{j=\ell_1+\ldots+\ell_{i-1}}^{\ell_1+\ldots+\ell_i-1} \mathcal{L}_j \qquad (i = 1, ..., M) \qquad (100)$$

$$\mathcal{L}_j = (\Delta_0 \cup \Delta_1) \times \{j\} \cup \Delta_0 \times [j, j+1] \ . \qquad (101)$$

Here we define $\ell_0 = 0$ and use abbreviated notations such as $\{\sigma_0 \times \{j\} | \sigma_0 \in \Delta_0\} =: \Delta_0 \times \{j\}$. We choose $\{\ell_i\}$ so that in every $\mathcal{S}_i$ we have an even number of endpoints of $Z$ error chains. Once we perform the MBQS measurements on a stack $\mathcal{S}_i$, we analyze the measurement outcomes at 1-cells in it, as explained above. One can construct

the recovery 1-chains by connecting 0-cells at which the error is detected via relations eq. (98) and (99), within $\mathcal{S}_i$. When the error is on a 1-chain $\sigma_0 \times [j, j+1]$ in the last layer of $\mathcal{S}_i$, the sum of the error chain and the recovery chain may not be a 1-cycle. Continuing construction of recovery chains for $M$ steps, however, the sum of the total recovery chain and the total recovery chain becomes a 1-cycle, given the assumption that there is no $Z$ errors at $\Delta_1 \times \{L_3\} \cup \Delta_0 \times [L_3 - 1, L_3]$.

We treat the recovery chains constructed in $\mathcal{S}_i$ as the same manner as the byproduct operators in the simulation after $\mathcal{S}_i$. In particular, the parameter $\xi_4$ is adjusted according to the recovery chain, which would suppress the impact of errors. Let $r_1^{(j)}$ be the projection of $\boldsymbol{r}_1$ to the boundary, $\Delta_1 \times \{j\}$ (Fig. 6 (c)):

$$r_1^{(j)} = \sum_{\sigma_1 \in \Delta_1} \# \left( \boldsymbol{r}_1 \cap \sum_{k=0}^{j} \sigma_1 \times \{k\} \right) \times \sigma_1 \ .$$

Importantly, the sum of projected error and recovery chains is a 1-cycle on the time slice at $x_3 = L_3$,

$$z_1^{(L_3)} = r_1^{(L_3)} + e_1^{(L_3)} \ . \qquad (102)$$

When we reach $x_3 = L_3$, we post-process the byproduct operator as well as the final recovery chain. The final state is thus

$$|\psi_{\text{fin}}\rangle = Z(z_1^{(L_3)})X(e_1'^{(L_3)})U^{E+R}(L_3\delta t)|\psi_{2\text{d}}\rangle_{\Delta_1 \times \{L_3\}} \ , \qquad (103)$$

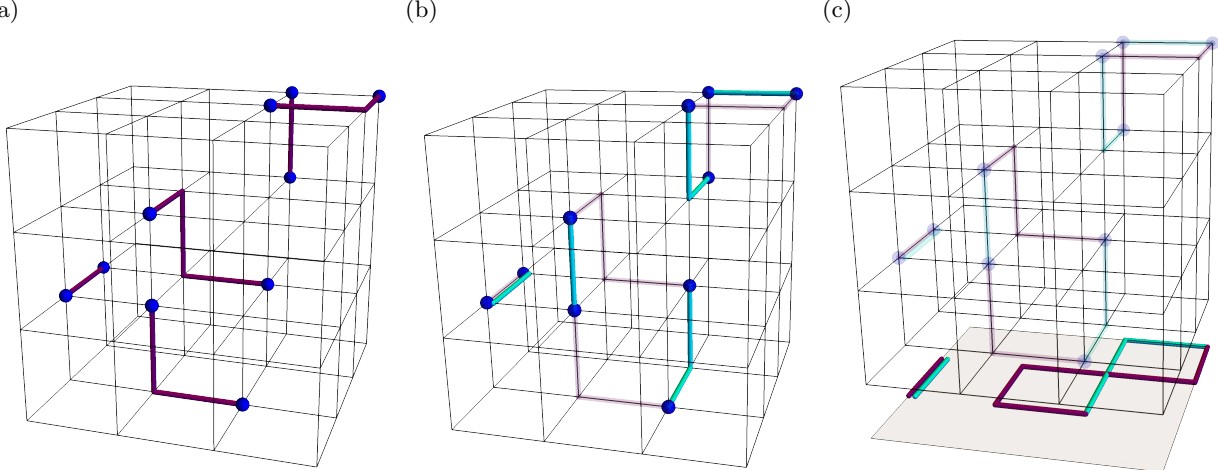

FIG. 6. (Color online) An example of the Gauss law enforcement. The time direction is vertical from top to bottom. (a) The purple (black) lines represent $Z(\boldsymbol{e}_1)$, the $Z$ errors on 1-cells. The blue (black) balls represent 0-cells where the number of error chains surrounding it is odd, meaning that the set of $X$ eigenvalues does not satisfies the symmetry equations: $X(\boldsymbol{\partial}^*\boldsymbol{\sigma_0}) \neq 1$ (bulk) or $X(\boldsymbol{\partial}^*\boldsymbol{\sigma_0}) \neq (-1)^{Q(\sigma_0)}$ (boundary). (b) The cyan (gray) lines are the 1-chains that connect two balls with the shortest paths, which we use as recovery chains $Z(\boldsymbol{r}_1)$. (c) The error and recovery chains are projected to the final time slice. The net contribution from the errors and their recovery chain is a 1-cycle, $Z(z_1)$.

where $U^{E+R}(t)$ is given by

$$
\begin{aligned}
&U^{E+R}(t) \\
&= \prod_{j=0}^{x_3-1} \left( \prod_{\sigma_1} e^{-i\tilde{\xi}_4^{E+R}(\sigma_1,j)X(\sigma_1)} \prod_{\sigma_2} e^{-i\tilde{\xi}_1^{E+R}(\sigma_2,j)Z(\partial\sigma_2)} \right),
\end{aligned}
\tag{104}
$$

$$
\tilde{\xi}_1^{E+R}(\sigma_2,j) := \tilde{\xi}_1^E(\sigma_2,j) ,
\tag{105}
$$

$$
\tilde{\xi}_4^{E+R}(\sigma_1,j) := \tilde{\xi}_4^E(\sigma_1,j) \cdot (-1)^{\#(\boldsymbol{r}_1 \cap \boldsymbol{\lambda}^{(j)})} ,
\tag{106}
$$

with $t = x_3 \delta t$ and

$$
\boldsymbol{\lambda}^{(j)} := \begin{cases} \sum_{k=0}^{\ell(j)} \sigma_1 \times \{k\} & (j \geq \ell_1) \\ 0 & (j < \ell_1) \end{cases} ,
\tag{107}
$$

where $\ell(j) = \sum_{m=1}^{A_j} \ell_m$ with $A_j$ the largest integer such that $\ell(j) \leq j$.

The quantum state prepared with the simulation with the error-correction procedure thus satisfies the Gauss law constraint. We remark that, if we relax the assumption that there is no error at the final layer, endpoints in $\Delta_0 \times \{L_3\}$ of error chains in the vicinity of the boundary cannot be identified. Also there can be an odd number of endpoints in the final stack $\mathcal{S}_M$. (The latter should be properly handled by selecting an even number of endpoints to be corrected.) In general, due to the presence of the error chain one of whose endpoints is not identified, the sum of the recovery and error chain would not be a 1-cycle, but there would be extra contributions of open 1-chains whose endpoints are at the boundary. Such open chains will violate the Gauss law constraint. When the error rate is small, the errors that cause the violation will be localized near the boundary $x_3 = L_3$.

## B. Gauss law enforcement by energy cost

In the same setup as above, we consider MBQS with the energy cost term. Again, we focus on $Z(\boldsymbol{e}_1)$. Here we consider the time $x_d = j$ and $e_0$ ($e_1$) is the restriction of $\boldsymbol{e}_1$ to $[j, j+1]$ ($\{j\}$). The state after one step of the procedure with the faulty resource state reads

$$
\begin{aligned}
&\prod_{\sigma_1 \in \Delta_1} X(\sigma_1)^{s_4(\sigma_1)} e^{-i\xi_4 X(\sigma_1)} \prod_{\sigma_0 \in \Delta_0} X(\partial^*\sigma_0)^{s_3(\sigma_0)} e^{-i(-1)^{a(\sigma_0;e_0)}\xi_3 X(\partial^*\sigma_0)} \\
&\prod_{\sigma_1 \in \Delta_1} Z(\sigma_1)^{s_2(\sigma_1)+a(\sigma_1;e_1)} \prod_{\sigma_2 \in \Delta_2} Z(\partial\sigma_2)^{s_1(\sigma_2)} e^{-i\xi_1 Z(\partial\sigma_2)} Z(b_1) X(b_1') |\psi_{2d}\rangle_{\Delta_1 \times \{j\}} ,
\end{aligned}
\tag{108}
$$

The angle $\xi_3$ is chosen based on the former measurement outcomes $\tilde{b}_1 := b_1 + \tilde{s}_2$ with $\tilde{s}_2 = \sum_{\sigma_1} s_2(\sigma_1)\sigma_1$:

$$
\xi_3 = -\delta t \, \Lambda \, (-1)^{Q(\sigma_0)} (-1)^{a(\partial^*\sigma_0;\tilde{b}_1)} .
\tag{109}
$$

This choice would give us unitaries that suppress contributions with errors $Z(\sigma_1)^{a(\sigma_1;e_1)}$, if $e_0 = 0$ [57].

## V. GENERALIZATIONS

In this section we generalize the results in Section III in several directions. We generalize the gauge group from $\mathbb{Z}_2$ to $\mathbb{Z}_N$, and at the same time allow the parameters $(d, n)$ of the model $M_{(d,n)}$ to be arbitrary. We also discuss a correspondence between the Euclidean path integral of the lattice gauge theory and the measurement-based quantum simulation of the model. Finally, we propose an MBQS protocol for the Kitaev Majorana chain.

### A. MBQS of $M_{(d,n)}^{(\mathbb{Z}_N)}$

In this subsection We introduce the MBQS protocol for the Abelian lattice gauge theory with $(n-1)$-form fields in $d$ spacetime dimensions. We choose the gauge group to be $\mathbb{Z}_N$ with integer $N \geq 2$. The case with gauge group $\mathbb{R}$ is discussed in Appendix C.

#### 1. State $gCS_{(d,n)}^{(\mathbb{Z}_N)}$

We begin by reviewing the qudit, the $N$-dimensional generalization of the qubit. We define the operators $Z$ and $X$ by

$$Z|a\rangle = \omega^a|a\rangle, \quad X|a\rangle = |a+1\rangle, \qquad (110)$$

with $\omega := e^{2\pi i/N}$ and $a \in \{0, 1, \dots, N-1 \mod N\}$. They satisfy

$$Z^N = X^N = 1, \, Z^\dagger = Z^{-1}, \, X^\dagger = X^{-1}, \, ZX = \omega XZ. \tag{111}$$

We define the $X$-basis by

$$|\widetilde{a}\rangle := \frac{1}{\sqrt{N}} \sum_b \omega^{-ab}|b\rangle, \qquad (112)$$

which satisfy $X|\widetilde{a}\rangle = \omega^a|\widetilde{a}\rangle$. The analog of the Hadamard operator is the Fourier transform

$$F := \sum_a |\widetilde{a}\rangle\langle a| = \frac{1}{\sqrt{N}} \sum_{a,b} \omega^{-ab}|b\rangle\langle a|. \qquad (113)$$

It satisfies $XF = FZ$. The controlled-$Z$ operator is defined as

$$CZ := \sum_{a,b} \omega^{ab}|ab\rangle\langle ab|. \qquad (114)$$

We generalize the state $|+\rangle$ as

$$|+\rangle := |\widetilde{0}\rangle = \frac{1}{\sqrt{N}} \sum_a |a\rangle. \qquad (115)$$

Let us consider a $d$-dimensional hypercubic lattice $\mathcal{C}$. The $n$-cells in this lattice can be expressed as

$$
\begin{aligned}
\boldsymbol{\sigma}_n &= \boldsymbol{\sigma}_{\alpha_1,\cdots,\alpha_n}(x) \\
&= \left\{ x + \sum_{i=1}^n t_i e_{\alpha_i} \,\middle|\, 0 \leq t_i \leq 1 \quad \forall i \in \{1, \dots, n\} \right\},
\end{aligned}
\tag{116}
$$

where $1 \leq \alpha_1 < \cdots < \alpha_n \leq d$ are the directions in which the cell is stretched, $x$ is the position of the site at a corner of the cell, and $e_{\alpha_i}$ is the unit vector in the $\alpha_i$-th direction. We define the boundary operator as

$$
\begin{aligned}
\boldsymbol{\partial\sigma}_{\alpha_1,\cdots,\alpha_n}(x) = \sum_{i=1}^n (-1)^{i-1} \big( &\boldsymbol{\sigma}_{\alpha_1,\cdots,\widehat{\alpha_i},\cdots,\alpha_n}(x + e_{\alpha_i}) \\
&- \boldsymbol{\sigma}_{\alpha_1,\cdots,\widehat{\alpha_i},\cdots,\alpha_n}(x) \big),
\end{aligned}
\tag{117}
$$

where $\widehat{\alpha_i}$ means $\alpha_i$ is removed from the subscript. One can confirm that $\boldsymbol{\partial}^2 = 0$ holds and that the definition of $\boldsymbol{\partial}$ is consistent with that in simplicial homology [58].

Let $C_k \equiv C_k(\mathcal{C}; \mathbb{Z}_N)$ be the Abelian group consisting of formal finite sums $\sum_{\boldsymbol{\sigma}_k \in \Delta_k} a(\boldsymbol{\sigma}_k)\boldsymbol{\sigma}_k$ with $a(\boldsymbol{\sigma}_k) \in \{0, 1, \dots, N-1 \mod N\} \simeq \mathbb{Z}_N$. We place a qudit on every $(n-1)$-cell $\boldsymbol{\sigma}_{n-1} \in \Delta_{n-1}$ and on every $n$-cell $\boldsymbol{\sigma}_n \in \Delta_n$. For each $n$-chain $\boldsymbol{c}_n \in C_n$ with $\boldsymbol{c}_i = \sum_{\boldsymbol{\sigma}_i \in \Delta_i} a(\boldsymbol{c}_i; \boldsymbol{\sigma}_i)$ $(a(\boldsymbol{c}_i; \boldsymbol{\sigma}_i) \in \{0, 1, \dots, N-1 \mod N\})$, define

$$X(\boldsymbol{c}_n) := \prod_{\boldsymbol{\sigma}_n \in \Delta_n} (X_{\boldsymbol{\sigma}_n})^{a(\boldsymbol{c}_n; \boldsymbol{\sigma}_n)}. \qquad (118)$$

We similarly define Pauli $Z$ operators on $n$-cells and Pauli $X/Z$ operators on $(n-1)$-cells.

The Hamiltonian that defines the generalized cluster state is now given by

$$
\begin{aligned}
H_{\mathcal{C}} = -&\sum_{\boldsymbol{\sigma}_{n-1} \in \Delta_{n-1}} \left( K(\boldsymbol{\sigma}_{n-1}) + K(\boldsymbol{\sigma}_{n-1})^\dagger \right) \\
-&\sum_{\boldsymbol{\sigma}_n \in \Delta_n} \left( K(\boldsymbol{\sigma}_n) + K(\boldsymbol{\sigma}_n)^\dagger \right),
\end{aligned}
\tag{119}
$$

whose ground state is given by the cluster state $|\psi_{\mathcal{C}}\rangle = |\mathrm{gCS}_{(d,n)}^{(\mathbb{Z}_N)}\rangle$ that satisfies

$$K(\boldsymbol{\sigma}_n)|\psi_{\mathcal{C}}\rangle = K(\boldsymbol{\sigma}_{n-1})|\psi_{\mathcal{C}}\rangle = |\psi_{\mathcal{C}}\rangle$$
$$\text{for all } \boldsymbol{\sigma}_n \in \Delta_n, \, \boldsymbol{\sigma}_{n-1} \in \Delta_{n-1}, \qquad (120)$$
$$|\psi_{\mathcal{C}}\rangle = \mathcal{U}_{CZ}|+\rangle^{\otimes(\Delta_n \sqcup \Delta_{n-1})}. \qquad (121)$$

The stabilizers and the unitary $\mathcal{U}_{CZ}$ are now defined as

$$K(\boldsymbol{\sigma}_n) := X_{\boldsymbol{\sigma}_n} Z(\boldsymbol{\partial\sigma}_n), \qquad (122)$$

$$K(\boldsymbol{\sigma}_{n-1}) := X_{\boldsymbol{\sigma}_{n-1}} \prod_{\boldsymbol{\sigma}_n \in \Delta_n} Z(\boldsymbol{\sigma}_n)^{a(\boldsymbol{\partial\sigma}_n; \boldsymbol{\sigma}_{n-1})}, \qquad (123)$$

$$\mathcal{U}_{CZ} := \prod_{\substack{\boldsymbol{\sigma}_n \in \Delta_n \\ \boldsymbol{\sigma}_{n-1} \in \Delta_{n-1}}} CZ_{\boldsymbol{\sigma}_n, \boldsymbol{\sigma}_{n-1}}^{a(\boldsymbol{\partial\sigma}_n; \boldsymbol{\sigma}_{n-1})}. \qquad (124)$$

See Fig. 7 for an illustration with $(d, n) = (3, 2)$.

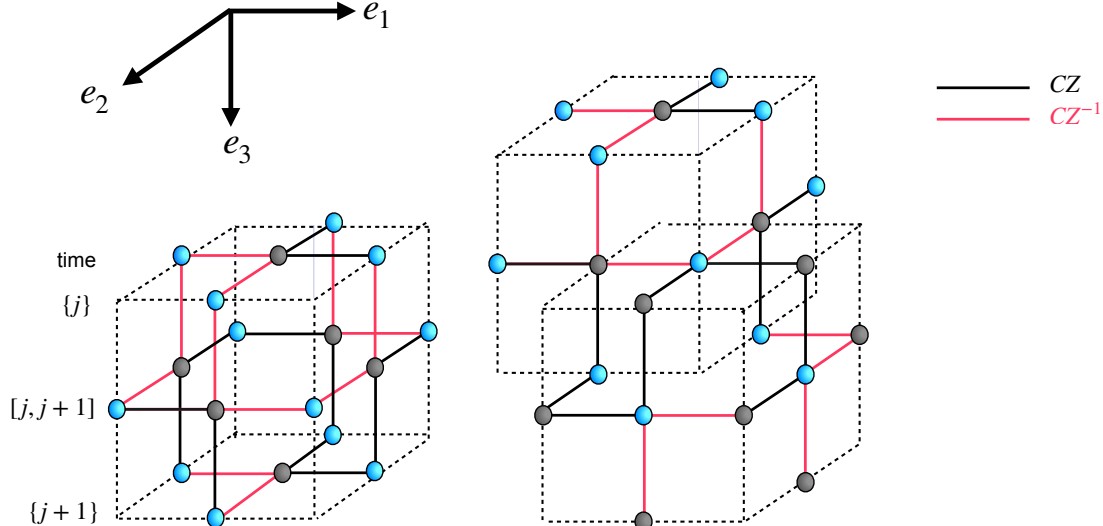

FIG. 7. (Color online) The cluster state for $M_{(3,2)}^{(\mathbb{Z}_N)}$, (2+1)d $\mathbb{Z}_N$ lattice gauge theories. (Left) The primary lattice. The black lines represent the $CZ$ gate, and the pink (gray) lines represent $(CZ^{-1})$ gate. (Right) The primary and the dual lattices.

### 2. Model $M_{(d,n)}^{(\mathbb{Z}_N)}$

For gauge group $\mathbb{Z}_N$, the model $M_{(d,n)}$ is generalized to $M_{(d,n)}^{(\mathbb{Z}_N)}$ defined by the action

$$I[\{u_{\boldsymbol{\sigma}_{n-1}}\}] = -J \sum_{\boldsymbol{\sigma}_n} \left( u(\boldsymbol{\partial\sigma}_n) + u(\boldsymbol{\partial\sigma}_n)^* \right) . \qquad (125)$$

The field $u_{\boldsymbol{\sigma}_{n-1}}$ takes values in $u \in \{1, \omega, ..., \omega^{N-1}\}$. The partition function of $M_{(d,n)}^{(\mathbb{Z}_N)}$ as a classical spin model is given by

$$Z_{(d,n)}^{(\mathbb{Z}_N)} = \sum_{\text{config.}} e^{-\beta I[\{u_{\boldsymbol{\sigma}_{n-1}}\}]} . \qquad (126)$$

### 3. Hamiltonian formulation of $M_{(d,n)}^{(\mathbb{Z}_N)}$

As a quantum lattice model, $M_{(d,n)}^{(\mathbb{Z}_N)}$ is described by the Hamiltonian

$$H = -\frac{1}{2} \sum_{\sigma_{n-1}} (X_{\sigma_{n-1}} + \text{h.c.}) - \frac{\lambda}{2} \sum_{\sigma_n} (Z(\partial\sigma_n) + \text{h.c.}). \qquad (127)$$

Suppose that we have external charge sources with charges $Q_s \in \{0, 1, \ldots, N-1 \bmod d\}$ associated with $(n-2)$-cells $s$. Physical states must obey the Gauss law constraint

$$G(\sigma_{n-2}) = \begin{cases} \omega^{Q_s} & \text{for } \sigma_{n-2} = s, \\ 1 & \text{otherwise,} \end{cases} \qquad (128)$$

where

$$G(\sigma_{n-2}) := \prod_{\sigma_{n-1} \in \Delta_{n-1}} (X_{\sigma_{n-1}})^{-a(\partial\sigma_{n-1};\sigma_{n-2})}. \qquad (129)$$

See Appendix B for a derivation of the quantum model as a limit of the classical model.

### 4. MBQS protocols for $M_{(d,n)}^{(\mathbb{Z}_N)}$

One Trotter step consists of measurements (1)-(4) given in Table III. These measurements have the following effects on the wave function.

(1) $\boldsymbol{\sigma}_n = \sigma_n \times \{j\}$

We measure the qudit on $\boldsymbol{\sigma}_n$ along

$$\mathcal{M}_{(\mathbb{Z}_N, A)} := \left\{ e^{i(\xi X + \text{h.c.})} |s\rangle \mid s = 0, 1, \ldots, N-1 \bmod N \right\}. \qquad (130)$$

Using the identity (A3), we get

$$\prod_{\sigma_n} \frac{1}{\sqrt{N}} Z(\partial\sigma_n)^s e^{-i(\xi Z(\partial\sigma_n) + \text{h.c.})} . \qquad (131)$$

(2) $\boldsymbol{\sigma}_{n-1} = \sigma_{n-1} \times \{j\}$

We measure the qudit on $\boldsymbol{\sigma}_{n-1}$ along

$$\mathcal{M}_{(\mathbb{Z}_N, X)} := \{|\tilde{s}\rangle \mid s = 0, ..., N-1 \bmod N\}. \qquad (132)$$

It affects the qubit at $\boldsymbol{\sigma}_n = \sigma_{n-1} \times [j, j+1]$. We apply the identity (A1). We use $a(\boldsymbol{\partial\sigma}_n; \boldsymbol{\sigma}_{n-1}) = (-1)^n$ to get

$$\prod_{\sigma_{n-1}} \frac{1}{\sqrt{N}} \left( F^{(-1)^{n+1}} Z^s \right)_{\sigma_{n-1}} . \qquad (133)$$

| basis | $\mathcal{M}_{(\mathbb{Z}_N,A)}$ $\rightarrow$ | $\mathcal{M}_{(\mathbb{Z}_N,X)}$ $\rightarrow$ | $\mathcal{M}_{(\mathbb{Z}_N,A)}$ $\rightarrow$ | $\mathcal{M}_{(\mathbb{Z}_N,B)}$ |
|---|---|---|---|---|
| layer | $pt$ | $pt$ | $I$ | $I$ |
| $d$-dim cell | $\boldsymbol{\sigma_n}$ | $\boldsymbol{\sigma_{n-1}}$ | $\boldsymbol{\sigma_{n-1}}$ | $\boldsymbol{\sigma_n}$ |
| $(d-1)$-dim cell | $\sigma_n$ | $\sigma_{n-1}$ | $\sigma_{n-2}$ | $\sigma_{n-1}$ |
| # | (1) | (2) | (3) | (4) |

TABLE III. Measurement pattern for model $M_{(d,n)}^{(N)}$.

Here the state lives on the interval $[j, j+1]$.

(3) $\boldsymbol{\sigma_{n-1} = \sigma_{n-2} \times [j, j+1]}$

We measure the qudit on $\boldsymbol{\sigma}_{n-1}$ along $\mathcal{M}_{(\mathbb{Z}_N,A)}$ to implements the energy cost for Gauss law violation. We apply the identity (A3). We note that $a(\boldsymbol{\partial}(\sigma_{n-1} \times [j, j+1]); \boldsymbol{\sigma}_{n-1}) = a(\partial\sigma_{n-1}; \sigma_{n-2})$ and find

$$\prod_{\sigma_{n-2}} \left( \frac{1}{\sqrt{N}} e^{-i(\xi \prod_{\sigma_{n-1}} Z_{\sigma_{n-1}}^{a(\partial\sigma_{n-1};\sigma_{n-2})} + \text{h.c.})} \left( \prod_{\sigma_{n-1}} Z_{\sigma_{n-1}}^{a(\partial\sigma_{n-1};\sigma_{n-2})} \right)^s \right) \tag{134}$$

Here the state lives on the interval $[j, j+1]$.

(4) $\boldsymbol{\sigma_n = \sigma_{n-1} \times [j, j+1]}$

We measure the qudit on $\boldsymbol{\sigma}_n$ along

$$\mathcal{M}_{(\mathbb{Z}_N,B)} := \{ e^{i(\xi Z + \text{h.c})} |\widehat{s}\rangle \mid s = 0, \ldots, N-1 \bmod N \}. \tag{135}$$

We apply the identity (A1) with $a(\boldsymbol{\partial}\boldsymbol{\sigma}_n; \boldsymbol{\sigma}_{n-1}) = (-1)^{n-1}$ to get

$$\prod_{\sigma_{n-1}} \frac{1}{\sqrt{N}} \left( X^{s(-1)^n} e^{-i(\xi X + \text{h.c.})} F^{(-1)^n} \right)_{\sigma_{n-1}}. \tag{136}$$

Combining (1)-(4), we get

$$U_{(d,n)}^{(\mathbb{Z}_N)}(\{\xi_i\}) := \left( \prod_{\sigma_{n-1}} \left( X^{(-1)^n s_4} e^{-i(\xi_4 X + \text{h.c.})} \right)_{\sigma_{n-1}} \right) \left( \prod_{\sigma_{n-2}} e^{-i(\xi_3 \prod_{\sigma_{n-1}} X_{\sigma_{n-1}}^{(-1)^n a(\partial\sigma_{n-1};\sigma_{n-2})} + \text{h.c.})} \right)$$
$$\prod_{\sigma_{n-2}} \left( \prod_{\sigma_{n-1}} X_{\sigma_{n-1}}^{(-1)^n a(\partial\sigma_{n-1};\sigma_{n-2})} \right)^{s_3(\sigma_{n-2})} \left( \prod_{\sigma_{n-1}} Z_{\sigma_{n-1}}^{s_2(\sigma_{n-1})} \right) \prod_{\sigma_n} Z(\partial\sigma_n)^{s_1(\sigma_n)} e^{-i(\xi_1 Z(\partial\sigma_n) + \text{h.c.})}. \tag{137}$$

We used the relations $F^{\pm 1} Z F^{\mp 1} = X^{\pm 1}$ and $F^{\pm 1} X F^{\mp 1} = Z^{\mp 1}$. Writing the wave function with the Trotterized time evolution as

$$|\psi(t)\rangle = T_{(d,n)}^{(\mathbb{Z}_N)}(t)|\psi(0)\rangle , \tag{138}$$

$$T_{(d,n)}^{(\mathbb{Z}_N)}(t) := \left( \prod_{\sigma_1} e^{i(X + \text{h.c.})\delta t/2} \prod_{\sigma_{n-2}} e^{i\Lambda\omega^{Q_{\sigma_{n-2}}} (\prod_{\sigma_{n-1}} X_{\sigma_{n-1}}^{a(\partial\sigma_{n-1};\sigma_{n-2})} + \text{h.c.})\delta t} \prod_{\sigma_n} e^{i\lambda(Z(\partial\sigma_n) + \text{h.c.})\delta t/2} \right)^j \tag{139}$$

($t = j\delta t$) and denoting the byproduct operators coming from $j$-th time step as $\Sigma^{(j)}$, we obtain

$$U_{(d,n)}^{(\mathbb{Z}_N)}(\{\xi_i\}) \left( \prod_{k=1}^{j} \Sigma^{(k)} \right) |\psi(t)\rangle = \left( \prod_{k=1}^{j+1} \Sigma^{(k)} \right) |\psi(t + \delta t)\rangle \tag{140}$$

with appropriate choices of $\{\xi_i\}_{i=1,3,4}$.

## B. Euclidean path integral and Hamiltonian MBQS

Our MBQS with the quantum Hamiltonian derived from $M_{(d,n)}^{(\mathbb{Z}_N)}$ is done by measurements on $\mathrm{gCS}_{(d,n)}$, and it is suggestive that the spacetime structure of the classical action $I_{(d,n)}$ resembles the structure of the entangler in $\mathrm{gCS}_{(d,n)}$. The aim of this section is to make such a quantum-classical correspondence manifest in terms of the Euclidean path integral or the partition function of the model $M_{(d,n)}^{(\mathbb{Z}_N)}$. It is a version of the correspondence found in [23], specialized to the model $M_{(d,n)}^{(\mathbb{Z}_N)}$.

For the gauge group $\mathbb{Z}_N$, let us define

$$|\phi_{(d,n)}^{(\mathbb{Z}_N)}\rangle := \left(e^{i\kappa(X+\mathrm{h.c.})}|0\rangle\right)^{\otimes \boldsymbol{\Delta}_n}|+\rangle^{\otimes \boldsymbol{\Delta}_{n-1}}. \tag{141}$$

Its overlap with the generalized cluster state (121) is the probability amplitude for obtaining the special measurement outcomes $s = 0$ in the measurement of qubits on $n$-cells with the A-type basis (130) and those on $(n-1)$-cells with $X$-basis. Now we replace $\kappa$ to an imaginary parameter $\kappa = i\beta J$, then this quantity is proportional to the partition function $Z_{d,n}^{(\mathbb{Z}_N)}$ defined in (126). Indeed we have

$$Z_{d,n}^{(\mathbb{Z}_N)} = N^{|\boldsymbol{\Delta}_n| + \frac{1}{2}|\boldsymbol{\Delta}_{n-1}|} \langle \phi_{(d,n)}^{(\mathbb{Z}_N)}|\mathrm{gCS}_{(d,n)}^{(\mathbb{Z}_N)}\rangle\Big|_{\kappa \to i\beta J}. \tag{142}$$

The case with the gauge group $\mathbb{R}$ is discussed in Appendix C.

For the gauge group $\mathbb{Z}_2$, we developed the imaginary-time MBQS in Section III C. If we use the A-type measurement basis state and define (70)

$$|\tilde{\phi}_{(d,n)}^{(\mathbb{Z}_2)}(\alpha)\rangle := e^{-\alpha}\left(e^{\alpha X}|0,0\rangle + \sqrt{\sinh(2|\alpha|)}|-,1\rangle\right)^{\otimes \boldsymbol{\Delta}_n}$$
$$|+\rangle^{\otimes \boldsymbol{\Delta}_{n-1}} \tag{143}$$

we have

$$\langle \tilde{\phi}_{(d,n)}^{(\mathbb{Z}_2)}(\beta J)|\mathrm{gCS}_{(d,n)}^{(\mathbb{Z}_2)}\rangle \propto \sum_{\{S_{\boldsymbol{\sigma}_{n-1}}\}} e^{-\beta I[\{S_{\boldsymbol{\sigma}_{n-1}}\}]}. \tag{144}$$

The left-hand side of (144), which is real and positive as can be seen from the right-hand side, is the probability amplitude to obtain (143) as the outcome of the simultaneous non-adaptive measurement of all the qubits on $n$-cells in the A-type basis and those on $(n-1)$-cells in the $X$-basis.

We can also consider an extended operator $W(\boldsymbol{C})$, which is a generalization of the Wilson loop. The operator is supported on an $(n-1)$-cycle $\boldsymbol{C} \in \ker(\boldsymbol{\partial}_{n-1})$, where $\boldsymbol{\partial}_{n-1}$ is the boundary operator $\boldsymbol{\partial}$ acting on $n$-chains $\boldsymbol{C}_n$. The unnormalized expectation value of $W(\boldsymbol{C})$ is

$$\langle W(\boldsymbol{C})\rangle := \sum_{\{S_{\boldsymbol{\sigma}_{n-1}}\}} S(\boldsymbol{C})e^{-\beta I[\{S_{\boldsymbol{\sigma}_{n-1}}\}]}. \tag{145}$$

The relation (144) generalizes to

$$\langle W(\boldsymbol{C})\rangle \propto \langle \tilde{\phi}_{(d,n)}^{(\mathbb{Z}_2)}(\beta J)|Z(\boldsymbol{C})|\mathrm{gCS}_{(d,n)}^{(\mathbb{Z}_2)}\rangle. \tag{146}$$

This relation implies that there is a constant-depth unitary circuit $U_W$ such that $\langle W(\boldsymbol{C})\rangle \propto \langle 0,0|^{\otimes \Delta_n}\langle 0|^{\otimes \Delta_{n-1}}U_W|0,0\rangle^{\otimes \Delta_n}|0\rangle^{\otimes \Delta_{n-1}}$. One can perform the Hadamard test [25] to estimate the matrix element of $U_W$ to obtain the the expectation value of the generalized Wilson loop operators within polynomial time.

## C. Kitaev Majorana chain

Here we propose a measurement-based simulation scheme for a fermionic system, namely Kitaev's Majorana chain [28] defined by the Hamiltonian

$$H_{\mathrm{K}} = H_{\mathrm{hop}} + H_P, \tag{147}$$

where

$$H_{\mathrm{hop}} = w\sum_{j\in\mathbb{Z}}(-c_j^\dagger c_{j+1} + c_j c_{j+1} + \mathrm{h.c.}),$$
$$H_P = -\mu\sum_{j\in\mathbb{Z}}(c_j^\dagger c_j - 1/2). \tag{148}$$

More precisely, we wish to implement the Trotterized time evolution operator

$$\left(e^{-iH_{\mathrm{hop}}\delta t}e^{-iH_P\delta t}\right)^n \tag{149}$$

via measurements. Unlike [59], where a different scheme for the Kitaev chain was proposed, our scheme involves measurements of fermion parities and is motivated by a relation between the Jordan-Wigner transformation and measurements [60].

To describe the resource state used for simulation, let us consider a two-dimensional square lattice $\mathcal{C}$. As shown in Fig. 8(a), we assign some orientations to edges $\boldsymbol{e} \in \boldsymbol{\Delta}_1$ and faces $\boldsymbol{f} \in \boldsymbol{\Delta}_2$. On each vertex $\boldsymbol{v} \in \boldsymbol{\Delta}_0$ we introduce a complex fermion $c_{\boldsymbol{v}}$ such that $\{c_{\boldsymbol{v}}, c_{\boldsymbol{v}'}^\dagger\} = \delta_{\boldsymbol{v}\boldsymbol{v}'}$, $\{c_{\boldsymbol{v}}, c_{\boldsymbol{v}'}\} = 0$. It can be decomposed into a pair of Majorana fermions $\gamma_{\boldsymbol{v}}, \gamma_{\boldsymbol{v}}'$ as $c_{\boldsymbol{v}} = (\gamma_{\boldsymbol{v}} + i\gamma_{\boldsymbol{v}}')/2$, $c_{\boldsymbol{v}}^\dagger = (\gamma_{\boldsymbol{v}} - i\gamma_{\boldsymbol{v}}')/2$. We define

$$P_{\boldsymbol{v}} := -i\gamma_{\boldsymbol{v}}\gamma_{\boldsymbol{v}}' = 1 - 2c_{\boldsymbol{v}}^\dagger c_{\boldsymbol{v}} \tag{150}$$

and label its eigenvalue and eigenstate as $(-1)^p$ and $|p\rangle_{\boldsymbol{v}} = (c_{\boldsymbol{v}}^\dagger)^p|0\rangle_{\boldsymbol{v}}$ respectively, with $p \in \{0,1\}$. Note that, on a single vertex $\boldsymbol{v}$, the operators $(P_{\boldsymbol{v}}, \gamma_{\boldsymbol{v}}, \gamma_{\boldsymbol{v}}')$ obey the same algebraic relations as the Pauli operators $(Z, X, Y)$, so that $|p\rangle_{\boldsymbol{v}} = (\gamma_{\boldsymbol{v}})^p|0\rangle_{\boldsymbol{v}}$. We also define a hermitian operator squaring to 1,

$$S_{\boldsymbol{e}} = i\gamma_{\boldsymbol{v}_-}'\gamma_{\boldsymbol{v}_+}, \tag{151}$$

where the vertices $\boldsymbol{v}_\pm = \boldsymbol{v}_\pm(\boldsymbol{e})$ are defined by the relation $\boldsymbol{v}_+ - \boldsymbol{v}_- = \boldsymbol{\partial}\boldsymbol{e}$. On each edge $\boldsymbol{e} \in \boldsymbol{\Delta}_1$ we introduce a

qubit. Following [60], we define the operator $CS_{\boldsymbol{e}}$ to be $S_{\boldsymbol{e}}$ controlled by the qubit on $\boldsymbol{e}$ and set

$$\mathcal{U}_{CS} := \prod_{\boldsymbol{e} \in \boldsymbol{\Delta}_1} CS_{\boldsymbol{e}} \qquad (152)$$

with the following ordering. Within a horizontal layer the operators $CS_{\boldsymbol{e}}$ commute with each other, and we let them appear in the product simultaneously. We order such layers so that as we go down in Fig. 8(a) we go to the left within the product (152) [61].

We now define the cluster state $|\psi_{\mathcal{C}}\rangle$ [60] by

$$|\psi_{\mathcal{C}}\rangle := \mathcal{U}_{CS} \left( |0\rangle^{\otimes \boldsymbol{\Delta}_0} \otimes |+\rangle^{\otimes \boldsymbol{\Delta}_1} \right). \qquad (153)$$

Its stabilizers are [62]

$$\mathcal{U}_{CS} P_{\boldsymbol{v}} \mathcal{U}_{CS}^{-1} = P_{\boldsymbol{v}} Z(\boldsymbol{\partial}^* \boldsymbol{v}) \qquad (154)$$

for an arbitrary vertex $\boldsymbol{v}$,

$$\mathcal{U}_{CS} X_{\boldsymbol{e}} \mathcal{U}_{CS}^{-1} = X_{\boldsymbol{e}} S_{\boldsymbol{e}} Z_{\boldsymbol{e}'} = \begin{array}{c} i\gamma'_{\boldsymbol{v}_-} \\ | \\ X_{\boldsymbol{e}} \\ | \\ Z_{\boldsymbol{e}'} \!\!-\!\! \gamma_{\boldsymbol{v}_+} \end{array} \qquad (155)$$

for a vertical edge $\boldsymbol{e}$, and

$$\mathcal{U}_{CS} X_{\boldsymbol{e}} \mathcal{U}_{CS}^{-1} = X_{\boldsymbol{e}} S_{\boldsymbol{e}} Z_{\boldsymbol{e}'} = \begin{array}{c} \gamma'_{\boldsymbol{v}_-} \!\!-\!\! X_{\boldsymbol{e}} \!\!-\!\! i\gamma_{\boldsymbol{v}_+} \\ | \\ Z_{\boldsymbol{e}'} \end{array} \qquad (156)$$

for a horizontal edge $\boldsymbol{e}$.

Using the operators $S$ and $P$ on a one-dimensional lattice we can rewrite the Hamiltonian $H_{\mathrm{K}} = H_{\mathrm{hop}} + H_P$ as

$$H_{\mathrm{hop}} = w \sum_{e \in \Delta_1} S_e \,, \qquad H_P = \frac{\mu}{2} \sum_{v \in \Delta_0} P_v \,. \qquad (157)$$

where the vertices $v \in \Delta_0$ and the edges $e \in \Delta_1$ are those of a one-dimensional lattice. Written in this way, it is manifest that different terms commute with each other within each of $H_{\mathrm{hop}}$ and $H_P$, so that the evolution operator can be written as

$$\left( \prod_{e \in \Delta_1} e^{-iw\delta t S_e} \prod_{v \in \Delta_0} e^{-i\frac{\mu}{2}\delta t P_v} \right)^n \qquad (158)$$

Let us now consider a reduced two-dimensional lattice $\mathcal{C}_{\mathrm{red}}$ which is periodic in the 2-direction and has a boundary, which is to be identified with the one-dimensional lattice for the Majorana chain. See Fig. 8(b) and (c). We assign a non-negative integer $j \geq 0$ for each layer, so that as cells of the two-dimensional lattice, the vertices are $\boldsymbol{v} = v \times \{j\}$, horizontal edges are $\boldsymbol{e} = e \times \{j\}$, and

vertical edges are $\boldsymbol{e} = v \times [j, j+1]$. In the middle of simulation the state of the total system takes the form

$$|\psi_{2d}\rangle = \mathcal{U}_{CS} \left( \mathcal{O}_{\mathrm{bp}} |\psi_{1d}\rangle \otimes |0\rangle_{\mathrm{bulk}} \otimes |+\rangle_{\mathrm{bulk}} \right), \qquad (159)$$

where $\mathcal{O}_{\mathrm{bp}}$ is a product of $\gamma_v$ and $P_v$. Let us consider a horizontal edge $\boldsymbol{e} = e \times \{j\}$ and its boundary vertices $\boldsymbol{v}_{\pm}$. See Fig. 8(c). The relation

$$e^{-i\xi X_{\boldsymbol{e}}} CS_{\boldsymbol{e}} = CS_{\boldsymbol{e}} e^{-i\xi X_{\boldsymbol{e}} S_{\boldsymbol{e}}} \qquad (160)$$

implies that

$$_{\boldsymbol{e}}\langle s| e^{-i\xi X_{\boldsymbol{e}}} CS_{\boldsymbol{e}} |+\rangle_{\boldsymbol{e}} = \frac{1}{\sqrt{2}} (S_{\boldsymbol{e}})^s e^{-i\xi S_{\boldsymbol{e}}}, \quad s = \pm 1, \quad (161)$$

or equivalently

$$\mathbb{P}_{\boldsymbol{e}}^{(A)}(s, \xi) CS_{\boldsymbol{e}} |+\rangle_{\boldsymbol{e}} |\psi\rangle_{\boldsymbol{v}_- \boldsymbol{v}_+}$$
$$= \frac{1}{\sqrt{2}} e^{i\xi X_{\boldsymbol{e}}} |s\rangle_{\boldsymbol{e}} (S_{\boldsymbol{e}})^s e^{-i\xi S_{\boldsymbol{e}}} |\psi\rangle_{\boldsymbol{v}_- \boldsymbol{v}_+}, \qquad (162)$$

where

$$\mathbb{P}_{\boldsymbol{e}}^{(A)}(s, \xi) := \frac{1 + (-1)^s e^{i\xi X_{\boldsymbol{e}}} Z_{\boldsymbol{e}} e^{-i\xi X_{\boldsymbol{e}}}}{2} \,. \qquad (163)$$

Thus we can implement the hopping $(S_e)$ term in the Hamiltonian, up to a byproduct operator $(S_e)^s$, by measuring the edge qubit with the basis

$$\mathcal{M}_A = \left\{ e^{i\xi X} |s\rangle \,\Big|\, s = 0, 1 \right\}. \qquad (164)$$

See Fig. 9(a).

Next let us consider an edge that extends within the bulk in the 2-direction. See Fig. 8(d). We find

$$\mathbb{P}_{\boldsymbol{v}_-}^{(P)}(t) \mathbb{P}_{\boldsymbol{e}}^{(B)}(s, \xi) CS_{\boldsymbol{e}} (\gamma_{\boldsymbol{v}_-})^p |0+0\rangle_{\boldsymbol{v}_- \boldsymbol{e} \boldsymbol{v}_+}$$
$$= \frac{1}{2} (P_{\boldsymbol{v}_+})^{s+1} e^{-i\xi P_{\boldsymbol{v}_+}} e^{i\xi Z_{\boldsymbol{e}}} (\gamma_{\boldsymbol{v}_-})^t (\gamma_{\boldsymbol{v}_+})^{t+p} |0s0\rangle_{\boldsymbol{v}_- \boldsymbol{e} \boldsymbol{v}_+}, \qquad (165)$$

where

$$\mathbb{P}_{\boldsymbol{v}_-}^{(P)} := \frac{1 + (-1)^t P_{\boldsymbol{v}_-}}{2} \qquad (166)$$

and

$$\mathbb{P}_{\boldsymbol{e}}^{(B)}(s, \xi) := \frac{1 + (-1)^s e^{i\xi Z_{\boldsymbol{e}}} X_{\boldsymbol{e}} e^{-i\xi Z_{\boldsymbol{e}}}}{2} \,. \qquad (167)$$

In (165), one may move $(\gamma_{\boldsymbol{v}_+})^t$ to the left, changing $e^{-i\xi P}$ to $e^{-i(-1)^t \xi P}$. This implies that we can teleport [63] $|\psi\rangle = \sum_p \psi_p |p\rangle$ at $\boldsymbol{v}_-$ to $e^{-i(-1)^t \xi P} |\psi\rangle$ at $\boldsymbol{v}_+$ by measuring the fermion in the $P$-basis

$$\mathcal{M}_P = \{ |0\rangle, |1\rangle \} \qquad (168)$$

at $\boldsymbol{v}_-$ and by measuring the qubit on $\boldsymbol{e}$ in the eigenbasis

$$\mathcal{M}_B = \{ e^{i\xi Z} |\pm\rangle \} \qquad (169)$$

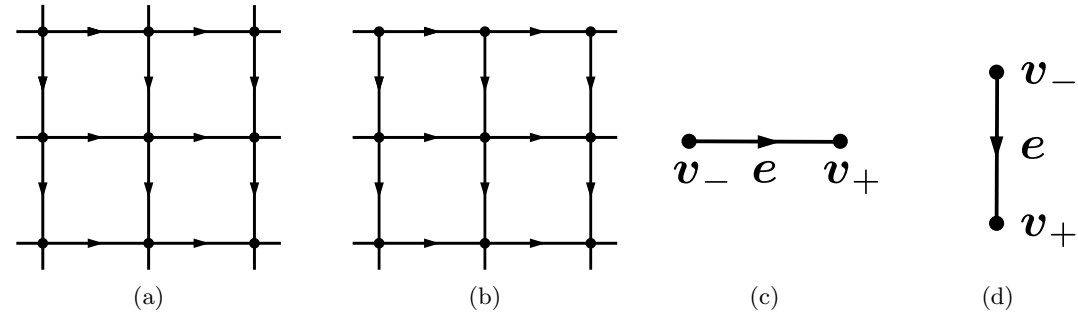

FIG. 8. (a) We place fermions on vertices, and qubits on edges. (b) We introduce a boundary to be identified with a one-dimensional lattice for the Kitaev chain. (c) An edge and its ends within a one-dimensional chain. (d) A vertical edge and its ends.

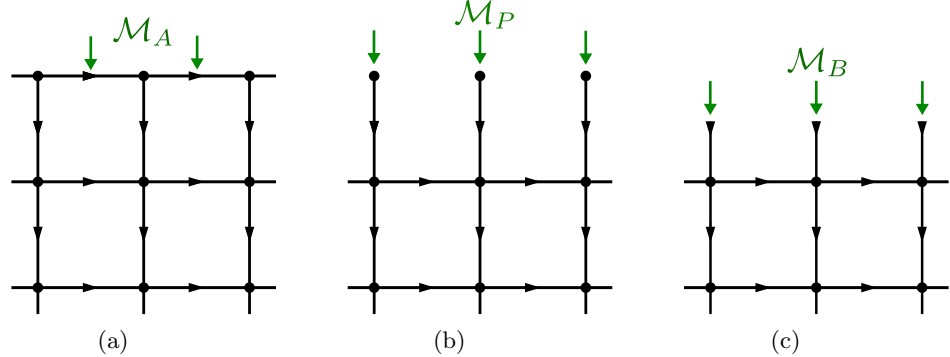

FIG. 9. The measurement bases $\mathcal{M}_{\text{hop}}$, $\mathcal{M}_P$ $\mathcal{M}_{XY}$ for the indicated qubits (arrows on edges) and fermions (dots).

for $e^{i\xi Z} X e^{-i\xi Z}$, up to byproduct operators. See Fig. 9(b) and (c). By applying (162) and (165) repeatedly, we see that the following measurement pattern, with measurement angles $\xi$ chosen adaptively, realizes the time evolution (158) for the Kitaev chain:

| basis $\rightarrow$ | $\mathcal{M}_A \rightarrow$ | $\mathcal{M}_P \rightarrow$ | $\mathcal{M}_B \rightarrow$ |
|---|---|---|---|
| layer | $pt$ | $pt$ | $I$ |
| 2d cell | $\boldsymbol{\sigma}_1$ | $\boldsymbol{\sigma}_0$ | $\boldsymbol{\sigma}_1$ |
| 1d cell | $\sigma_1$ | $\sigma_0$ | $\sigma_0$ |

(170)

Let us discuss the computation of physical observables. Natural local observables of the Kitaev chain are the individual terms in the Hamiltonian. The observable $P_v$ in $H_P$ can be computed simply by measuring it at $v$. The computation of the observable $S_e$ is more interesting. At the end of simulation the state is of the form

$$|\Psi\rangle = \prod_{e\in\Delta_1} CS_e \, \mathcal{O}_{\text{bp}}|\psi\rangle_{\Delta_0}|+\rangle_{\Delta_1},\qquad (171)$$

where $\mathcal{O}_{\text{bp}}$ is the byproduct operator. If we measure $X$

on edge $e$ with outcome $s_e \in \{0,1\}$, the resulting state is

$$\frac{1+(-1)^{s_e}X_e}{2}|\Psi\rangle$$
$$= \left(\prod_{e'\in\Delta_1} CS_{e'}\right)\frac{1+(-1)^{s_e}X_e S_e}{2}\mathcal{O}_{\text{bp}}|\psi\rangle_{\Delta_0}\otimes|+\rangle_{\Delta_1}$$
$$= \left(\prod_{e'\in\Delta_1} CS_{e'}\right)\frac{1+(-1)^{s_e}S_e}{2}\mathcal{O}_{\text{bp}}|\psi\rangle_{\Delta_0}\otimes|+\rangle_{\Delta_1}.$$

(172)

This means that the outcome of measuring $X_e$ on $|\Psi\rangle$ is the same as the outcome of measuring $S_e$ on $\mathcal{O}_{\text{bp}}|\psi\rangle_{\Delta_0}$. The knowledge of the measurement outcomes throughout the simulation allows us to know whether the operator $S_e$ commutes or anti-commute with $\mathcal{O}_{\text{bp}}$. Thus we can compute $S_e$ for the state $|\psi\rangle$ by measuring $X_e$ on edge $e$.

## VI. SPT ORDER OF THE GENERALIZED CLUSTER STATE

A nontrivial phase, in the sense of the topological order, is defined as the depth of the local quantum circuit required to prepare the state from a product state being large. A wave function in an SPT phase can be prepared

with a finite depth local quantum circuit. The SPT phase is nontrivial if the state requires a large-depth local quantum circuit when we demand the circuit to commute with the symmetry of interest.

The gauging procedure [38, 64–67] is a powerful tool to diagnose an SPT order. Upon gauging, trivial and non-trivial SPT Hamiltonians belong to two distinct topological phases. This method has been applied to several models to demonstrate that they have nontrivial SPT orders. In the original argument [64], the Levin-Gu SPT state, which is an SPT with a 0-form symmetry, was minimally coupled to a 1-form gauge field, and the gauged model was shown to possess excitations with double-semionic braiding statisics, which differs from the braiding statistics found in the gauged version of trivial SPT order. The method can be employed for detecting SPT orders protected by higher-form symmetries as well [38], and the argument suggested that the RBH cluster state has a non-trivial SPT order protected by a $\mathbb{Z}_2 \times \mathbb{Z}_2$ 1-form symmetry [66]. Our generalized cluster state $\mathrm{gCS}_{(d,n)}$ is a natural extension of these states, and it is plausible that it has an SPT order with higher-form symmetries. In this section, we discuss gauging the Hamiltonian that defines $\mathrm{gCS}_{(d,n)}$.

First we will see that the generalized cluster state $\mathrm{gCS}_{(d,n)}$ possesses $(d-n)$- and $(n-1)$-form $\mathbb{Z}_2$ symmetries. Then we define the gauging map and discuss its properties. The definition we employ is the one discussed in [38], for example. (We can regard the map as a result of minimally coupling gauge degrees of freedom. See Appendix E.) Then we apply the method to our generalized cluster state to argue that it possesses a nontrivial SPT order protected by $(d-n)$- and $(n-1)$-form $\mathbb{Z}_2$ symmetries.

### A. Symmetries of the generalized cluster states

The generalized cluster state $\mathrm{gCS}_{(d,n)}$ possesses higher form symmetries. They are $(d-n)$- and $(n-1)$-form $\mathbb{Z}_2$ symmetries generated by the following operators, $S(\boldsymbol{\mathcal{M}})$ and $S(\boldsymbol{\mathcal{M}}')$, respectively:

$$S(\boldsymbol{\mathcal{M}}) = \prod_{\boldsymbol{\sigma}_n \in \boldsymbol{\mathcal{M}}} X(\boldsymbol{\sigma}_n), \quad S(\boldsymbol{\mathcal{M}}') = \prod_{\boldsymbol{\sigma}_{n-1} \in \boldsymbol{\mathcal{M}}'} X(\boldsymbol{\sigma}_{n-1})$$
(173)

with

$$\boldsymbol{\mathcal{M}} \in \ker(\boldsymbol{\partial}_n) = \{\boldsymbol{z_n} \,|\, \boldsymbol{\partial z_n} = 0\} \,, \quad (174)$$
$$\boldsymbol{\mathcal{M}}' \in \ker(\boldsymbol{\partial}^*_{d-n+1}) = \{\boldsymbol{z_{n-1}} \,|\, \boldsymbol{\partial^* z_{n-1}} = 0\} \,. \quad (175)$$

We refer to $\boldsymbol{\mathcal{M}}$ and $\boldsymbol{\mathcal{M}}'$ respectively as $n$- and $(d-n+1)$-brane operators (generalizing "membrane" operators). The existence of such symmetries can be shown by taking a product of corresponding stabilizers over the closed manifold:

$$\prod_{\boldsymbol{\sigma}_n \in \boldsymbol{\mathcal{M}}} K_{(d,n)}(\boldsymbol{\sigma}_n) = \prod_{\boldsymbol{\sigma}_n \in \boldsymbol{\mathcal{M}}} X(\boldsymbol{\sigma}_n) Z(\boldsymbol{\partial \sigma}_n)$$
$$= \prod_{\boldsymbol{\sigma}_n \in \boldsymbol{\mathcal{M}}} X(\boldsymbol{\sigma}_n) \,. \quad (176)$$

A similar calculation can be done for the other type of stabilizers as well. Special cases of symmetry generators in (173) are $X(\boldsymbol{\partial \sigma}_{n-1})$ and $X(\boldsymbol{\partial^* \sigma}_{n-2})$. See (96).

### B. Gauging map

We now introduce the gauging map. Let $\mathcal{H}_{n-1}$ ($\mathcal{H}_n$) be the Hilbert space for all the qubits on $(n-1)$-cells ($n$-cells). Let us define the symmetric subspaces

$$\mathcal{H}_0^{sym} = \{|\psi\rangle \in \mathcal{H}_n \otimes \mathcal{H}_{n-1} \,| \\ S(\boldsymbol{\mathcal{M}})S(\boldsymbol{\mathcal{M}}')|\psi\rangle = |\psi\rangle \text{ for } \forall \boldsymbol{\mathcal{M}}, \boldsymbol{\mathcal{M}}'\} \quad (177)$$

and

$$\mathcal{H}_1^{sym} = \{|\psi\rangle \in \mathcal{H}_{n-1} \otimes \mathcal{H}_n \,| \\ T(\boldsymbol{\mathcal{N}})T(\boldsymbol{\mathcal{N}}')|\psi\rangle = |\psi\rangle \text{ for } \forall \boldsymbol{\mathcal{N}}, \boldsymbol{\mathcal{N}}'\} \quad (178)$$

with

$$T(\boldsymbol{\mathcal{N}}) = \prod_{\boldsymbol{\sigma}_n \in \boldsymbol{\mathcal{N}}} Z(\boldsymbol{\sigma}_n), \quad T(\boldsymbol{\mathcal{N}}') = \prod_{\boldsymbol{\sigma}_{n-1} \in \boldsymbol{\mathcal{N}}'} Z(\boldsymbol{\sigma}_{n-1}) \,, \quad (179)$$

where $\boldsymbol{\mathcal{N}} = \{\boldsymbol{z_n} \,|\, \boldsymbol{\partial z_n} = 0\}$ ($\boldsymbol{\mathcal{N}}' = \{\boldsymbol{z_{n-1}} \,|\, \boldsymbol{\partial^* z_{n-1}} = 0\}$) is a set of cycles. One can show that

$$\dim \mathcal{H}_0^{sym} = \dim \mathcal{H}_1^{sym} \,. \quad (180)$$

For an arbitrary chain $\boldsymbol{c_n} = \sum_{\boldsymbol{\sigma}_n} a(\boldsymbol{\sigma}_n)\boldsymbol{\sigma}_n$, we write $|\boldsymbol{c_n}\rangle = |\{a(\boldsymbol{\sigma}_n)\}\rangle$. Let us consider the linear map $\Gamma : \mathcal{H}_n \otimes \mathcal{H}_{n-1} \to \mathcal{H}_{n-1} \otimes \mathcal{H}_n$ defined by

$$\Gamma\left(|\boldsymbol{c_n}\rangle \otimes |\boldsymbol{c_{n-1}}\rangle\right) = |\boldsymbol{\partial c_n}\rangle \otimes |\boldsymbol{\partial^* c_{n-1}}\rangle \,. \quad (181)$$

Its restriction to $\mathcal{H}_0^{sym}$ induces the gauging map $\Gamma : \mathcal{H}_0^{sym} \to \mathcal{H}_1^{sym}$ (up to a normalization factor discussed in Appendix E). The gauging map $\Gamma$ defines the transformation $A^\Gamma$ of a symmetry respecting operator $A$ via

$$\Gamma\left(A|\psi\rangle\right) = A^\Gamma\left(\Gamma|\psi\rangle\right) \quad \forall|\psi\rangle \in \mathcal{H}_0^{sym} \,. \quad (182)$$

Now let us discuss some properties of $\Gamma$. $\Gamma$ brings the generators of the symmetries to identity: $S(\boldsymbol{\mathcal{M}}) \xrightarrow{\Gamma} I$, $S(\boldsymbol{\mathcal{M}}') \xrightarrow{\Gamma} I$. Namely,

$$\Gamma\left(S(\boldsymbol{\mathcal{M}})\,S(\boldsymbol{\mathcal{M}}')|\boldsymbol{c_n}\rangle \otimes |\boldsymbol{c_{n-1}}\rangle\right) \\ = I \cdot \Gamma(|\boldsymbol{c_n}\rangle \otimes |\boldsymbol{c_{n-1}}\rangle) \,.$$

The output states, $\Gamma(|\psi_1\rangle)$ and $\Gamma(|\psi_2\rangle)$, are identical if and only if there exists $S(\boldsymbol{\mathcal{M}})$ such that $S(\boldsymbol{\mathcal{M}})|\psi_1\rangle =$

$|\psi_2\rangle$. $\Gamma$ is locality preserving, gap preserving, bijective and isometric.

Below we will make use of the following argument [64]. Let us consider gauging two Hamiltonians

$$H_1 \xrightarrow{\Gamma} H_1^\Gamma \ , \tag{183}$$

$$H_2 \xrightarrow{\Gamma} H_2^\Gamma \ . \tag{184}$$

If the two topological orders that $H_1^\Gamma$ and $H_2^\Gamma$ possess differ, then the SPT orders that $H_1$ and $H_2$ have cannot be the same. Indeed, if there were a path [68] that connects $H_1$ and $H_2$ without breaking the symmetry or closing the energy gap, then there would also be a path that connects $H_1^\Gamma$ and $H_2^\Gamma$, which is a contradiction.

### C.  Mapping the generalized cluster states

Now we consider the trivial Hamiltonian defined on $n$-cells and $(n-1)$-cells in $d$-dimensions,

$$H_{\text{trivial}} := - \sum_{\boldsymbol{\sigma}_n \in \boldsymbol{\Delta}_n} X(\boldsymbol{\sigma}_n) - \sum_{\boldsymbol{\sigma}_{n-1} \in \boldsymbol{\Delta}_{n-1}} X(\boldsymbol{\sigma}_{n-1}) \ , \tag{185}$$

and the Hamiltonian for the gCS$_{(d,n)}$,

$$H_{\text{gCS}} := - \sum_{\boldsymbol{\sigma}_n \in \boldsymbol{\Delta}_n} K(\boldsymbol{\sigma}_n) - \sum_{\boldsymbol{\sigma}_{n-1} \in \boldsymbol{\Delta}_{n-1}} K(\boldsymbol{\sigma}_{n-1}) \ . \tag{186}$$

Let us first consider gauging the trivial Hamiltonian. The operator $X(\boldsymbol{\sigma}_n)$ is mapped to $X(\boldsymbol{\partial\sigma}_n)$, and $X(\boldsymbol{\sigma}_{n-1})$ to $X(\boldsymbol{\partial}^*\boldsymbol{\sigma}_{n-1})$. Following [64] we add $Z(\boldsymbol{\partial\sigma}_{n+1})$ (if $n+1 \leq d$) and $Z(\boldsymbol{\partial}^*\boldsymbol{\sigma}_{n-2})$ (if $n-2 \geq 0$) to the Hamiltonian so that the fluxes vanish (the gauge fields are flat) in the ground states. Therefore we obtain

$$
\begin{aligned}
&H_{\text{trivial}}^{\text{gauged}} \\
&= - \underbrace{\sum_{\boldsymbol{\sigma}_n \in \boldsymbol{\Delta}_n} X(\boldsymbol{\partial\sigma}_n) - \sum_{\boldsymbol{\sigma}_{n-2} \in \boldsymbol{\Delta}_{n-2}} Z(\boldsymbol{\partial}^*\boldsymbol{\sigma}_{n-2})}_{=:H_{n-1}} \\
&\quad - \underbrace{\sum_{\boldsymbol{\sigma}_{n-1} \in \boldsymbol{\Delta}_{n-1}} X(\boldsymbol{\partial}^*\boldsymbol{\sigma}_{n-1}) - \sum_{\boldsymbol{\sigma}_{n+1} \in \boldsymbol{\Delta}_{n+1}} Z(\boldsymbol{\partial\sigma}_{n+1})}_{=:H_n} \ .
\end{aligned}
\tag{187}
$$

Here $H_n$ and $H_{n-1}$ define two decoupled theories on $n$-cells and $(n-1)$-cells, respectively. For the generalized toric code $H_n$, there are $N_{\text{site}} \cdot {}_dC_n$ physical qubits living on $n$-cells and $k := {}_dC_n$ logical qubits, where ${}_dC_n = d!/n!(d-n)!$. There are $k$ pairs of anti-commuting logical operators, where a logical Pauli $Z$ operator acts on $n$-dimensional hyperplanes and a logical Pauli $X$ operator acts on $(d-n)$-dimensional hyperplanes which are non-contractable on a torus. See [69] for details.

On the other hand, gauging the gCS Hamiltonian gives

$$H_{\text{gCS}}^{\text{gauged}} = - \sum_{\boldsymbol{\sigma}_n \in \boldsymbol{\Delta}_n} K^\Gamma(\boldsymbol{\sigma}_n) - \sum_{\boldsymbol{\sigma}_{n-1} \in \boldsymbol{\Delta}_{n-1}} K^\Gamma(\boldsymbol{\sigma}_{n-1}) \tag{188}$$

where

$$K^\Gamma(\boldsymbol{\sigma}_n) = Z(\boldsymbol{\sigma}_n)X(\boldsymbol{\partial\sigma}_n) \ , \tag{189}$$

$$K^\Gamma(\boldsymbol{\sigma}_{n-1}) = Z(\boldsymbol{\sigma}_{n-1})X(\boldsymbol{\partial}^*\boldsymbol{\sigma}_{n-1}) \ . \tag{190}$$

We do not need to add extra terms to make the fluxes vanish because $K^\Gamma(\boldsymbol{\partial\sigma}_{n+1}) = Z(\boldsymbol{\partial\sigma}_{n+1})$, $K^\Gamma(\boldsymbol{\partial}^*\boldsymbol{\sigma}_{n-2}) = Z(\boldsymbol{\partial}^*\boldsymbol{\sigma}_{n-2})$. The Hamiltonian (188) is the Hadamard transform of the original Hamiltonian $H_{\text{gCS}}$, so the ground state described by the mapped Hamiltonian $H_{\text{gCS}}^\Gamma$ is still short-range entangled, meaning it does not have a topological order. Since the gauged version of the trivial Hamiltonian possesses a nontrivial topological order, the ungauged Hamiltonian $H_{\text{gCS}}$ is in a nontrivial SPT phase.

### D.  Brane operators and projective representation

Another approach to probing a nontrivial SPT order is to find a projective representation [70] by introducing boundaries [33, 71]. Recall that we have $K(\boldsymbol{\sigma}_n)$ and $K(\boldsymbol{\sigma}_{n-1})$ associated with $\boldsymbol{\sigma}_n \in \boldsymbol{\Delta}_n$ and $\boldsymbol{\sigma}_{n-1} \in \boldsymbol{\Delta}_{n-1}$, respectively. In the bulk, we have a symmetry generators, each of which is a product of stabilizers over a closed manifold. Namely, the $(n-1)$-form symmetry is generated by $(d-n+1)$-brane operators, and $(d-n)$-form symmetry is generated by $n$-brane operators. In this section, we consider a lattice with boundary and we wish to show that the action of the symmetry generator at the boundary forms a projective representation, which indicates that the bulk theory is an SPT protected by the symmetry.

Let us consider a $d$-dimensional lattice, which is periodic in the $(x^2, x^3, \ldots, x^d)$-directions with length $L_i$ ($i = 2, \ldots, d$). Consider an $n$-brane $\boldsymbol{B}$ extended in the $(x^1, x^2, \ldots, x^n)$-directions and a $(d-n+1)$-brane $\tilde{\boldsymbol{B}}$ extended in the $(x^1, x^{n+1}, \ldots, x^d)$-directions. The former corresponds to a union of $n$-cells within a hyperplane extended in the $(x^1, x^2, \ldots, x^n)$-directions. The latter corresponds to a union of $(n-1)$-cells extended in $(x^2, \cdots, x^n)$-directions intersecting with a hyperplane extended in the $(x^1, x^{n+1}, \cdots, x^d)$-directions. For definiteness, we consider the case where the boundaries at $x_1 = 0, L_1$ consist of $(n-1)$-cells. We denote the boundary at $x_1 = 0$ as $\boldsymbol{L}$ and that at $x_1 = L_1$ as $\boldsymbol{R}$. The same argument below holds as well when we take $n$-cells as boundaries.

Now the brane operators are defined as

$$M(\boldsymbol{B}) = \prod_{\boldsymbol{\sigma}_{n-1} \in \boldsymbol{B}} K(\boldsymbol{\sigma}_{n-1}) \tag{191}$$

$$= \prod_{\boldsymbol{\sigma}_{n-1} \in \boldsymbol{B}} X(\boldsymbol{\sigma}_{n-1}) \prod_{\boldsymbol{\sigma}_n \in \partial \boldsymbol{B} \cap (\boldsymbol{L} \cup \boldsymbol{R})} Z(\boldsymbol{\sigma}_n) \tag{192}$$

$$\tilde{M}(\tilde{\boldsymbol{B}}) = \prod_{\boldsymbol{\sigma}_n \in \tilde{\boldsymbol{B}}} K(\boldsymbol{\sigma}_n) \tag{193}$$

$$= \prod_{\boldsymbol{\sigma}_n \in \tilde{\boldsymbol{B}}} X(\boldsymbol{\sigma}_n) . \tag{194}$$

We consider the restriction of these operators to the boundary [72]:

$$M^i(\boldsymbol{B}) = \prod_{\boldsymbol{\sigma}_n \in \partial \boldsymbol{B} \cap i} Z(\boldsymbol{\sigma}_n) \quad (i = \boldsymbol{L}, \boldsymbol{R}) , \tag{195}$$

$$\tilde{M}^i(\tilde{\boldsymbol{B}}) = \prod_{\boldsymbol{\sigma}_n \in \tilde{\boldsymbol{B}} \cap i} X(\boldsymbol{\sigma}_n) \quad (i = \boldsymbol{L}, \boldsymbol{R}) . \tag{196}$$

Now, since for an arbitrary pair of chains $\boldsymbol{c}, \boldsymbol{c}'$ we have $Z(\boldsymbol{c})X(\boldsymbol{c}') = (-1)^{|\boldsymbol{c} \cap \boldsymbol{c}'|}X(\boldsymbol{c}')Z(\boldsymbol{c})$, and $|(\partial \boldsymbol{B} \cap i) \cap (\tilde{\boldsymbol{B}} \cap i)| = 1$ for $i = \boldsymbol{L}, \boldsymbol{R}$, we obtain

$$\{M^i(\boldsymbol{B}), \tilde{M}^i(\tilde{\boldsymbol{B}})\} = 0 \quad (i = \boldsymbol{L}, \boldsymbol{R}) . \tag{197}$$

This implies that the brane operators restricted to a boundary furnish a projective representation of $\mathbb{Z}_2 \times \mathbb{Z}_2$, with each factor generated by one type of brane operator. This observation supports our claim that the gCS possesses a nontrivial SPT order protected by $\mathbb{Z}_2$ $(d-n)$- and $(n-1)$-form symmetries. This generalizes a result in [66] for gCS$_{(3,2)}$.

For the Majorana state (153) with fermionic symmetry, it was argued [73] in [60] that one can find a nontrivial commutation relation between the fermionic $\mathbb{Z}_2^F$ 0-form symmetry and bosonic $\mathbb{Z}_2$ 1-form symmetry, the latter of which has fermionic operators at its endpoints. The SPT class for such pair of symmetries is suggested to be nontrivial, although there is only one SPT class.

# VII. SYMMETRIES IN MEASUREMENT-BASED QUANTUM SIMULATION AND A BULK-BOUNDARY CORRESPONDENCE

In this section we study the interplay between the symmetries of the lattice models to be simulated and those of the bulk cluster states that simulate the lattice models. We propose that MBQS is a type of bulk-boundary, or holographic, correspondence between the simulated theory $M_{(d,n)}$ and the system given by the generalized cluster state $|\text{gCS}_{(d,n)}\rangle$, generalizing [74]. We discuss the case of $(d,n) = (3,1)$ and $(3,2)$ explicitly. The reinterpretation of the stabilizers [42] as gauge symmetries discussed in Appendix D also supports our proposal.

## A. Ising model $M_{(3,1)}$

The Hamiltonian (13) of the model $M_{(3,1)}$ is invariant under the simultaneous sign flip of $Z(\sigma_0)$ for all vertices $\sigma_0 \in \Delta_0$. This is an an ordinary (0-form) global $\mathbb{Z}_2$ symmetry generated by $\prod_{\sigma_0} X(\sigma_0)$.

In the middle of the simulation, the qubits that remain unmeasured reside on a three-dimensional lattice, which is periodic in two (1- and 2-) directions and has a boundary [75]. The simulated state $|\psi_{2d}\rangle$ of the Ising model is defined on the qubits at the vertices of the two-dimensional square lattice that is identified with the boundary of the three-dimensional lattice. The total state $|\psi_{3d}\rangle$ takes the form

$$|\psi_{3d}\rangle = \mathcal{U}_{CZ}\big(\mathcal{O}_{\text{bp}}|\psi_{2d}\rangle \otimes |+\rangle_{\text{bulk}}\big) . \tag{198}$$

Here $\mathcal{O}_{\text{bp}}$ is the byproduct operator, *i.e.*, a product of Pauli operators acting on $|\psi_{2d}\rangle$, which arises from the preceding measurements. The state $|+\rangle_{\text{bulk}}$ is the tensor product of the $X$-eigenstates with eigenvalue $+1$ ($|+\rangle$) over all the bulk qubits and the vertex qubits on the boundary. The entangler $\mathcal{U}_{CZ}$ is the product of all the controlled-$Z$ gates between neighboring pairs of a vertex and an edge.

Let $g$ be the generator of the zero-form global symmetry $\mathbb{Z}_2$, and $U_g = \prod_{\sigma_0} X(\sigma_0)$ its representation on the Hilbert space of the Ising model. We wish to understand the effect of $U_g$ on $|\psi_{3d}\rangle$ in the coupled boundary-bulk system. We have

$$\mathcal{U}_{CZ}\big(\mathcal{O}_{\text{bp}}U_g|\psi_{2d}\rangle \otimes |+\rangle_{\text{bulk}}\big)$$
$$= \mathcal{U}_{CZ}\left(\big(\mathcal{O}_{\text{bp}}U_g\mathcal{O}_{\text{bp}}^{-1}\big)\mathcal{O}_{\text{bp}}|\psi_{2d}\rangle \otimes |+\rangle_{\text{bulk}}\right) \tag{199}$$
$$= \mathcal{U}_{CZ}\big(\mathcal{O}_{\text{bp}}U_g\mathcal{O}_{\text{bp}}^{-1}\big)\mathcal{U}_{CZ}^{-1}|\psi_{3d}\rangle$$

We note that $\mathcal{O}_{\text{bp}}U_g\mathcal{O}_{\text{bp}}^{-1}$ equals $(-1)^m U_g$, where $(-1)^m$ is a sign determined by the outcomes of the preceding measurements. Therefore, the operator $\mathcal{U}_{CZ}\big(\mathcal{O}_{\text{bp}}U_g\mathcal{O}_{\text{bp}}^{-1}\big)\mathcal{U}_{CZ}^{-1}$ equals $(-1)^m \prod_{\sigma_0} K(\sigma_0)$, where $K(\sigma_0) = \mathcal{U}_{CZ}(X(\sigma_0) \otimes I_{\text{bulk}})\mathcal{U}_{CZ}^{-1}$ for a boundary vertex $\sigma_0$.

The state $|\psi_{3\text{d}}\rangle$ in (198) is invariant under $K(\sigma_0)$ when $\sigma_0 \in \boldsymbol{\Delta}_0 \backslash \Delta_0$ is a bulk vertex. This motivates us to define

$$\boldsymbol{U}_g(\Lambda) := \mathcal{U}_{CZ}(\mathcal{O}_{\text{bp}}U_g\mathcal{O}_{\text{bp}}^{-1})\mathcal{U}_{CZ}^{-1} \prod_{\boldsymbol{\sigma}_0 \in \boldsymbol{\Delta}_0 \backslash \Delta_0} K(\boldsymbol{\sigma}_0)^{\Lambda(\sigma_0)} \tag{200}$$

$$= (-1)^m \prod_{\boldsymbol{\sigma}_0 \in \boldsymbol{\Delta}_0} K(\boldsymbol{\sigma}_0)^{\Lambda(\boldsymbol{\sigma}_0)} . \tag{201}$$

for an arbitrary gauge parameter (0-cochain) $\Lambda : C_0 \to \mathbb{Z}_2$ whose boundary value is constrained to be $\Lambda(\sigma_0) = 1$ for $\sigma_0 \in \Delta_0$. The action of $\boldsymbol{U}_g$ on $|\psi_{3\text{d}}\rangle$ is equivalent to the action of $U_g$ on $|\psi_{2d}\rangle$:

$$\boldsymbol{U}_g(\Lambda)|\psi_{3\text{d}}\rangle = \mathcal{U}_{CZ}\big(\mathcal{O}_{\text{bp}}(U_g|\psi_{2d}\rangle) \otimes |+\rangle_{\text{bulk}}\big) . \tag{202}$$

We now argue that the relation (202) is a manifestation of a new kind of bulk-boundary, or holographic, correspondence. In such a correspondence, a global symmetry of the boundary theory is identified with a gauge symmetry of the bulk theory. Indeed in the current set-up, the symmetry generator $U_g$ of the boundary becomes the product of $K(\boldsymbol{\sigma}_0) = X(\boldsymbol{\sigma}_0)Z(\boldsymbol{\partial}^*\boldsymbol{\sigma}_0)$ over $\boldsymbol{\sigma}_0 \in \boldsymbol{\Delta}_0$, and the operator $K(\boldsymbol{\sigma}_0)$ generates the gauge transformation of the bulk theory defined by the Hamiltonian

$$\boldsymbol{H} = -\sum_{\boldsymbol{\sigma}_1 \in \boldsymbol{\Delta}_1} X(\boldsymbol{\sigma}_1)Z(\boldsymbol{\partial}\boldsymbol{\sigma}_1) \qquad (203)$$

and the Gauss law constraint $X(\boldsymbol{\sigma}_0)Z(\boldsymbol{\partial}^*\boldsymbol{\sigma}_0) = 1$. It is a $(3+1)$-dimensional Ising model coupled to the topological $\mathbb{Z}_2$ gauge theory with gauge field $X(\boldsymbol{\sigma}_1)$, and has the cluster state $|\mathrm{gCS}_{(3,1)}\rangle$ as the unique ground state.

### B.  Gauge theory $M_{(3,2)}$

In this subsection, we use the asterisk $(*)$ to denote quantities (such as bulk $i$-cells $\boldsymbol{\sigma}_i^* \in \boldsymbol{\Delta}_i^*$) associated with the dual lattice. Let us consider the $\mathbb{Z}_2$ gauge theory defined by the Hamiltonian (15).

The electric $\mathbb{Z}_2$ one-form symmetry is generated by

$$U_g(z_1^*) := X(z_1^*) \qquad (204)$$

for $z_1^* \in \ker(\partial_1^*)$. The operator $X(z_1^*)$ commutes with the Hamiltonians (15). Moreover, the action of $X(z_1^*)$ on the physical Hilbert space is invariant under a local deformation of $z_1^*$ because of the Gauss law constraint (16), except when it crosses the location of an external charge and changes its sign. This means that $X(z_1^*)$ is a topological defect operator that generates the $\mathbb{Z}_2$ one-form symmetry under which Wilson lines are charged. For each generator of the dual 1-homology, the operator $X(z_1^*)$ defines a logical Pauli X operator in the toric code limit $\lambda \to \infty$ in the deconfined phase.

Next let us consider the Wilson loop operator

$$Z(z_1) \qquad (205)$$

for an arbitrary 1-cycle $z_1 \in Z_1$. The Wilson loop exhibits the area law in the confined phase and the perimeter law in the deconfined phase [20, 22]. In the $\mathbb{Z}_2$ topological field theory (realized in the low-energy limit with $\lambda > \lambda_c$ or in the toric code limit $\lambda \to \infty$ of our $\mathbb{Z}_2$ gauge theory), $Z(z_1)$ is a generator of the magnetic $\mathbb{Z}_2$ one-form symmetry [76]. Since $Z(z_1)$ does not commute with the electric term in the Hamiltonian of the gauge theory, the one-form symmetry is absent except in these limits. For each generator of the 1-homology, the operator $Z(z_1)$ defines a logical Pauli Z operator in the toric code limit $\lambda \to \infty$ in the deconfined phase.

Recall that the RBH cluster state is defined on the qubits placed on the edges and the faces of the three-dimensional cubic lattice $\mathcal{C}$, with qubits placed on edges

$\boldsymbol{\sigma}_1 \in \boldsymbol{\Delta}_1$ and faces $\boldsymbol{\sigma}_2 \in \boldsymbol{\Delta}_2$. The RBH cluster state $|\psi_\mathcal{C}\rangle$ is the simultaneous eigenstate, with eigenvalue $+1$, of the stabilizers $K(\boldsymbol{\sigma}_1) = X(\boldsymbol{\sigma}_1)Z(\boldsymbol{\partial}\boldsymbol{\sigma}_1)$ and $K(\boldsymbol{\sigma}_2) = X(\boldsymbol{\sigma}_2)Z(\boldsymbol{\partial}^*\boldsymbol{\sigma}_2)$. In the middle of the simulation, the system is again defined on the reduced lattice $\mathcal{C}_{\mathrm{red}}$. The state of the system differs from the RBH cluster state only on the boundary qubits which encode the gauge theory state and can be written again as (198), where this time $|+\rangle$ is the tensor product of copies of $|+\rangle$ over all the bulk qubits and the face qubits on the boundary. The same argument as in Section (VII A) applied to $U_g(z_1^*)$ gives the relation

$$\boldsymbol{U}_g(\boldsymbol{c}_2^*)|\psi_{3\mathrm{d}}\rangle = \mathcal{U}_{CZ}\big(\mathcal{O}_{\mathrm{bp}}(U_g(z_1^*)|\psi_{2d}\rangle)\otimes|+\rangle_{\mathrm{bulk}}\big), \quad (206)$$

where

$$\boldsymbol{U}_g(\boldsymbol{c}_2^*) := \mathcal{U}_{CZ}\big(\mathcal{O}_{\mathrm{bp}}U_g(z_1^*)\mathcal{O}_{\mathrm{bp}}^{-1}\big)\mathcal{U}_{CZ}^{-1}$$
$$\prod_{\boldsymbol{\sigma}_2^* \in \boldsymbol{\Delta}_2^* \setminus \Delta_2^*} K(\boldsymbol{\sigma}_2^*)^{a(\boldsymbol{c}_2^*;\boldsymbol{\sigma}_2^*)} \qquad (207)$$

$$= \pm \prod_{\boldsymbol{\sigma}_2^* \in \boldsymbol{\Delta}_2^*} K(\boldsymbol{\sigma}_2^*)^{a(\boldsymbol{c}_2^*;\boldsymbol{\sigma}_2^*)} \qquad (208)$$

and $\boldsymbol{c}_2^*$ is an arbitrary 2-chain on the dual lattice such that its restriction to the boundary coincides with $z_1^*$. The operator $K(\boldsymbol{\sigma}_2^*) = X(\boldsymbol{\sigma}_2^*)Z(\boldsymbol{\partial}^*\boldsymbol{\sigma}_2^*)$ generates the gauge transformation of the bulk theory defined by the Hamiltonian

$$\boldsymbol{H} = -\sum_{\boldsymbol{\sigma}_2 \in \boldsymbol{\Delta}_2} X(\boldsymbol{\sigma}_2)Z(\boldsymbol{\partial}\boldsymbol{\sigma}_2) \qquad (209)$$

and the Gauss law constraint $X(\boldsymbol{\sigma}_2^*)Z(\boldsymbol{\partial}^*\boldsymbol{\sigma}_2^*) = 1$. It is a $(3+1)$-dimensional generalized Ising model coupled to the topological $\mathbb{Z}_2$ gauge theory (with 1-form gauge symmetry) with 2-form gauge field $X(\boldsymbol{\sigma}_2)$, and has the RBH cluster state as the unique ground state. Thus the bulk-boundary correspondence discussed in Section VII A for the Ising model naturally generalizes to the gauge theory.

We summarize in Table IV the global and gauge symmetries of the resource state in the bulk and the simulated theory on the boundary.

## VIII.  CONCLUSIONS & DISCUSSION

In this work we introduced a family of resource states $\mathrm{gCS}_{(d,n)}$ (generalized cluster states), and showed that the Hamiltonian quantum simulation of Wegner's model $M_{(d,n)}$, with an $(n-1)$-form gauge field for the $\mathbb{Z}_2$ gauge group in $d$ spatial dimensions, can be implemented by specifically adapted single qubit measurements of $\mathrm{gCS}_{(d,n)}$. We devised methods to enforce the Gauss law constraint based on syndrome measurements and the energy penalty. We also generalized the simulation protocols to the gauge group $\mathbb{Z}_N$. By attaching ancillas and allowing two-qubit measurements, we can also perform the imaginary-time quantum simulation. We studied the

| | global symmetry / generator | gauge symmetry /generator |
|---|---|---|
| $\text{gCS}_{(d,n)}$ (bulk) | $(n-1)$-form / $(d-n+1)$-brane | $(n-1)$-form / $K(\boldsymbol{\sigma}_{n-1} = \boldsymbol{\sigma}^*_{d-n+1})$ |
| | $(d-n)$-form / $n$-brane | $(d-n)$-form / $K(\boldsymbol{\sigma}_n)$ |
| $M_{(d,n)}$ (boundary) | $(n-1)$-form / $(d-n)$-brane | $(n-2)$-form / $X(\partial^* \sigma_{n-2})$ |

TABLE IV. Symmetries studied in Sections VI and VII as well as in Appendix D. The $(n-1)$- and $(d-n)$-form symmetries of the generalized cluster state $\text{gCS}_{(d,n)}$ can be regarded both as global (Section VI) and gauge (Section VII and Appendix D) symmetries. The simulated model $M_{(d,n)}$ on the boundary has distinct but related global and gauge symmetries (if both exist). The global symmetry exists as a consequence of the Gauss law constraint. When $n = 1$, $M_{(d,n=1)}$ is the Ising model and has no gauge symmetry.

correspondence (for gauge group $\mathbb{Z}_N$) between the generalized cluster states and the statistical partition functions of $M_{(d,n)}$ regarded as classical spin models. We also proposed a measurement-based protocol for simulating the Kitaev Majorana chain. We demonstrated that $\text{gCS}_{(d,n)}$ has a symmetry-protected topological order with respect to generalized global symmetries that are related to the gauge symmetries of the simulated gauge theories.

For the model $M_{(d,n)}$ with the gauge symmetry generated by $G(\sigma_{n-2})$, the analogue of the 1-form symmetry of the resource state $\text{gCS}_{(3,2)}$ is the $(n-1)$-form symmetry of $\text{gCS}_{(d,n)}$. We expect that this higher-form symmetry can be used for the syndromes and the enforcement of gauge invariance for the model $M_{(3,2)}$ would generalize to $M_{(d,n)}$.

It is possible, at least formally, to make the gauge group continuous. In Appendix C, we discuss an approach to simulating the non-compact $U(1)$ gauge theory using the continuous-variable cluster state [77, 78]. The method we present for this gauge group should be regarded formal due to the divergence coming from integrating over non-compact variables, and in experiments the related issue is the imperfection of the continuous-variable cluster state due to the finite squeezing. Nonetheless, we draw readers' attention to recent development in generating large scale cluster states using photons [79–82]. At the moment, the best approach to simulating compact $U(1)$ theories is to take $N$ large in the model $M^{(\mathbb{Z}_N)}_{(d,n)}$. It would also be important to generalize the MBQS scheme to non-Abelian gauge groups.

Another future direction is the measurement-based simulation of more realistic high energy theories such as QED, QCD, the Standard Model and the Grand Unified Theories, as well as quantum-many body systems in condensed matter physics. In this respect, quantum simulation of Kitaev's Majorana chain we considered in

this work would give us a hint on how to combine Dirac or Weyl fermions in a resource state. It would also be interesting to relate the SPT order of the bulk resource states to the dynamical phases of the simulated models, where the bulk-boundary correspondence we observed in our work may be useful.

The presence of an SPT order has been suggested to be an important ingredient of resource states for the ability to perform the (universal) MBQC [83–94]. More recently, it was shown that certain quantum states with the long-range entanglement can be obtained with constant depth operations via measuring states in non-trivial SPT phases [60, 95–98], and there have been emerging new aspects of complexity in quantum-many body systems through the lens of measurements. In Appendix F we construct a tensor network representation of the generalized cluster state and show that a projective representation appears via the action of generalized global symmetries. It would be nice to find a continuous deformation of the tensor network within the corresponding SPT phase. As a related future direction, it would also be interesting to relate the ability to perform MBQS of a quantum-many body system with a symmetry to the SPT order of an appropriate resource state.

**ACKNOWLEDGEMENT**

We thank Lento Nagano for helpful discussions. HS thanks Tzu-Chieh Wei for helpful guidance and comments on manuscripts. We also acknowledge the usefulness of the textbook [99]. HS was partially supported by the Materials Science and Engineering Divisions, Office of Basic Energy Sciences of the U.S. Department of Energy under Contract No. DESC0012704. The work of TO is supported in part by the JSPS Grant-in-Aid for Scientific Research No. 21H05190.

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

$$R^{-1}(\theta)\hat{Q}R(\theta) = \cos(\theta)\,\hat{Q} - \sin(\theta)\,\hat{P}\,, \qquad (210)$$

$$R^{-1}(\theta)\hat{P}R(\theta) = \sin(\theta)\,\hat{Q} + \cos(\theta)\,\hat{P}\,. \qquad (211)$$

The Fourier transform is a special case with $F = R\left(\frac{\pi}{2}\right)$.

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

## Appendix A: Proof of equations

In this appendix, we prove some identities related to measurements. To make the presentation compact, we give proofs in the qudit case, which reduces to the qubit case by taking $N = 2$, $\omega = -1$, and $F = H$. We denote $F|s\rangle$ by $|\tilde{s}\rangle$, which are the $X$-eigenstates with eigenvalues $\omega^s$.

**Proof of** (34), (55), **and** (136)

Consider a general state $|\Psi\rangle$ on qudit 0. We wish to transport it to qudit 1 by a measurement in the basis $\{f(Z)|\tilde{s}\rangle\big|s = 0, 1, \ldots, N-1\}$, where $f$ is an arbitrary function such that $f(Z)^\dagger = f(Z)^{-1}$. The choice $f(Z) = 1$ corresponds to basis $\mathcal{M}_{(X)}$ in (54) and (132), and $f(Z) = e^{i(\xi Z + \text{h.c.})}$ corresponds to $\mathcal{M}_{(B)}$ in (52) and (135). We claim that the following identity holds:

$$_0\langle\tilde{s}|f(Z_0)^\dagger(CZ_{01})^\epsilon|\Psi\rangle_0|+\rangle_1 = \frac{1}{\sqrt{N}}F_1^{-\epsilon}Z_1^s f(Z_1)^\dagger|\Psi\rangle_1\,. \qquad (A1)$$

Here $\epsilon = \pm 1$. The equations (34), (55), and (136) follow from (A1).

To prove (A1), we expand $|+\rangle$ in the $Z$-basis and use the relations $Z|\tilde{s}\rangle = |\widetilde{s-1}\rangle$ and $F^{\pm 1}|s\rangle = |\widetilde{\pm s}\rangle$. The

LHS of (A1) becomes

$$\frac{1}{\sqrt{N}}\sum_{t=0}^{N-1}\langle\tilde{s}|Z^{\epsilon t}f(Z)^\dagger|\Psi\rangle|t\rangle_1$$

$$= \frac{1}{\sqrt{N}}\sum_{t=0}^{N-1}\langle\widetilde{\epsilon t}|Z^s f(Z)^\dagger|\Psi\rangle|t\rangle_1$$

$$= \frac{1}{\sqrt{N}}\sum_{t=0}^{N-1}\langle t|F^{-\epsilon}Z^s f(Z)^\dagger|\Psi\rangle|t\rangle_1\,, \qquad (A2)$$

which equals the RHS of (A1).

**Proof of** (30), (51), **and** (131)

Suppose that the $|+\rangle$ state (115) of qudit 0 is entangled with a general state $|\Psi\rangle$ of qudits $1, \ldots, m$ by CZ gates or their inverses. We consider measuring qudit 0 in the basis $\{f(X)|s\rangle\big|s = 0, 1, \ldots, N-1\}$, where $f$ is again an arbitrary function. The choice $f(X) = e^{i(\xi X + \text{h.c.})}$ corresponds to basis $\mathcal{M}_{(A)}$. The effect of such a measurement on the state $|\Psi\rangle$ is expressed as the identity

$$_0\langle s|f(X_0)^\dagger\prod_{a=1}^m(CZ_{0a})^{\epsilon_a}|+\rangle_0|\Psi\rangle_{1\ldots m}$$

$$= \frac{1}{\sqrt{N}}f\left(\prod_{a=1}^m(Z_a)^{-\epsilon_a}\right)^\dagger\left(\prod_{a=1}^m Z_a^{\epsilon_a}\right)^s|\Psi\rangle_{1\ldots m}\,. \qquad (A3)$$

The equations (30), (51), and (131) follow from (A3).

To prove (A3), we again expand $|+\rangle$ in the $Z$-basis, insert $1 = \sum_u |\tilde{u}\rangle\langle\tilde{u}|$, and use the relation $\langle\tilde{u}|t\rangle = \omega^{tu}/\sqrt{N}$. The LHS of (A3) becomes

$$\frac{1}{N^{3/2}}\sum_{t,u}f(\omega^u)^*\omega^{-su}\left(\omega^u\prod_a Z_a^{\epsilon_a}\right)^t|\Psi\rangle_{1\ldots m}\,. \qquad (A4)$$

The summation over $t$ forces $\omega^u$ to equal $\prod_a Z_a^{-\epsilon_a}$, proving (A3).

**A qumode identity**

We have

$$_0\langle m_p|f(\hat{Q}_0)^\dagger(CZ_{01})^\epsilon|\Psi\rangle_0|0_p\rangle_1 = \left(F^\epsilon e^{-im\hat{Q}}f(\hat{Q})^\dagger|\Psi\rangle\right)_1. \qquad (A5)$$

The proof is similar to the one for (A1).

Next, consider a qumode 0 in state $|0_p\rangle$ and qumodes $1,\ldots,m$ in a general state $|\Psi\rangle$. We prove the relation

$$_0\langle s_q|f(\hat{P}_0)^\dagger\prod_{a=1}^n(CZ_{0a})^{\epsilon_a}|0_p\rangle_0|\Psi\rangle_{1\ldots m}$$

$$= \frac{1}{\sqrt{2\pi}}\left(\prod_{a=1}^m e^{i\epsilon_a\hat{Q}_a}\right)^s f\left(\sum_a \epsilon_a\hat{Q}_a\right)^\dagger|\Psi\rangle_{1\ldots m}\,. \qquad (A6)$$

The relation (A6) can be shown by using the definition $CZ_{0a} = e^{i\hat{Q}_0\hat{Q}_a}$ and by noting that

$$e^{-iu\hat{Q}_a\hat{Q}_b}F(\hat{P}_a)e^{iu\hat{Q}_a\hat{Q}_b} = F(\hat{P}_a + u\hat{Q}_b) \qquad (A7)$$

for a general function $F$.

## Appendix B: Continuous-time limit of model $M_{(d,n)}^{(\mathbb{Z}_N)}$

In this section, we give a derivation of the continuous-time limit for the model $M_{(d,n)}^{(\mathbb{Z}_N)}$, generalizing the discussion for gauge theories [20, 106, 107]. Consider the $\mathbb{Z}_N$ variable $u \in \{1, \omega, ..., \omega^{N-1}\}$ with $\omega = e^{2\pi i/N}$. We define for a $k$-chain $c_k$,

$$u(c_k) = \prod_{\sigma_k} u^{a(c_k;\sigma_k)} . \tag{B1}$$

Consider an anisotropic action

$$I = -J_t \sum_{\boldsymbol{\sigma}_n^t} u(\partial \boldsymbol{\sigma}_n^t) - J_s \sum_{\boldsymbol{\sigma}_n^s} u(\partial \boldsymbol{\sigma}_n^s) + \text{h.c.} . \tag{B2}$$

Here, $\boldsymbol{\sigma}_n^s$ ($\boldsymbol{\sigma}_n^t$) is $n$-cells stretched in spatial (temporal) directions. The gauge transformation that leaves the action invariant is defined for each $(n-2)$-cell and given by

$$\mathcal{G}(\boldsymbol{\sigma}_{n-2}) : u_{\boldsymbol{\sigma}_{n-1}} \to \omega^{a(\boldsymbol{\sigma}_{n-2};\partial\boldsymbol{\sigma}_{n-1})} u_{\boldsymbol{\sigma}_{n-1}} \ \forall \boldsymbol{\sigma}_{n-1} \in \Delta_{n-1} . \tag{B3}$$

We use the gauge transformation to bring temporal fields to unity,

$$u(\boldsymbol{\sigma}_{n-1}^t) = 1 \quad \text{with} \quad \boldsymbol{\sigma}_{n-1}^t = \boldsymbol{\sigma}_{i_1,\cdots,i_{n-2},d}(x) . \tag{B4}$$

Then, using (117), the first term in the action becomes

$$u(\partial \boldsymbol{\sigma}_n^t) + \text{h.c.} = \left(u(\boldsymbol{\sigma}_{i_1,\cdots,i_{n-1}}(x+e_d))\right)^{(-1)^{n-1}} \left(u(\boldsymbol{\sigma}_{i_1,\cdots,i_{n-1}}(x))\right)^{(-1)^n} + \text{h.c.}$$
$$= u(\boldsymbol{\sigma}_{i_1,\cdots,i_{n-1}}(x+e_d))u(\boldsymbol{\sigma}_{i_1,\cdots,i_{n-1}}(x))^* + \text{h.c.} . \tag{B5}$$

Now the two factors on the right hand side are defined on the same $(n-1)$-cell in respective time slices, thus we can simply express it as

$$u(\sigma_{n-1}^{(j+1)})u(\sigma_{n-1}^{(j)})^* + \text{h.c.} . \tag{B6}$$

We can further rewrite it using a norm and we obtain

$$I = -\sum_j \left[ J_t \sum_{\sigma_{n-1}} \left(2 - |u(\sigma_{n-1}^{(j+1)}) - u(\sigma_{n-1}^{(j)})|^2\right) + J_s \sum_{\sigma_n} (u(\partial\sigma_n^{(j)}) + \text{h.c.}) \right]$$
$$\overset{\text{redef.}}{\to} \sum_j \left[ J_t \sum_{\sigma_{n-1}} |u(\sigma_{n-1}^{(j+1)}) - u(\sigma_{n-1}^{(j)})|^2 - J_s \sum_{\sigma_n} (u(\partial\sigma_n^{(j)}) + \text{h.c.}) \right] . \tag{B7}$$

Now, we consider the partition function $Z = \sum_{\{u\}} e^{-\beta I[\{u\}]}$, and express it as the trace of a product of transfer matrices, $Z = \text{tr}[(e^{-\tau H})^T]$. We will identify the parameters and take the continuum limit as follows:

$$\beta J_t \to \infty \tag{B8}$$
$$\beta J_s \to \lambda e^{-\beta J_t C_1^t} \tag{B9}$$
$$\tau = e^{-\beta J_t C_1^t} \tag{B10}$$

with $C_1^t \equiv 2(1 - \cos(\frac{2\pi i}{N}))$.

The diagonal element of the transfer matrix between $j$

and $j+1$ is identified as

$$\langle \{u\}|(1 - \tau H)|\{u\}\rangle \simeq e^{\beta J_s \sum_{\sigma_n} (u(\partial\sigma_n^{(j)}) + \text{h.c.})}$$
$$\simeq 1 + \beta J_s \sum_{\sigma_n} (u(\partial\sigma_n^{(j)}) + \text{h.c.}) , \tag{B11}$$

which gives us the identification for the diagonal part of the Hamiltonian with the $\mathbb{Z}_N$ phase operator,

$$H_{\text{diag}} = -\lambda \sum_{\sigma_n} (Z(\partial\sigma_n) + \text{h.c.}) . \tag{B12}$$

Next, we consider a single-shift transition, $\{u\}_{\Delta_{n-1}} \to \{u'\}_{\Delta_{n-1}} = \{\omega^{\pm 1} u_{\sigma_{n-1}}\} \cup \{u\}_{\Delta_{n-1}\setminus\sigma_{n-1}}$. This gives the

minimum energy cost $2J_t(1 - \cos(\frac{2\pi i}{N})) \equiv J_t C_1^t$ from the temporal term and $J_s C_1^s$ from the spatial term, the latter of which depends on specific field configurations. We identify in the continuous-time limit as

$$\langle \{u'\}|(-\tau H)|\{u\}\rangle = e^{-\beta J_t C_1^t} e^{\beta J_s C_1^s}$$
$$= \tau + \mathcal{O}(\tau^2) . \tag{B13}$$

Transitions that involve more than one shift of the field become higher order contributions on the right hand side. Thus we can identify

$$H_{\text{off-diag}} = - \sum_{\sigma_{n-1}} (X_{\sigma_{n-1}} + \text{h.c.}) . \tag{B14}$$

The total Hamiltonian in the continuous-time limit is then

$$H = - \sum_{\sigma_{n-1}} (X_{\sigma_{n-1}} + \text{h.c.}) - \lambda \sum_{\sigma_n} (Z(\partial\sigma_n) + \text{h.c.}) . \tag{B15}$$

Finally, we examine the gauge symmetry of this theory. We have used the gauge transformation to bring the temporal fields to unity, and there are residual gauge transformations that leave the temporal fields invariant. That is, we consider the gauge transformation constant over time:

$$\prod_j \mathcal{G}(\sigma_{n-2}^{(j)}) \tag{B16}$$

where $\sigma_{n-2}^{(j)} = \sigma_{i_1,\cdots,i_{n-2}}(x,j)$ with $x$ being the spatial coordinate, $j$ being the time coordinate, and $i_1 < ... < i_{n-2} < d$. Namely, the product is taken over time for $(n-2)$-cells that do not stretch in the time ($d$-th) direction. This leaves us the gauge transformation that does not depend on time, and it can be written as

$$G(\sigma_{n-2}) = \prod_{\sigma_{n-1}} X_{\sigma_{n-1}}^{a(\partial\sigma_{n-1};\sigma_{n-2})} . \tag{B17}$$

**Appendix C: Gauge group $\mathbb{R}$**

**1. Hamiltonian formulation of $M_{(d,n)}^{(\mathbb{R})}$**

In this section, we consider the model $M_{(d,n)}$ with continuous groups. The model consists of the gauge degrees of freedom on the $(n-1)$-cells $\sigma_{n-1}$ of the $(d-1)$-dimensional lattice. The group element on each $\sigma_{n-1}$ is given by

$$U_{\sigma_{n-1}} = e^{i\theta_{\sigma_{n-1}}} \tag{C1}$$

with the gauge field $\theta_{\sigma_{n-1}}$.

When $\theta_{\sigma_{n-1}} \in [0, 2\pi)$, we refer the group as $U(1)$. On the other hand, $\theta_{\sigma_{n-1}} \in (-\infty, +\infty)$, we refer the group as *non-compact $U(1)$* or simply $\mathbb{R}$. Our MBQS considered in

the following part corresponds to the non-compact $U(1)$ group.

We write its conjugate momentum as

$$L_{\sigma_{n-1}} = -i\frac{\partial}{\partial\theta_{\sigma_{n-1}}} . \tag{C2}$$

The canonical commutation relation is

$$[\theta_{\sigma_{n-1}}, L_{\sigma'_{n-1}}] = i\delta_{\sigma_{n-1},\sigma'_{n-1}} . \tag{C3}$$

The Hamiltonian is given by

$$H = \frac{g^2}{2} \sum_{\sigma_{n-1}\in\Delta_{n-1}} L_{\sigma_{n-1}}^2 - \frac{1}{g^2} \sum_{\sigma_n\in\Delta_n} \cos[\theta(\partial\sigma_n)] . \tag{C4}$$

with $\theta(\partial\sigma_n) := \sum_{\sigma_1\in\partial\sigma_n} a(\partial\sigma_n;\sigma_{n-1})\theta_{\sigma_{n-1}}$. The Gauss law reads

$$G(\sigma_{n-2}) = Q(\sigma_{n-2}) \tag{C5}$$
$$G(\sigma_{n-2}) := - \sum_{(\partial\sigma_{n-1})_+\ni\sigma_{n-2}} L_{\sigma_{n-1}} + \sum_{(\partial\sigma_{n-1})_-\ni\sigma_{n-2}} L_{\sigma_{n-1}} \tag{C6}$$

for each $(n-2)$-cell $\sigma_{n-2} \in \Delta_{n-2}$. Here $Q(\sigma_{n-2}) \in \mathbb{R}$ is the external charge. We add to the Hamiltonian an energy cost term,

$$\Lambda(G(\sigma_{n-2}) - Q(\sigma_{n-2}))^2 , \tag{C7}$$

($\Lambda > 0$) to enforce the Gauss law constraint.

**2. MBQS of $M_{(d,n)}^{(\mathbb{R})}$**

We consider MBQS of the model $M_{(d,n)}^{(\mathbb{R})}$ using the continuous-variable cluster state [77, 78]. We introduce the qumode basis by the eigenstate of $\hat{P}$ and $\hat{Q}$ operators, *i.e.* $[\hat{Q}, \hat{P}] = i$, such that

$$\hat{P}|t_p\rangle = t|t_p\rangle, \ \hat{Q}|s_q\rangle = s|s_q\rangle, \tag{C8}$$

The Pauli operators are generalized to the Weyl-Heisenberg operators as

$$Z(s) = e^{is\hat{Q}}, \ X(s) = e^{-is\hat{P}} . \tag{C9}$$

Note that

$$Z(s)|t_p\rangle = |(t+s)_p\rangle, \ X(s)|t_q\rangle = |(t+s)_q\rangle . \tag{C10}$$

The $|+\rangle$ state corresponds to $|0_p\rangle$ and the $CZ$ gate is defined by

$$CZ_{a,b} = e^{i\hat{Q}_a\hat{Q}_b} \tag{C11}$$

and the Fourier transformation, which corresponds to the Hadamard gate, is defined by

$$F = \exp\left[\frac{i\pi}{4}(\hat{P}^2 + \hat{Q}^2)\right] . \qquad (C12)$$

This Fourier transformation flips the basis as $F|s_q\rangle = |s_p\rangle$ [108]. Also, the following equalities hold [109]:

$$F^4 = I, \; F^\dagger \hat{Q} F = -\hat{P}, \; F^\dagger \hat{P} F = \hat{Q} . \qquad (C13)$$

The measurement pattern is as follows:

(1) $\boldsymbol{\sigma}_n = \sigma_n \times \{j\}$

Let $D_P^\phi = e^{i\phi(\hat{P})}$ and consider the measurement in the basis

$$\mathcal{M}_{(\mathbb{R},A)} := \left\{ D_P^\phi X(m)|0_q\rangle \,\middle|\, m \in \mathbb{R} \right\} . \qquad (C14)$$

(2) $\boldsymbol{\sigma}_{n-1} = \sigma_{n-1} \times \{j\}$ Consider the measurement in the basis

$$\mathcal{M}_{(X)} := \{ Z(m)|0_p\rangle \,|\, m \in \mathbb{R} \} . \qquad (C15)$$

(3) $\boldsymbol{\sigma}_1 = \sigma_0 \times [j, j+1]$ We impose the Gauss law constraint by introducing the energy cost term. We consider the measurement basis

$$\mathcal{M}_{(\mathbb{R},A)} := \left\{ D_P^\phi X(m)|0_q\rangle \,\middle|\, m \in \mathbb{R} \right\} . \qquad (C16)$$

(4) $\boldsymbol{\sigma}_1 = \sigma_1 \times [j, j+1]$ Let $D_Q^\phi$ be $D_Q^\phi = e^{i\phi(\hat{Q})}$. Consider the measurement basis

$$\mathcal{M}_{(\mathbb{R},B)} := \left\{ D_Q^\phi Z(m)|0_p\rangle \,\middle|\, m \in \mathbb{R} \right\} . \qquad (C17)$$

For the A-type measurement we use the identity (A6) and For the B-type measurement we use the identity (A5). The unitaries from the sequence (i)-(iv) result in the following unitary:

$$\prod_{\sigma_{n-1}} [F^{(-1)^{n-1}} e^{-im_4 \hat{Q}} e^{-i\phi_4(\hat{Q})}]_{\sigma_{n-1}}$$

$$\times \prod_{\sigma_{n-2}} \left[ \frac{1}{\sqrt{2\pi}} \left( \prod_{\sigma_{n-1}} e^{ia(\partial\sigma_{n-1};\sigma_{n-2})\hat{Q}_{\sigma_{n-1}}} \right)^{s_3(\sigma_{n-2})} \exp\left( -i\phi_3 \left( \sum_{\sigma_{n-1}} a(\partial\sigma_{n-1};\sigma_{n-2})\hat{Q}_{\sigma_1} \right) \right) \right]$$

$$\times \prod_{\sigma_{n-1}} \left( F^{(-1)^n} e^{-im_2 \hat{Q}} \right)_{\sigma_{n-1}}$$

$$\times \prod_{\sigma_n} \left[ \frac{1}{\sqrt{2\pi}} \left( \prod_{\sigma_{n-1}} e^{ia(\partial\sigma_n;\sigma_{n-1})\hat{Q}_{\sigma_{n-1}}} \right)^{s_1(\sigma_n)} \exp\left( -i\phi_1 \left( \sum_{\sigma_{n-1}} a(\partial\sigma_n;\sigma_{n-1})\hat{Q}_{\sigma_{n-1}} \right) \right) \right]$$

$$= \prod_{\sigma_{n-1}} [e^{-im_4(-1)^{n-1}\hat{P}} e^{-i\phi_4((-1)^{n-1}\hat{P})}]_{\sigma_{n-1}}$$

$$\times \prod_{\sigma_{n-2}} \left[ \frac{1}{\sqrt{2\pi}} \left( \prod_{\sigma_{n-1}} e^{i(-1)^{n-1}a(\partial\sigma_{n-1};\sigma_{n-2})\hat{P}_{\sigma_{n-1}}} \right)^{s_3(\sigma_{n-2})} \exp\left( -i\phi_3 \left( \sum_{\sigma_{n-1}} (-1)^{n-1}a(\partial\sigma_{n-1};\sigma_{n-2})\hat{P}_{\sigma_1} \right) \right) \right]$$

$$\times \prod_{\sigma_{n-1}} e^{-im_2\hat{Q}_{\sigma_{n-1}}}$$

$$\times \prod_{\sigma_n} \left[ \frac{1}{\sqrt{2\pi}} \left( \prod_{\sigma_{n-1}} e^{ia(\partial\sigma_n;\sigma_{n-1})\hat{Q}_{\sigma_{n-1}}} \right)^{s_1(\sigma_n)} \exp\left( -i\phi_1 \left( \sum_{\sigma_{n-1}} a(\partial\sigma_n;\sigma_{n-1})\hat{Q}_{\sigma_{n-1}} \right) \right) \right] \qquad (C18)$$

When the measurement outcome is $s_1 = m_2 = s_3 = m_4 = 0$ everywhere, we choose the function $\phi_i$ ($i = 1, 3, 4$) as follows:

$$\phi_1(x) = -\delta t \frac{1}{g^2} \cos(x) , \quad \phi_3(x) = \Lambda \, \delta t \, (x - q)^2 , \qquad (C19)$$

$$\phi_4(x) = \delta t \frac{g^2}{2} x^2 . \qquad (C20)$$

with $q$ chosen as $Q(\sigma_{n-2})$ for the measurement at $\sigma_{n-2} \in \Delta_{n-2}$. By identifying the gauge field $\theta_{\sigma_{n-1}}$ with $\hat{Q}_{\sigma_{n-1}}$ and the conjugate momentum $L_{\sigma_{n-1}}$ with $(-1)^{n-1}\hat{P}_{\sigma_{n-1}}$, we obtain the desired time evolution unitary for the non-compact $U(1)$ lattice gauge theory:

$$e^{-iH\delta t} \simeq \prod_{\sigma_{n-1}} \exp\left(-i\delta t \frac{g^2}{2} L_{\sigma_{n-1}}^2\right) \prod_{\sigma_{n-2}} \exp\left(-i\Lambda\,\delta t\,(G(\sigma_{n-2}) - Q(\sigma_{n-2}))^2\right) \prod_{\sigma_n} \exp\left(i\delta t \frac{1}{g^2}\cos(\theta_{\sigma_n})\right) . \qquad \text{(C21)}$$

To deal with the byproduct operators with $m_i \neq 0$, we just need to shift the argument of the function $\phi_i(s)$. This follows from the fact that we only have $X(s)$ and $Z(s)$ for byproduct operators, so arranging the unitary in the appropriate form only requires the commutation relations $\phi(\hat{Q})X(s) = X(s)\phi(\hat{Q}+s)$ and $\phi(\hat{P})Z(s) = Z(s)\phi(\hat{P}+s)$ for a general function $\phi(x)$.

Experimentally, if one wishes to realize the MBQS with the measurement bases (C19) using photons, non-Gaussian function $\phi_1(x) = \delta t \frac{1}{g^2}\cos(x)$ would be hard to realize. One may instead consider a regime where the approximation $\cos(x) \simeq 1 - \frac{1}{2}x^2$ is valid: namely, the continuum limit, where the lattice spacing $a$ is small. One can also improve the approximation by expressing higher-order terms in cosine using an expansion with Gaussian gates. See *e.g.* [110, 111].

### 3. A formal correspondence between $\mathrm{gCS}_{(d,n)}^{(\mathbb{R})}$ and the Euclidean path integral

For gauge group $\mathbb{R}$, we define

$$|\phi_{(d,n)}^{(\mathbb{R})}(J)\rangle := \left(e^{-i\phi(\hat{P})}|0_p\rangle\right)^{\otimes \Delta_n} |0_p\rangle^{\otimes \Delta_{n-1}} . \qquad \text{(C22)}$$

We note that $|0_p\rangle = (2\pi)^{-1/2} \int \mathrm{d}\theta |\theta_q\rangle$. We find

$$\langle \phi_{(d,n)}^{(\mathbb{R})}(J) | \mathrm{gCS}_{(d,n)}^{(\mathbb{R})} \rangle = \langle 0_p|^{\otimes \Delta_{n-1}} \prod_{\sigma_n} \frac{1}{\sqrt{2\pi}} \exp(i\phi(\sum_{\sigma_{n-1} \in \partial\sigma_n} a(\partial\sigma_n; \sigma_{n-1})\hat{Q}_{\sigma_{n-1}}))|0_p\rangle^{\otimes \Delta_{n-1}}$$

$$= \frac{1}{(2\pi)^{|\Delta_{n-1}|}} \frac{1}{(2\pi)^{|\Delta_n|/2}} \int \prod_{\sigma_{n-1}} \mathrm{d}\theta_{\sigma_{n-1}} \mathrm{d}\theta'_{\sigma_{n-1}} \langle \theta_q | \theta'_q \rangle_{\sigma_{n-1}} \prod_{\sigma_n} \exp(i\phi(\sum_{\sigma_{n-1} \in \partial\sigma_n} a(\partial\sigma_n; \sigma_{n-1})\theta_{\sigma_{n-1}}))$$

$$= \frac{1}{(2\pi)^{|\Delta_{n-1}|}} \frac{1}{(2\pi)^{|\Delta_n|/2}} \int \prod_{\sigma_{n-1}} \mathrm{d}\theta_{\sigma_{n-1}} \exp(i\sum_{\sigma_n} \phi(\sum_{\sigma_{n-1} \in \partial\sigma_n} a(\partial\sigma_n; \sigma_{n-1})\theta_{\sigma_{n-1}}))$$

$$= \frac{1}{(2\pi)^{|\Delta_{n-1}|}} \frac{1}{(2\pi)^{|\Delta_n|/2}} \int \prod_{\sigma_{n-1}} \mathrm{d}\theta_{\sigma_{n-1}} \exp(iI[\{\theta_{\sigma_{n-1}}\}]) \qquad \text{(C23)}$$

with

$$\phi(x) = J(e^{ix} + e^{-ix}) . \qquad \text{(C24)}$$

We note that the right hand side of the equation (C23) is divergent since the integration is over $(-\infty, +\infty)$ and the integrand is periodic. Therefore the equation is formal and requires a regularization (including division by the divergent volume of the gauge group) or gauge fixing. Note that the rotation $J \to i\beta J$ brings the expression to the Euclidean path integral.

## Appendix D: Stabilizer operators as gauge transformations

In [42, 112], a reinterpretation of the MBQC as gauge theory is given. In particular, stabilizer operators of the 1-dimensional cluster state ($\mathrm{gCS}_{(1,1)}$ in our notation) are reinterpreted as gauge transformations. In this appendix we show that this extends very naturally to $\mathrm{gCS}_{(d,n)}$ for general $d$ and $n$.

Let us consider the generalized stabilizer state $|\mathrm{gCS}_{(d,n)}\rangle$ defined by the stabilizers

$$K(\sigma_n) = X(\sigma_n)Z(\partial\sigma_{n-1}),$$
$$K(\sigma_{n-1}) = X(\sigma_{n-1})Z(\partial^*\sigma_{n-1}). \qquad \text{(D1)}$$

(In this Appendix, we use normal fonts throughout.) Let

us consider measuring the operators (corresponding to the B-type measurement defined in Section III)

$$B[q_{n-1}(\sigma_{n-1})]$$
$$= \cos(2\xi)X(\sigma_{n-1}) + (-1)^{q_{n-1}(\sigma_{n-1})}\sin(2\xi)Y(\sigma_{n-1}) \tag{D2}$$

and their measurement results $s^*_{d-n+1}(\sigma^*_{d-n+1})$ at $\sigma_{n-1} = \sigma^*_{d-n+1}$ [113], as well as

$$B[q^*_{d-n}(\sigma^*_{d-n})]$$
$$= \cos(2\xi)X(\sigma^*_{d-n}) + (-1)^{q^*_{d-n}(\sigma^*_{d-n})}\sin(2\xi)Y(\sigma^*_{d-n}) \tag{D3}$$

and their measurement results $s_n(\sigma_n)$ at $\sigma^*_{d-n} = \sigma_n$. Recall that a $k$-chain is a linear map

$$C_k \to \mathbb{Z}_2 , \tag{D4}$$

where $C_k$ is the group consisting $k$-chains. We denote the group of $k$-cochains by $C^k$. The corresponding objects on the dual lattices will be denoted by $C^*_k$ and $(C^*)^k$. We regard various quantities as cochains on the primal and dual lattices as cochains:

$$q_{n-1} \in C^{n-1} , \quad s^*_{d-n+1} \in (C^*)^{d-n+1} , \tag{D5}$$

$$q^*_{d-n} \in (C^*)^{d-n} , \quad s_n \in C^n . \tag{D6}$$

Let us consider gauge transformations with gauge parameters $\Lambda_{n-1} \in C^{n-1}$ and $\Lambda^*_{d-n} \in (C^*)^{d-n}$, generated by the operators

$$K(\Lambda_{n-1}) := \prod_{\sigma_{n-1} \in \Delta_{n-1}} K(\sigma_{n-1})^{\Lambda_{n-1}(\sigma_{n-1})} , \tag{D7}$$

$$K(\Lambda^*_{d-n}) := \prod_{\sigma^*_{d-n} \in \Delta^*_{d-n}} K(\sigma^*_{d-n})^{\Lambda^*_{d-n}(\sigma^*_{d-n})} . \tag{D8}$$

Conjugation by the operator $K(\Lambda_{n-1})$ induces the transformations $B[q_{n-1}(\sigma_{n-1})] \mapsto B[(q_{n-1} + \Lambda_{n-1})(\sigma_{n-1})]$ and $B[q^*_{d-n}(\sigma^*_{d-n})] \mapsto (-1)^{(d\Lambda_{n-1})(\sigma_n)}B[q^*_{d-n}(\sigma^*_{d-n})]$. Similarly, conjugation by $K(\Lambda^*_{d-n})$ gives $B[q_{n-1}(\sigma_{n-1})] \mapsto (-1)^{(d^*\Lambda^*_{d-n})(\sigma^*_{d-n+1})}B[q_{n-1}(\sigma_{n-1})]$ and $B[q^*_{d-n}(\sigma^*_{d-n})] \mapsto B[(q^*_{d-n} + \Lambda^*_{d-n})(\sigma^*_{d-n})]$. Taken together, the gauge transformations of the parameters are

$$q_{n-1} \mapsto q_{n-1} + \Lambda_{n-1} , \quad s_n \mapsto s_n + d\Lambda_{n-1} ,$$
$$s^*_{d-n+1} \mapsto s^*_{d-n+1} + d^*\Lambda^*_{d-n} , \quad q^*_{d-n} \mapsto q^*_{d-n} + \Lambda^*_{d-n} . \tag{D9}$$

Next, let us consider the operators (corresponding to the A-type measurement)

$$A[p^*_{d-n+1}(\sigma^*_{d-n+1})]$$
$$= \cos(2\xi)Z(\sigma^*_{d-n+1})$$
$$+ (-1)^{p^*_{d-n+1}(\sigma^*_{d-n+1})}\sin(2\xi)Y(\sigma^*_{d-n+1}) \tag{D10}$$

and their measurement results $t_{n-1}(\sigma_{n-1})$ at $\sigma_{n-1} = \sigma^*_{d-n+1}$, as well as

$$A[p_n(\sigma_n)] = \cos(2\xi)Z(\sigma_n) + (-1)^{p_n(\sigma_n)}\sin(2\xi)Y(\sigma_n) \tag{D11}$$

and their measurement results $t^*_{d-n}(\sigma^*_{d-n})$ at $\sigma^*_{d-n} = \sigma_n$. Considerations similar to the above give the gauge transformations

$$t_{n-1} \mapsto t_{n-1} + \Lambda_{n-1} , \quad p_n \mapsto p_n + d\Lambda_{n-1} ,$$
$$p^*_{d-n+1} \mapsto p^*_{d-n+1} + d^*\Lambda^*_{d-n} , \quad t^*_{d-n} \mapsto t^*_{d-n} + \Lambda^*_{d-n} . \tag{D12}$$

We confirmed, in the $(d,n) = (3,2)$ case, that the simulated unitary evolution including the byproduct operators in (58) is invariant under the gauge transformations (D9) and (D12) in the bulk. Note that at the boundary $x_3 = 0$, the stabilizer $K(\boldsymbol{\sigma}_1)$ $(\boldsymbol{\sigma}_1 = \sigma_1 \times \{0\})$ is not a symmetry of the resource state with an arbitrary initial wave function $|\psi_{2d}\rangle_{\Delta_1 \times \{0\}}$, so there is not a gauge transformation corresponding to this stabilizer.

## Appendix E: Gauging map via minimally coupled gauge fields

The aim of this appendix is to derive the gauging map (181) by minimally coupling gauge fields. For simplicity we use normal fonts and focus on the gauging map $\Gamma : \mathcal{H}^{sym}_n \to \mathcal{H}^{sym}_{n-1}$. The other half of (181) can be treated in the same way.

Minimal coupling means introducing new qubits on $(n-1)$-cells and replacing the operator $Z(\partial^*\sigma_{n-1})$ by the product $Z(\partial^*\sigma_{n-1})Z'(\sigma_{n-1})$, where the gauge field $Z'(\sigma_{n-1})$ is the Pauli Z operator for the new qubit on $\sigma_{n-1}$. The product is invariant under the gauge transformation generated by

$$G(\sigma_n) = X(\sigma_n)X'(\partial\sigma_n) . \tag{E1}$$

The spaces $\mathcal{H}^{sym}_n$ and $\mathcal{H}^{sym}_{n-1}$ consist of the states invariant under the symmetry operators $X(z_n)$ and $Z'(z_{n-1})$, for which $z_n$ and $z_{n-1}$ satisfy $\partial z_n = 0$ and $\partial^* z_{n-1} = 0$, respectively. Let $\mathcal{H}^{inv}_n \subset \mathcal{H}_n$ be the space of the states invariant under $G(\sigma_n)$. We define the linear map $\Xi : \mathcal{H}^{sym}_n \to \mathcal{H}^{inv}_n \otimes \mathcal{H}^{sym}_{n-1}$ such that $|\psi\rangle \in \mathcal{H}^{sym}_n$ is mapped to

$$\Xi(|\psi\rangle) := \mathcal{N}_0 \prod_{\sigma_n \in \Delta_n} \frac{1 + G(\sigma_n)}{2}|\psi\rangle \otimes |0\rangle' , \tag{E2}$$

where $\mathcal{N}_0$ is a normalization constant. We can expand

$$|\psi\rangle = \sum_{c_n \in C_n} \alpha(c_n)|c_n\rangle , \tag{E3}$$

where $\alpha(c_n)$ are complex coefficients. We note that inserting $\prod_{\sigma_n} G(\sigma_n)^{a(c_n;\sigma_n)}$ in front of $|c_n\rangle$ does not

change (E2). Thus we can write

$$\Xi(|\psi\rangle) = \mathcal{N}_0 \sum_{c_n \in C_n} \alpha(c_n) \prod_{\sigma_n \in \Delta_n} \frac{1 + G(\sigma_n)}{2} |0\rangle \otimes |\partial c_n\rangle' .$$
(E4)

Stripping off $(1 + G(\sigma_n))/2|0\rangle$ leads to the gauging map

$$\Gamma(|\psi\rangle) := \mathcal{N} \sum_{c_n \in C_n} \alpha(c_n)|\partial c_n\rangle' .$$
(E5)

The new normalization constant $\mathcal{N}$ can be determined by computing the norm of $\Gamma(|\psi\rangle)$: $\mathcal{N} = |Z_n|^{-1/2}$, where $|Z_n|$ is the number of $n$-cycles.

The gauging map of operators, $A \mapsto A^\Gamma$, is given by eliminating unprimed operators by solving the Gauss law constraint $G(\sigma_n) = 1$ for primed operators.

## Appendix F: Tensor network representation of gCS$_{d,n}$

In this appendix we study a tensor network representation of the generalized cluster state gCS$_{d,n}$. Since we only consider the $d$-dimensional bulk cell complex, we use non-bold symbols to simplify notations.

Our starting point is the expression (25) in terms of the entangler. By introducing complete sets of states, we write

$$|\text{gCS}_{d,n}\rangle = \sum_{c_n \in C_n} |c_n\rangle\langle c_n| \prod_{\substack{\sigma_{n-1} \in \Delta_{n-1} \\ \sigma_n \in \Delta_n}} CZ_{\sigma_{n-1},\sigma_n}^{a(\partial\sigma_n;\sigma_{n-1})}$$
$$\times |+\rangle^{\otimes \Delta_n}|+\rangle^{\otimes \Delta_{n-1}}$$
$$= \sum_{c_n \in C_n} |c_n\rangle\langle c_n|+\rangle^{\otimes \Delta_n}|\partial c_n\rangle^{(X)}$$
$$= \frac{1}{2^{|\Delta_n|/2}} \sum_{c_n \in C_n} |c_n\rangle|\partial c_n\rangle^{(X)} ,$$
(F1)

where the superscript $(X)$ indicates that the state is in the $X$-eigenbasis. To introduce tensors, we reexpand the expression in the $X$-eigenbasis for both $n$- and $(n-1)$-cells by applying $\sum_{c_n'} |c_n'\rangle^{(X)(X)}\langle c_n'|$ and $\sum_{c_{n-1}} |c_{n-1}\rangle^{(X)(X)}\langle c_{n-1}|$ from the left. We obtain

$$|\text{gCS}_{d,n}\rangle = \frac{1}{2^{|\Delta_n|/2}} \sum_{c_n,c_n',c_{n-1}} {}^{(X)}\langle c_n'|c_n\rangle$$
$$\times {}^{(X)}\langle c_{n-1}|\partial c_n\rangle^{(X)} \cdot |c_n'\rangle^{(X)}|c_{n-1}\rangle^{(X)} . \text{ (F2)}$$

We write $c_n = \sum_{\sigma_n} \gamma_{\sigma_n}\sigma_n$, $c_n' = \sum_{\sigma_n} \beta_{\sigma_n}\sigma_n$, and $c_{n-1} = \sum_{\sigma_{n-1}} \alpha_{\sigma_{n-1}}\sigma_{n-1}$. Let $\delta(\bullet,\bullet)$ denote the Kronecker delta $\delta_{\bullet,\bullet}^{\text{mod }2}$. Then we have ${}^{(X)}\langle c_n'|c_n\rangle = 2^{-|\Delta_n|/2} \prod_{\sigma_n} (-1)^{\beta_{\sigma_n}\gamma_{\sigma_n}}$, ${}^{(X)}\langle c_{n-1}|\partial c_n\rangle^{(X)} =$

$\prod_{\sigma_{n-1}} \delta(\alpha_{\sigma_{n-1}}, \sum_{\sigma_n} a(\partial\sigma_n;\sigma_{n-1})\gamma_{\sigma_n})$, so that

$$|\text{gCS}_{d,n}\rangle = \frac{1}{2^{|\Delta_n|}} \sum_{\alpha,\beta,\rho} T_{\sigma_n}[\beta_{\sigma_n}]_{\{\rho_{\sigma_n,\sigma_{n-1}}|\sigma_{n-1}\supset\partial\sigma_n\}}$$
$$\times T_{\sigma_{n-1}}[\alpha_{\sigma_{n-1}}]_{\{\rho_{\sigma_n,\sigma_{n-1}}|\partial\sigma_n\supset\sigma_{n-1}\}}$$
$$\times |\{\beta_{\sigma_n}\}\rangle^{(X)}|\{\alpha_{\sigma_{n-1}}\}\rangle^{(X)} .$$
(F3)

Here the sum is over $\alpha \in \{0,1\}^{|\Delta_{n-1}|}$, $\beta \in \{0,1\}^{|\Delta_n|}$, and $\rho \in \{0,1\}^{|\{(\sigma_n,\sigma_{n-1})|\partial\sigma_n\supset\sigma_{n-1}\}|}$. The tensors are given explicitly as

$$T_{\sigma_n}[\beta_{\sigma_n}]_{\{\rho_{\sigma_n,\sigma_{n-1}}|\sigma_{n-1}\subset\partial\sigma_n\}}$$
$$= \sum_{\gamma_{\sigma_n}=0,1} (-1)^{\beta_{\sigma_n}\gamma_{\sigma_n}} \prod_{\sigma_{n-1}\subset\partial\sigma_n} \delta(\rho_{\sigma_n,\sigma_{n-1}},\gamma_{\sigma_n}) \quad \text{(F4)}$$

and

$$T_{\sigma_{n-1}}[\alpha_{\sigma_{n-1}}]_{\{\rho_{\sigma_n,\sigma_{n-1}}|\partial\sigma_n\supset\sigma_{n-1}\}}$$
$$= \delta(\alpha_{\sigma_{n-1}}, \sum_{\sigma_n:\partial\sigma_n\supset\sigma_{n-1}} \rho(\sigma_n,\sigma_{n-1})) . \quad \text{(F5)}$$

For the the tensor $T_{\sigma_n}[\beta_{\sigma_n}]$ at the $n$-cell $\sigma_n$, we have index $\rho_{\sigma_n,\sigma_{n-1}}$ for each $(n-1)$-cell contained in the boundary of $\sigma_n$. For the tensor $T_{\sigma_{n-1}}[\alpha_{\sigma_{n-1}}]$ at the $(n-1)$-cell $\sigma_{n-1}$, we have such an index for each $n$-cell that contains $\sigma_{n-1}$ in its boundary.

As an example, let us consider gCS$_{1,1}$, i.e., the standard one-dimensional cluster state. The above reduces to the matrix product state representation given by the matrices

$$T_e[0] = \mathbf{1} , \quad T_e[1] = \sigma_z ,$$
$$T_v[0] = \mathbf{1} , \quad T_v[1] = \sigma_x , \quad \text{(F6)}$$

for $e \in \Delta_1$ and $v \in \Delta_0$. For neighboring $v$ and $e$, we have

$$T_v[0]T_e[0] = \mathbf{1} , \; T_v[0]T_e[1] = \sigma_z ,$$
$$T_v[1]T_e[0] = \sigma_x , \; T_v[1]T_e[1] = \sigma_x\sigma_z , \quad \text{(F7)}$$

where $\sigma_x$ and $\sigma_z$ are Pauli matrices. We have the relation

$$(-1)^{a\alpha_v}(-1)^{b\beta_e}T_v[\alpha_v]T_e[\beta_e]$$
$$= U(a,b)\, T_v[\alpha_v]T_e[\beta_e]\, U(a,b)^{-1} , \quad \text{(F8)}$$

where $U(a,b) = \sigma_z^a\sigma_x^b$ is a non-trivial projective representation of $\mathbb{Z}_2 \times \mathbb{Z}_2$. As noted in [83], such a relation can be used to show that the cluster state is in a non-trivial SPT order.

Coming back to the general $(d,n)$, for given $\sigma_{n-1}$, let $\sigma_n'$ be any of the $n$-cells such that $\partial\sigma_n' \supset \sigma_{n-1}$. The tensors $T_{\sigma_n}$ and $T_{\sigma_{n-1}}$ satisfy the relations

$$(-1)^{\beta_{\sigma_n}}T_{\sigma_n}[\beta_{\sigma_n}]_{\{\rho_{\sigma_n,\sigma_{n-1}}|\sigma_{n-1}\subset\partial\sigma_n\}}$$
$$= T_{\sigma_n}[\beta_{\sigma_n}]_{\{\rho_{\sigma_n,\sigma_{n-1}}+1|\sigma_{n-1}\subset\partial\sigma_n\}} \quad \text{(F9)}$$

and

$$T_{\sigma_{n-1}}[\alpha_{\sigma_{n-1}} + 1]_{\{\rho_{\sigma_n,\sigma_{n-1}}|\partial\sigma_n \supset \sigma_{n-1}\}}$$
$$= T_{\sigma_{n-1}}[\alpha_{\sigma_{n-1}}]_{\{\rho_{\sigma_n,\sigma_{n-1}} + \delta_{\sigma_n,\sigma'_n}|\partial\sigma_n \supset \sigma_{n-1}\}} , \qquad \text{(F10)}$$

which guarantee the invariance of $|\text{gCS}_{d,n}\rangle$ under $K(\sigma_n)$. Similarly, for given $\sigma_n$, let $\sigma'_{n-1}$ be any of the $(n-1)$-cells such that $\sigma'_{n-1} \subset \partial\sigma_n$. The same tensors also satisfy the relations

$$T_{\sigma_n}[\beta_{\sigma_n} + 1]_{\{\rho_{\sigma_n,\sigma_{n-1}}|\sigma_{n-1}\subset\partial\sigma_n\}}$$
$$= (-1)^{\rho_{\sigma_n,\sigma'_{n-1}}} T_{\sigma_n}[\beta_{\sigma_n}]_{\{\rho_{\sigma_n,\sigma_{n-1}}|\sigma_{n-1}\subset\partial\sigma_n\}} , \qquad \text{(F11)}$$

and

$$(-1)^{\alpha_{\sigma_{n-1}}} T_{\sigma_{n-1}}[\alpha_{\sigma_{n-1}}]_{\{\rho_{\sigma_n,\sigma_{n-1}}|\partial\sigma_n\supset\sigma_{n-1}\}}$$
$$= \prod_{\partial\sigma_n \supset \sigma_{n-1}} (-1)^{\rho_{\sigma_n,\sigma_{n-1}}}$$
$$\times T_{\sigma_{n-1}}[\alpha_{\sigma_{n-1}}]_{\{\rho_{\sigma_n,\sigma_{n-1}}|\partial\sigma_n\supset\sigma_{n-1}\}} , \qquad \text{(F12)}$$

which guarantee the invariance of $|\text{gCS}_{d,n}\rangle$ under $K(\sigma_{n-1})$.

Let us consider a $d$-dimensional lattice, which is periodic in the $(x^2, x^3, \ldots, x^d)$-directions with length $L_i = 1$ ($i = 2, \ldots, d$). Let us consider a unit hypercube $\{(x^1, \ldots, x^d) \mid 0 \le x^i < 1 \text{ for } i = 1, \ldots, d\}$ in the lattice.

(Note that we allow $x^i$ to be 0 but do not allow it to be 1.) We wish to consider an $n$-brane operator extended in the $(x^1, x^2, \ldots, x^n)$-directions and a $(d - n + 1)$-brane operator extended in the $(x^1, x^{n+1}, \ldots, x^d)$-directions. The former corresponds to the $n$-cell $\sigma_n$ (unique in the periodically identified hypercube) extended in the $(x^1, x^2, \ldots, x^n)$-directions, while the latter corresponds to the unique $(n - 1)$-cells $\sigma_{n-1}$ (at $x^1 = 0$) extended in the $(x^2, \ldots, x^n)$-directions. Let us consider the corresponding product of tensors $T_{\sigma_n}[\beta_{\sigma_n}]$ and $T_{\sigma_{n-1}}[\alpha_{\sigma_{n-1}}]$ with the shared indices such as $\rho_{\sigma_n,\sigma_{n-1}}$ summed over the values 0 and 1. This product satisfies a relation that naturally generalizes (F8), as a consequence of the relations (F9)-(F12). The projective representation $U(a, b)$ acts on one index of $T_{\sigma_{n-1}}[\alpha_{\sigma_{n-1}}]$ corresponding to the negative $x^1$-direction and another index of $T_{\sigma_n}[\beta_{\sigma_n}]$ corresponding to the positive $x^1$-direction. We note that in (F9) and (F12), the transformation of the tensors is expressed in terms of indices, which contain those in directions other than the $x^1$-direction. One can confirm in (F3) that such effect can be absorbed by redefinition of summed indices of other tensors associated with qubits living on the $n$- and $(d - n + 1)$-branes. We expect that more general tensor networks obeying the same relation provide deformations of states away from $|\text{gCS}_{(d,n)}\rangle$ within the SPT phase, and speculate that they possess the ability to simulate the lattice model $M_{(d,n)}$ via measurements.

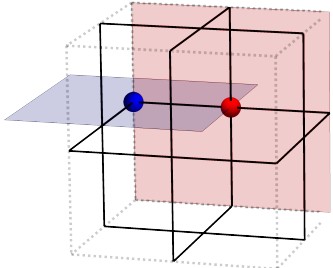

FIG. 10. (Color online) An example of the set-up considered in the analysis of the projective representation. We depict a three dimensional unit lattice with $x^2$- and $x^3$-directions taken periodic. The red vertical plane represents the 2-brane, and the blue horizontal one represents the dual 2-brane. The 2-cell (red ball) and the 1-cell (blue ball) are transformed with the global symmetry. The two branes intersect along the $x^1$-direction and the product of two tensors have two indices in this direction (left and right) on which $\mathbb{Z}_2 \times \mathbb{Z}_2$ acts in a projective representation.