# Peer review of "Measurement-based quantum simulation of Abelian lattice gauge theories"

_SciPost Physics_

## Round 1 · Referee Report · Anonymous (Referee 1) · 2022-12-5

Strengths

1) detailed presentation of a novel simulation scheme for lattice gauge theories using measurement-based quantum computations (MBQC) 2) in-depth and comprehensive study of relevant topics in this context, such as generalizations to different gauge groups, error corrections, imaginary time evolution, dualities 3) meaningful figures help in understanding of measurement pattern

Weaknesses

1) other recent theoretical developments/concepts in the context of quantum simulation for LGTs could be cited slightly more extensively (in the introduction)

Report

Gauge theories are of fundamental importance in particle and condensed matter physics. The rapid progress of quantum technologies promises to use experimental schemes for their quantum simulation. In this line of research, the authors provide in this article a novel measurement-based quantum simulation (MBQS) framework for lattice gauge theories, in which the fundamental idea is to implement (unitary) operators via sequential measurements.

In particular, the authors introduce generalized cluster states as the underlying resource states and then outline the Hamiltonian simulation of "generalized Ising models", leading to the class of M(d,n) models. The focus is on Z_2 and Z_N symmetry in arbitrary spacetime dimensions. The authors discuss comprehensively a detailed measurement pattern for the simulation protocol and show how the Gauss law can be enforced and corrected against errors. It is also demonstrated that imaginary time evolution can be performed, which is relevant for ground state preparations. Moreover, a MBQS scheme for a fermionic Majoranca model, and an interpretation in terms of a bulk-boundary correspondence are provided.

I consider the present article as a important and major contribution, which opens a promising new pathway for the simulation of lattice gauge theories. The authors provide a very comprehensive study of many important topics connected to the new MBQS scheme, which are relevant for practical implementations. While the concept of MBQC was invented around two decades ago, it was discovered only very recently that there could be a connection to gauge theories. In this vein, the authors make an important and timely contribution in view of the experimental advancement of NISQ technologies as promising platforms for practical implementations. The article has the potential for important follow-up work.

Requested changes

1) Ref. [42] is an important work, in which it was discovered for the first time that MBQC has an underlying gauge theory, and related topics such as SPT order have been discussed. It deserves to be cited as such more prominently in the introduction.

---

## Round 1 · Referee Report · Anonymous (Referee 2) · 2023-1-14

Strengths

1- The authors develop a novel measurement-based (MB) algorithm for the evolution of Abelian lattice gauge theories (in particular real- , but also imaginary-time) . They further perform a thorough analysis of how to suppress Gauss' law violating errors within their approach.

2- Additionally, they lay out interesting connections between the required generalised cluster states, their symmetries and SPT orders, and the MB simulation in detail.

3- The article is comprehensive and almost fully self-contained.

Weaknesses

1- The manuscript's length, together with a lot of technical details, makes it less easy to read.

Report

The quantum simulation of lattice gauge theories (LGTs) is a prominent example where future large-scale quantum computers are expected to outperform classical simulations. The authors' algorithms, which employ MB quantum computing to realise the quantum dynamics of Abelian LGTs (and related models), add a new perspective to this field that hasn't been much explored yet. The manuscript therefore meets SciPost's expectations to open a new pathway (here: to quantum simulate LGTs) with clear potential for follow-ups. Since it also meets all general acceptance criteria, I recommend the manuscript for publication.

Nevertheless, I have a few minor remarks and questions. I believe that the authors could further improve their manuscript and put it better into perspective by addressing the following points:

  1. How does MBQS - in terms of required resources - compare to standard approaches such as circuit-based QC with the same Trotter decomposition for real-time evolution? Naively, one trades circuit depth for more qubits, but a more quantitative statement could be helpful.

  2. For the imaginary-time evolution, can one make a connection to measurement-based VQE [PRL 126, 220501 (2021)]?

  3. I would like to point out that the suppression of gauge-variant contributions (e.g. by adding penalties) has a much longer history than the recent papers [26,27] cited in the introduction and in section III.B.C. For example, the static energy penalty was already suggested in early analog implementation proposals such as [Phys. Rev. Lett. 109, 175302 (2012)], but there also exist dynamical methods such as [Phys. Rev. Lett. 112, 120406 (2014), Phys. Rev. D 107, 014506 (2022)]. More generally, the stability of gauge invariance against violating terms has been known for a long time [Physics Letters B, 94(2), 135-140 (1980)].

  4. In section V.B, the authors generalise the results of Ref. [23] to show that the partition function Z has a natural realisation in terms of certain amplitudes such as Eq. (144). Combining the MB protocol with the tensor network representation representation of the generalised cluster states should give a tensor network representation for Z itself. At the same time, it is known that certain tensor networks that yield efficient state representations can also be interpreted as approximations of Euclidean path integrals also (see, e.g., arXiv:1807.02501). Could the authors comment on a possible connection between MBQS and tensor network states in the present context?

Requested changes

1- Adress at least some of the questions and comments of my above report in order to put the paper into a broader perspective.

2- Given the length of the manuscript, its readability might be improved by expanding and restructuring the introduction, e.g. by dedicating an explicit overview subsection that summarizes the main results.

3- To avoid that readers who are less familiar with algebraic topology get lost in the initial notations, I suggest to add a figure that illustrate cells and chains already in section II.A.

---

## Editorial Decision

resubmitted